



# An improved conceptual model of Quaternary global ice volume and the Mid-Pleistocene Transition

Felix Pollak[1,2], Frédéric Parrenin[1], Émilie Capron[1], Zanna Chase[2,3], Lenneke Jong[4,5], and
Etienne Legrain[6,7]

[1]Université Grenoble Alpes, CNRS, INRAE, IRD, Grenoble INP, IGE, Grenoble, France
[2]Institute for Marine and Antarctic Studies, University of Tasmania, Hobart, TAS, Australia
[3]Australian Research Council Centre for Excellence in Antarctic Science (ACEAS), University of Tasmania, Hobart, TAS, Australia
[4]Australian Antarctic Division, Department of Climate Change, Energy, the Environment and Water, Kingston, TAS, Australia
[5]Australian Antarctic Program Partnership, Institute for Marine and Antarctic Studies, University of Tasmania, Hobart, TAS, Australia
[6]Laboratoire de Glaciologie, Université libre de Bruxelles, Brussels, Belgium
[7]Department of Water and Climate, Vrije Universiteit Brussel, Brussels, Belgium

**Correspondence:** Felix Pollak (felix.pollak@univ-grenoble-alpes.fr)

**Abstract.** During the Quaternary period, spanning the last 2.6 million years, the characteristic frequency and amplitude of glacial-interglacial cycles evolved from low-amplitude 41,000-year cycles to high-amplitude 100,000-year cycles. This transition occurred around 1.2 to 0.8 million years ago and is referred to as the Mid-Pleistocene Transition (MPT). While the 41 kyr cycles are driven by changes in Earth's obliquity, which largely affect the incoming solar insolation, no apparent change

5   in external orbital forcing during this period could explain the shift towards 100 kyr cycles. Several theories have been put forward to explain this shift, including scenarios of both gradual and abrupt changes in the internal climate system throughout the Pleistocene. In order to test which theory best matches the observations, we have constructed a conceptual model capable of simulating changes in the global ice volume over the past 2.6 Ma and accurately reconstructing the MPT and its associated change in amplitude and frequency. Four different forcing scenarios are implemented, ranging from a purely orbitally driven

10  model to a ramp-like change in internal forcing. The model is in favour of a ramp-like forcing scenario, where the gradual change in internal forcing is limited in time and started around 2 Ma. These findings imply that the climate system had already undergone major changes in the early Pleistocene and support the idea of a long-term climatic shift as a cause of the MPT. For the best-performing model, we included an ice volume dependency in the state thresholds, demonstrating that glacial terminations during the past 900 ka are mainly driven by precession rather than obliquity.

## 15   1   Introduction

The Quaternary is the most recent geological epoch, covering the last 2.6 Ma. It is characterized by the alternance of cold glacial climate states and warmer interglacial periods. Glaciers and ice sheets expanded during glacial periods and retreated during interglacial periods. In the first half of the 20th century, Milutin Milankovitch made significant contributions to the astronomical



theory of climate, which links changes in Earth's orbital parameters to changes in the radiative forcing, which ultimately leads
20   to glacial-interglacial variability (Milankovitch, 1941). In his theory, the orbital variations of Earth's eccentricity, obliquity
and precession lead to variations in the insolation during boreal summer, which is the dominating driver of glacial cycles
(Ganopolski, 2024). Earth's obliquity varies on a cycle of approximately 41 kyr, while precession operates over a 19 kyr and a
kyr period, and eccentricity fluctuates on a 100 kyr and a 400 kyr timescale (Hays et al., 1976).

Two central problems arise from this theory: The first is known as the 100 kyr problem. The late Quaternary glacial cycles
(approximately the last 800 ka) were characterized by 100 kyr cycles, closely following the eccentricity signal. They show
a "sawtooth" pattern of long, roughly 90 kyr long glaciations, followed by short, roughly 10 kyr long deglaciations (Raymo
and Huybers, 2008). However, changes in insolation are mainly dominated by the obliquity and precession signal, while the
influence of eccentricity on the annual averaged global insolation is negligible (Imbrie et al., 2011; Barker et al., 2022). So
why is there a 100 kyr signal in the glacial cyclicity? This underlines the more complex interplay between eccentricity and
climate and the possible existence of feedback mechanisms or other non-linearities, which are needed to explain this influence
(Paillard, 2015; Imbrie et al., 2011).

The second issue is related to the shift of low-amplitude roughly 41 kyr cycles towards high-amplitude roughly 100 kyr
cycles during the mid-Pleistocene ($\sim$1.2 - 0.8 Ma) in the absence of any change in orbital forcing. This transition is known as
the Mid-Pleistocene Transition (MPT) (Imbrie et al., 2011; Barker and Knorr, 2023; Elderfield et al., 2012). The occurrence
of the MPT poses one of the most challenging open questions to the paleoclimatological community and has been the subject
of intense studies (Willeit et al., 2019; Legrain et al., 2023; Berends et al., 2021b; Ganopolski, 2024). Due to the lack of any
apparent change in orbital forcing, various internal feedback mechanisms of the climate system and non-linearities have been
put forward to explain this major climatic shift (Legrain et al., 2023; Berends et al., 2021b).

Clark and Pollard (1998) proposed the so-called regolith hypothesis as a potential cause of the MPT. Before the onset of the
Northern Hemisphere glaciations, 2.6 Ma, the North American and Eurasian surface was covered by a thick layer of regolith
(10-50 m), which was formed due to exposure to weathering over the last $10^7 - 10^8$ years. The lower friction and deformability
of the regolith increased the basal flow velocity, leading to thinner and wider ice sheets, which spread further south during the
early Pleistocene. Hence, the enlarged ablation zones became more sensitive to insolation, resulting in the observed obliquity-
driven 41 kyr world. The continuous glaciations gradually removed the regolith beneath, exposing the unweathered high-
friction crystalline bedrock, corresponding to reduced basal velocities and a reduced sensitivity to insolation. The increased
ice sheet stability would have resulted in the observed late-Pleistocene $\sim$100 kyr glacial-interglacial periodicity (Clark et al.,
2006; Berends et al., 2021b; Willeit et al., 2019).

Another commonly proposed hypothesis to explain the MPT concerns a gradual cooling trend due to a continuous long-
term decrease of atmospheric $CO_2$ concentrations throughout the Quaternary (Berends et al., 2021b; Willeit et al., 2019;
Scherrenberg et al., 2024). Based on such a long-term cooling trend, various feedback mechanisms, like ice sheet feedback
mechanisms or changes in the ocean circulations, have been proposed as a secondary trigger for the MPT (Berends et al.,
2021b).



Another open question is whether an abrupt or gradual scenario triggered the MPT. While a gradual change involves a linear change of a climatic parameter over the entire Quaternary, an abrupt scenario refers to the crossing of a climatic threshold,
which leads to a new, irreversible climate state. A scenario leading to such an abrupt transition is considered to have lasted only a limited time and not over the entire Quaternary (Legrain et al., 2023). While the $CO_2$ hypothesis is an example of a gradual theory, one proposed abrupt mechanism that could have contributed to the MPT is based on non-linear feedback effects between the ice sheets and the global climate. In particular, it has been proposed that the merging of the North American Laurentide and Cordilleran ice sheets posed a strong feedback on the glacial cycles (Berends et al., 2021b; Bintanja and van de Wal,
2008; Gregoire et al., 2012). A long-term cooling trend is suggested to have allowed the North American ice sheets to cross the threshold of around 45 m sl (meter sea level equivalent), needed to merge the Laurentide and Cordilleran ice sheets for the first time around 1 Ma (Bintanja and van de Wal, 2008; Berends et al., 2021b). While their merging would have led to further ice volume increase without any change in the solar forcing (Berends et al., 2021b), the increased ice volume would exhibit instabilities once exceeding 70 m sl, leading to rapid deglaciations triggered by the next insolation peak, resulting in
the observed 100 kyr cycles of the post-MPT world (Bintanja and van de Wal, 2008; Gregoire et al., 2012).

Climate models of varying complexity present a versatile tool for investigating these different hypotheses. On one end of the model hierarchy are Earth System Models that try to include the underlying physics and couple different compartments of the climate system, for example, the cryosphere with the carbon cycle to account for different feedback mechanisms. Their more realistic representation of the climate system comes at the cost of high computational demand. This is a particular
obstacle for the long simulation periods that are required ($\mathcal{O}$(Myr)), as well as for running large ensembles of simulations, needed for parameter tuning, sensitivity experiments, or investigation of various parameterizations. On the other end of the model hierarchy are so-called conceptual models that are simple, zero-dimensional (spatial dimensions) representations of the climate system. The basic idea of these types of models is to reduce the model complexity by focusing on a reduced number of highly aggregated macroscopic variables that try to reconstruct the full dynamics as closely as possible (Saltzman,
2001). Verbitsky and Crucifix (2023) point out that a mathematical model must be simple enough to physically interpret the simulation results. Due to the reduced number of model parameters, it becomes easier to interpret the influence of individual parameters or parameterizations on the simulation outcomes through their straightforward removal or inclusion. Furthermore, their reduced number of parameters makes the necessary tuning process more manageable and allows for long simulation periods ($\mathcal{O}$(Myr)). Hence, conceptual models serve as a crucial tool to investigate the MPT and glacial-interglacial variability
across the Quaternary.

Various conceptual models have been developed over the last decades, varying in their underlying assumptions and hypotheses. Many of them yield good results and can reconstruct the 100 kyr world with its characteristic saw-tooth pattern (Gildor and Tziperman, 2001; Imbrie et al., 2011; Parrenin and Paillard, 2012; Pérez-Montero et al., 2024) or even the MPT with its shift in amplitude and frequency and its specific timing (Paillard, 1998; Paillard and Parrenin, 2004; Legrain et al., 2023; Ganopolski,
85 2024).

A common approach is to attribute glacial-interglacial variability to relaxation oscillations between multiple equilibria (Paillard, 1998; Parrenin and Paillard, 2012; Legrain et al., 2023; Leloup and Paillard, 2022). In his initial work, Paillard (1998)





proposed a three-state model consisting of an interglacial, mild glacial, and a full glacial state. Transitions between model states depend on the ice volume and the insolation. To account for a change in forcing due to decreasing atmospheric $CO_2$

concentrations, he added a small linear trend in the radiative forcing and linearly increased one of the state thresholds. This model is able to accurately reconstruct the global ice volume over the past 2 Ma, with good results in the timing of terminations and the change in periodicity due to the MPT. In later work, Parrenin and Paillard (2012) presented an improved version of this model, which now only consists of a glaciation and a deglaciation state. While the deglaciation trigger depends on a combination of ice volume and insolation, the glacial inception is solely controlled by insolation. Moreover, they adapted the solar

forcing, such that the model takes a linear combination of three orbital parameters instead of a fixed insolation curve. A similar approach was used in the model of Imbrie et al. (2011). Indeed, the model results seem to depend on the chosen insolation forcing, since they differ in their contributions from obliquity and precession (Leloup and Paillard, 2022). Legrain et al. (2023) continued the work by implementing different internal forcing scenarios to test which of them is most likely to reproduce the MPT. In addition to the external solar forcing, they added an internal forcing to the model in the form of a varying deglaciation

threshold. They found that a gradual increase of this deglaciation threshold is more likely to reproduce the MPT than an abrupt change. Based on this finding, they suggested that a gradual decline in atmospheric $CO_2$ concentrations over the Pleistocene may have increased the deglaciation threshold, potentially causing the MPT.

Gildor and Tziperman (2001) used an idealized model to study the influence of different sea ice feedback mechanisms to explain the 100 kyr cycles. They suggested that the large extent of sea ice during late glacial periods would have reduced atmo-

spheric temperatures, humidity, and evaporation. Thus, the available precipitation would have decreased, leading to negative mass balances and quick terminations. Their model links the 100 kyr cycles fully to this sea ice switch, in the absence of any external forcing. However, their model fails to reproduce some key climatic features because of its high idealization, hence missing feedback mechanisms other than those related to sea ice changes.

Imbrie et al. (2011) proposed a phase-space model that again combines ice volume and orbital forcing as a deglaciation trig-

ger. They were able to reproduce the shift in frequency of glacial-interglacial cycles purely by orbital forcing, without changing any model parameters during the MPT. Hence, they linked the occurrence of 100 kyr glacial cycles to the eccentricity-driven amplitude modulation of precession. However, their model-data comparison was limited to a detrended benthic $\delta^{18}O$ curve rather than an ice volume reconstruction, thereby precluding the assessment of whether their model accurately reconstructs the change in amplitude over the MPT.

In a more recent work, Ganopolski (2024) attempted to set up a generalized Milankovitch Theory by incorporating model results from CLIMBER-2 (a more sophisticated Earth-System Model) into a conceptual model. The model can reproduce the glacial cycles of the Quaternary based on the nonlinear response of the climate system to the orbital forcing in the form of the eccentricity-driven amplitude modulation of precession and the existence of supercritical ice sheets. A gradual increase of the critical ice volume is added to model the MPT. The author associates this with the gradual removal of terrestrial sediments in

the Northern Hemisphere, which is needed to prolong the late-Pleistocene glacials.

Due to their higher complexity, it is more difficult to identify the main mechanisms driving the glacial-interglacial variability for more comprehensive climate models, which explicitly resolve various physical processes to improve the model accuracy. In





contrast, the strength of conceptual models is their simple conceptual framework, which allows for straightforward isolation of the key mechanisms, but therefore, they often lack a profound physical basis or depend on mathematical thresholds. In recent work, Pérez-Montero et al. (2024) presented a more sophisticated conceptual model to bridge these two approaches. Their model aims to simulate the interactions between the climate and the Northern Hemisphere ice sheets, despite a reduced spatial dimensionality in the fundamental equations. They obtain good results for various paleoclimatic records. However, they only focused on the last 800 ka.

Some of the above-mentioned conceptual models are limited by short simulation periods, which do not cover the MPT (Parrenin and Paillard, 2003; Pérez-Montero et al., 2024) or they rely on a single insolation metric for the orbital forcing (Ganopolski, 2024; Leloup and Paillard, 2022; Parrenin and Paillard, 2003; Paillard, 1998). In this study, we use the well-suited conceptual modelling approach to investigate the climatic variations of the Quaternary. Our objective is to develop a versatile conceptual model that can be easily applied to different forcing scenarios and can be run on different simulation periods, covering the entire MPT. Moreover, we reduce model biases by not relying on a single insolation metric, but rather use a linear combination of different orbital parameters. To account for potential feedback mechanisms in the climate system, we implement various internal forcing scenarios in the model, ranging from a purely orbital model to a ramp-like change in internal forcing. Here, the focus is on the general temporal structure of such a potential feedback mechanism, which allows for its physical interpretation. Our model is a continuation of the 2-state Legrain (2023) model and simulates the change in global ice volume. The modular design of our model facilitates easy adaptation to a wide range of research questions. We demonstrate this flexibility by modifying the model to investigate the sensitivity of large ice sheets to changes in orbital parameters during the late Pleistocene. Furthermore, we extrapolate the model to project glacial-interglacial variability over the next 250 kyr.

## 2 Methods

### 2.1 Base models

The model used in this study is an improved version of the conceptual model developed by Legrain et al. (2023), which is based on the work by Parrenin and Paillard (2012). The model is externally forced by obliquity and precession to simulate the global ice volume $v(t)$ in meter sea-level equivalent (m sl) over the Pleistocene. As input, it uses a linear combination of three orbital parameters normalized to zero mean and unit variance that can reproduce the insolation at most latitudes and seasons (Loutre, 1993; Imbrie et al., 2011). The following orbital parameters (dimensionless) are used:

$$\text{Esi: Precession} \sim e\sin(\omega), \tag{1}$$

$$\text{Eco: Phase-shifted precession} \sim e\cos(\omega), \tag{2}$$

$$\text{Ob: Obliquity} \sim \epsilon, \tag{3}$$

with $\omega$ the precession angle taken from the vernal equinox and $e$ the eccentricity.



The model has two different states, the glaciation state (**g**) and the deglaciation state (**d**). Therefore, the ice volume evolution $v(t)$ is driven by two linear differential equations, depending on the current model state:

$$\text{g}: \frac{\mathrm{d}v(t)}{\mathrm{d}t} = \underbrace{-\alpha_{\mathrm{Esi}}\mathrm{Esi}(t) - \alpha_{\mathrm{Eco}}\mathrm{Eco}(t) - \alpha_{\mathrm{O}}\mathrm{Ob}(t)}_{=-I_\alpha(t)} + \alpha_{\mathrm{g}} = -I_\alpha(t) + \alpha_g, \tag{4}$$

$$\text{d}: \frac{\mathrm{d}v(t)}{\mathrm{d}t} = \underbrace{-\alpha_{\mathrm{Esi}}\mathrm{Esi}(t) - \alpha_{\mathrm{Eco}}\mathrm{Eco}(t) - \alpha_{\mathrm{O}}\mathrm{Ob}(t)}_{=-I_\alpha(t)} + \alpha_{\mathrm{d}} - \frac{v(t)}{\tau_{\mathrm{d}}(t)} = -I_\alpha(t) + \alpha_{\mathrm{d}} - \frac{v(t)}{\tau_{\mathrm{d}}(t)}, \tag{5}$$

where $\alpha_{\mathrm{Esi}}$, $\alpha_{\mathrm{Eco}}$, $\alpha_{\mathrm{O}}$, $\alpha_{\mathrm{g}}$ and $\alpha_{\mathrm{d}}$ are constant model parameters in $\mathrm{m\,kyr^{-1}}$. $\tau_{\mathrm{d}}(t)$ is the relaxation time in $\mathrm{kyr}$.

A state change from a glaciation to a deglaciation (**g**) $\rightarrow$ (**d**) occurs when a combination of the current ice volume and the orbital forcing exceeds a critical threshold. Hence, a termination can be triggered for moderate orbital values if the ice volume is large, or vice versa when the current ice volume is moderate, but the orbital forcing is large:

$$\text{(i)} \qquad v(t) + \underbrace{(k_{\mathrm{Esi}}\mathrm{Esi}(t) + k_{\mathrm{Eco}}\mathrm{Eco}(t) + k_{\mathrm{O}}\mathrm{Ob}(t))}_{=I_k(t)} = v(t) + I_k(t) > v_0(t),$$

$$\text{(ii)} \qquad \underbrace{k_{\mathrm{Esi}}\mathrm{Esi}(t) + k_{\mathrm{Eco}}\mathrm{Eco}(t) + k_{\mathrm{O}}\mathrm{Ob}(t)}_{=I_k(t)} = I_k(t) \geq v_1, \tag{6}$$

where $k_{\mathrm{Esi}}$, $k_{\mathrm{Eco}}$ and $k_{\mathrm{O}}$ are constant model parameters in $\mathrm{m}$. While $v_0(t)$ is the deglaciation parameter in $\mathrm{m}$ and can vary in time depending on the specific model configuration, $v_1$ is another model threshold in $\mathrm{m}$ but constant in time.

Conversely, a transition from a deglaciation state to a glaciation state (**d**) $\rightarrow$ (**g**) occurs when:

$$\text{(i)} \qquad \underbrace{k_{\mathrm{Esi}}\mathrm{Esi}(t) + k_{\mathrm{Eco}}\mathrm{Eco}(t) + k_{\mathrm{O}}\mathrm{Ob}(t)}_{=I_k(t)} = I_k(t) < v_1,$$

$$\text{(ii)} \qquad v(t) + \underbrace{(k_{\mathrm{Esi}}\mathrm{Esi}(t) + k_{\mathrm{Eco}}\mathrm{Eco}(t) + k_{\mathrm{O}}\mathrm{Ob}(t))}_{=I_k(t)} = v(t) + I_k(t) \leq v_0(t). \tag{7}$$

In addition to the external forcing described by the orbital parameters, the model can also incorporate an internal forcing mechanism to account for non-linear feedback mechanisms within the climate system. The model has four different configurations:

(i) **ORB**: The orbital model is purely orbitally driven and the deglaciation parameter $v_0(t) = v_0$ and the relaxation time $\tau_d(t) = \tau_d$ are constant in time.

(ii) **ABR**: The abrupt model includes an abrupt change at $t_{\mathrm{abr}}$ for the deglaciation parameter and the relaxation time:

$$v_0(t) = \begin{cases} v_0', & \text{if } t \geq t_{abr} \\ v_0, & \text{if } t < t_{abr} \end{cases} \qquad\qquad \tau_d(t) = \begin{cases} \tau_d', & \text{if } t \geq t_{abr} \\ \tau_d, & \text{if } t < t_{abr} \end{cases} \tag{8}$$

with $v_0'$ and $v_0$ being constant values for the deglaciation parameter before and after the abrupt shift. And with $\tau_d'$ and $\tau_d$ being constant values for the relaxation time before and after the abrupt shift.





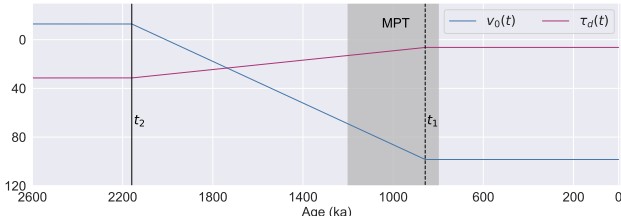

**Figure 1.** Time evolution of $v_0(t)$ and $\tau_d(t)$ for the RAMP model.

(iii) **GRAD**: The gradual model includes a gradual change of the deglaciation parameter and the relaxation time over the whole simulation period:

$$v_0(t) = v_0 - C_v t$$

$$\tau_d(t) = \tau_d - C_\tau t \tag{9}$$

where $v_0$ and $\tau_d$ are constants and define the values at $t = 0$. $C_v$ and $C_\tau$ describe the rate of change $(\mathrm{m\,kyr^{-1}})$ of the deglaciation parameter and the relaxation time, respectively.

(iv) **RAMP**: The ramp-like model includes a ramp-like change of the deglaciation parameter and the relaxation time. Both parameters change linearly during a period bounded by $t_1$ and $t_2$. Before and after this period, both parameters are constant:

$$v_0(t) = \begin{cases} v_0', & \text{if } t > t_2 \\ v_0 - \frac{v_0 - v_0'}{t_2 - t_1}\left(t - t_1\right), & \text{if } t_2 \geq t \geq t_1 \\ v_0, & \text{if } t_1 > t \end{cases} \qquad \tau_d(t) = \begin{cases} \tau_d', & \text{if } t > t_2 \\ \tau_d - \frac{\tau_d - \tau_d'}{t_2 - t_1}\left(t - t_1\right), & \text{if } t_2 \geq t \geq t_1 \\ \tau_d, & \text{if } t_1 > t \end{cases} \tag{10}$$

with $v_0'$ and $\tau_d'$ the constant values before the ramp and with $v_0$ and $\tau_d$ the constant values after the ramp. In the case that $t_1 = t_2$, the RAMP model converges towards the ABR model. If $t_2$ is set to the beginning of the simulation and $t_1 = 0$, the RAMP model converges to the GRAD model, with a linear trend over the whole simulation period. Due to this formulation, the RAMP model serves as a bridge between the ABR and the GRAD model and can be used to verify which scenario is more

likely. Furthermore, it allows us to investigate the duration and start point of the linear trend.

## 2.2 RAMP-l model

Beyond the standard RAMP model outlined above, we develop a slightly modified version. This version introduces three additional parameters, referred to as the *l*-parameters, hence it is denoted as the RAMP-l model. Here, the term $I_k$ in the





threshold equations (Eq. 6, 7) which is linked to the orbital forcing is adapted by the unitless parameters $l_{\mathrm{Esi}}$, $l_{\mathrm{Eco}}$ and $l_{\mathrm{O}}$:

$$\tilde{I}_k = [k_{\mathrm{Esi}} + l_{\mathrm{Esi}} v(t)] \, \mathrm{Esi}(t) + [k_{\mathrm{Eco}} + l_{\mathrm{Eco}} v(t)] \, \mathrm{Eco}(t) + [k_{\mathrm{O}} + l_{\mathrm{O}} v(t)] \, \mathrm{Ob}(t)$$
$$= \underbrace{k_{\mathrm{Esi}} \mathrm{Esi}(t) + k_{\mathrm{Eco}} \mathrm{Eco}(t) + k_{\mathrm{O}} \mathrm{Ob}(t)}_{=I_k(t)} + \underbrace{l_{\mathrm{Esi}} v(t) \mathrm{Esi}(t) + l_{\mathrm{Eco}} v(t) \mathrm{Eco}(t) + l_{\mathrm{O}} v(t) \mathrm{Ob}(t)}_{=I_l(t)}$$
$$= I_k(t) + I_l(t) \, . \tag{11}$$

This new parameterization adds to the $I_k$ term another term, called $I_l$, which depends on the ice volume. A larger ice volume leads to a larger $I_l$ term. With the increasing global ice volume during the late Quaternary, this term is becoming more significant than it was in the early Quaternary. The multiplication of the ice volume and the orbital parameters ensures that these three

additional terms mainly affect the forcing at time points of large ice volume and extrema in the respective orbital parameter. Therefore, this parameterization implements the sensitivity of large ice volumes to changes in orbital parameters.

To investigate the influence of the different orbital forcing scenarios, we do not include this model for the comparison of the four standard models (ORB, ABR, GRAD, RAMP), since these models only differ in their internal forcing, which allows isolating this effect on the simulation outcome. We only use the RAMP-l model to identify the influence of the additional

l-parameters and to see how this affects the simulated outcomes compared to the standard RAMP model. Furthermore, we can investigate the role of the three different orbital parameters on large ice volumes and test the hypothesis that large ice volumes become more sensitive to precession in the late Quaternary, and thus, these glacial terminations are mainly determined by precession rather than obliquity (Barker et al., 2025).

## 2.3 Global ice volume reconstructions

The different model configurations are all fitted to a global sea level reconstruction by Berends et al. (2021a). Since the simulated global ice volume in the model is expressed in meter sea level equivalent (m sl), it can be directly compared to sea level curves. An optimal set of parameters is inferred by using a Monte Carlo method which best fits the model to the target sea level data (see next section). The Berends sea level data reconstructs the past 3.6 Ma, based on an inverse forward modelling approach that aims to disentangle the coupled signals of ice volume and ocean temperature, present in benthic $\delta^{18}$O records

(Berends et al., 2021a). It uses the LR04 stack of benthic $\delta^{18}$O as a forcing (Lisiecki and Raymo, 2005) and has a high temporal resolution of 100 years.

To avoid relying on a single data set, we also fit the different model configurations to the sea level reconstruction by Rohling et al. (2022). In contrast to Berends et al. (2021a), they used the process modelling approach by Rohling et al. (2021) to deconvolve the sea level signal from the LR04 (Lisiecki and Raymo, 2005) benthic $\delta^{18}$O signal (Rohling et al., 2022). Despite

their different approaches to reconstructing the global sea level signal from the LR04 stack, both produce curves in strong agreement with each other, with a mean offset of 3.3 m (Rohling et al., 2022). In the period before 2.6 Ma, the offset is greater. In general, the Berends record is smoother compared to the Rohling record, which arises from a stronger inertia in changes in the ice volume present in the Berends model (Rohling et al., 2022).



## 2.4 Parameter tuning

Depending on the configuration, the model has 12 (ORB), 14 (GRAD), 15 (ABR), 16 (RAMP), or 19 (RAMP-l) free parameters. We apply a Monte Carlo method to find an optimal set of parameters for each configuration, similar to the method described by Legrain et al. (2023). Likewise, we use the Python package *emcee* (Foreman-Mackey et al., 2013) which implements the affine invariant ensemble sampler for Markov chain Monte Carlo algorithms, introduced by Goodman and Weare (2010). Due to an increase in numerical efficiency, we use a larger number of walkers, varying between 100 and 500, to explore the parameter space more rigorously. Moreover, instead of using a simple *stretch move* (Goodman and Weare, 2010) as Legrain et al. (2023) have done, we apply a combination of different moves for the proposal. That is, a mixture of 90% *Differential Evolution (DE)* moves and 10% *DE Snooker* moves, as suggested by Ter Braak and Vrugt (2008).

Furthermore, to verify convergence and escape possible local minima, we also use the Python package *ptemcee* (Vousden et al., 2016; Foreman-Mackey et al., 2013), which implements parallel tempering in the *emcee* framework. Here, we use 20 different temperatures for the tuning. Another sampler that we implemented is *dynesty* (Speagle, 2020; Koposov et al., 2024) which is a dynamic nested sampling package in Python. This modular approach with multiple different samplers allows for simple switching between the samplers and the most suitable sampler for each new parameterization can be selected to obtain the best results.

The Log-Likelihood function for the optimization is defined as:

$$\ln \mathcal{L} = -\frac{1}{2} \sum_{i=1}^{N} \left( \frac{\hat{y}_i - y_i}{\sigma} \right)^2, \tag{12}$$

where $\hat{y}$ is the modelled ice volume, $y$ the target ice volume reconstruction and $\sigma$ its corresponding standard deviation.

## 2.5 Evaluation of model performance

The initial conceptual model based on the work by (Legrain et al., 2023) is further refined to include adjustments in its tuning, its parameterization (e.g. time-dependent $\tau_d$ parameter, deprecated truncation function), and the addition of a fourth model configuration (RAMP). To evaluate the effectiveness of these modifications, we conduct identical simulations as described in Legrain et al. (2023). Hence, we run the ORB, ABR, and GRAD models over the past 2 Ma. The resultant root mean square errors (RMSE) between the simulated global ice volumes and the reconstruction by Berends (Berends et al., 2021a) are compared with the originally reported RMSEs (Legrain et al., 2023). Furthermore, we analyze the RMSE of the new RAMP configuration relative to the three original model configurations.

To evaluate the performance of the four different model configurations, we use the common Bayesian Information Criterion (BIC) (Mitsui and Crucifix, 2017; Mitsui et al., 2022):

$$\text{BIC} = -2\ln \mathcal{L}(\hat{\theta}) + K \ln N, \tag{13}$$

where K is the number of free parameters used in the model, $N$ the number of data points and $\mathcal{L}(\hat{\theta})$ is the likelihood function of the best model parameters $\hat{\theta}$. The lower the BIC, the better it can describe the observed data points (Mitsui et al., 2022).





**Table 1.** Improved model (this study) results for 2 Myr simulation wrt original model by Legrain et al. (2023).

| Model | RMSE | RMSE | Improvement |
|---|---|---|---|
| (2 Myr simulation) | (Legrain et al., 2023) | (this study) | |
| Orbital (ORB) | 18.1 m | 16.0 m | 2.1 m |
| Abrupt (ABR) | 14.4 m | 12.0 m | 2.4 m |
| Gradual (GRAD) | 13.9 m | 11.7 m | 2.2 m |
| Ramp-like (RAMP) | NA | 11.6 m | NA |

Parameters that do not significantly improve model performance are penalized in the BIC, such that models that perform equally, but with a lower number of parameters are preferred (Mitsui et al., 2022). The difference in BIC values for two models approximates the Bayes factor for large N (Mitsui and Crucifix, 2017). Hence, the evidence in favour of model $i$ against model $j$ is quantified by the $\Delta \mathrm{BIC}_{ij}$ (Mitsui and Crucifix, 2017):

$$\Delta \mathrm{BIC}_{ij} = \mathrm{BIC}_j - \mathrm{BIC}_i. \tag{14}$$

As a general rule of thumb (Raftery, 1995; Mitsui et al., 2022), the evidence of model $i$ against model $j$ is considered weak if $0 < \Delta \mathrm{BIC}_{ij} < 2$, positive if $2 < \Delta \mathrm{BIC}_{ij} < 6$, strong if $6 < \Delta \mathrm{BIC}_{ij} < 10$ and very strong if $\Delta \mathrm{BIC}_{ij} > 10$.

## 3 Results

### 3.1 Model improvements

Table 1 presents a comparison of the RMSEs for the three distinct model configurations, as detailed in Legrain et al. (2023),
against the results obtained with the new model modifications, i.e. the time-dependent $\tau_d$ parameter, the deprecated truncation function and the enhanced tuning procedure. Each of the original model configurations (ORB, ABR, GRAD) is improved by more than 2 m when evaluated over the past 2 Ma. It remains evident that the gradual model continues to exhibit the lowest RMSE, while the orbital model exhibits the largest. Moreover, the newly incorporated RAMP model further reduces the RMSE of the previously superior GRAD model (Legrain et al., 2023) by 2.3 m.

### 3.2 Comparison of the different model configurations


Figure 2 shows the simulated global ice volume over the past 2.6 Ma for the four different model configurations.

(i) **ORB model:** The ORB model can identify most of the terminations and endpoints of the interglacials fairly accurately (Fig. 2a). The mean global ice volume over the entire Pleistocene ($\bar{v}$) is 33.5 m for the ORB model and for the Berends
reconstruction. A general trend of increasing glacial global ice volume during the late Pleistocene is also visible, while the amplitude of the glacial-interglacial cycles (GICs) stays almost constant. During the first 1 Myr of simulation, the GICs are



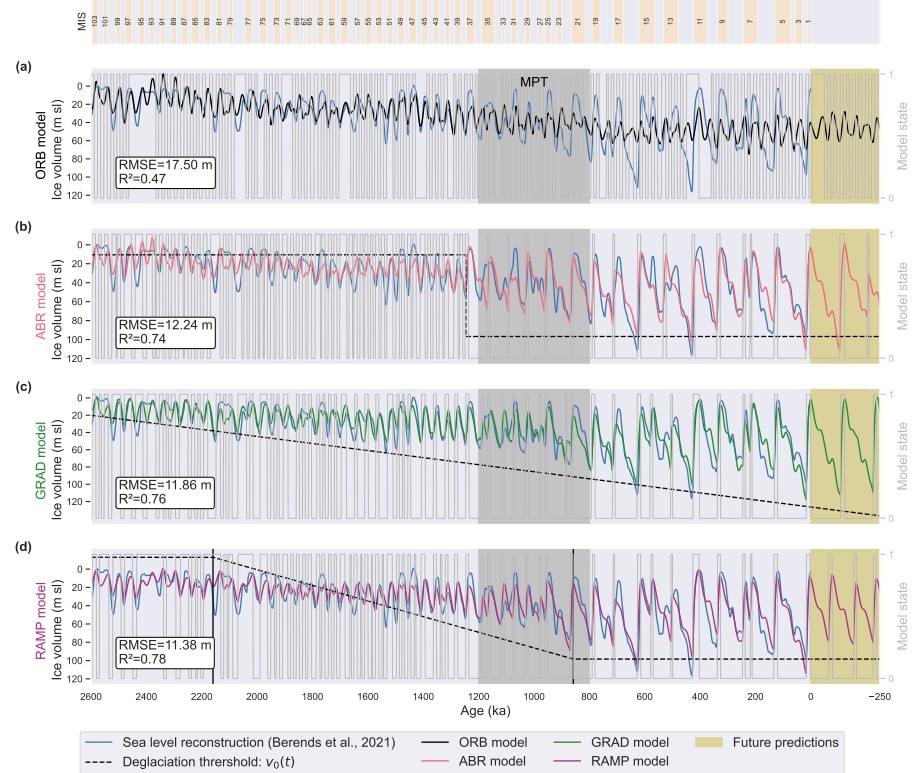

**Figure 2.** Comparison of the four different model configurations. The simulated ice volume curves are shown for the ORB (**a**, black curve), ABR (**b**, red curve), GRAD (**c**, green curve) and the RAMP model (**d**, purple curve) over the whole Pleistocene. Each simulation is plotted over the ice volume reconstruction by Berends et al. (2021a) (blue curve). The deglaciation parameter $v_0(t)$ (black dotted line) is shown for each simulation, indicating the internal forcing scenario. The yellow shaded area marks the future 250 kyr and the respective predictions by each model. The grey-shaded area highlights the classical perspective of where the MPT is located in time. MIS boundaries given according to Lisiecki and Raymo (2005).

bounded by -13 m and 43 m with a $\bar{v}$ of 20 m. The last 1 Myr are bounded by 23 m and 75 m with a $\bar{v}$ of 48 m. This corresponds to a trend of almost 30 m in $\bar{v}$ between the early and late Pleistocene. However, the ORB model does not capture the amplitude of the GICs correctly. The amplitude stays almost constant throughout the Pleistocene, which is in contrast to the observed

increase in amplitude visible in the Berends data. Although for the early Pleistocene, the minima and maxima for the global ice volume during the interglacials and glacials are still similar to the reconstructed ones, they are systematically too large during interglacials and too small during glacial periods. The ORB model cannot produce glacial maxima with more than 75 m of global ice volume. Moreover, most of its interglacials are characterized by weak melting events, which lead to an accumulation of global ice volume over the simulated period. During the last 1 Myr, the global ice volume never fell below 23 m, while the

Berends reconstruction exhibits Interglacials with almost 0 m in global ice volume.



**Table 2.** Statistical quantities for the four different model configurations for the three different simulation periods used. Shown are the root mean square error (RMSE), the coefficient of determination ($R^2$) and the Bayesian information criterion (BIC).

| | 2 Myr simulation | | | 2.6 Myr simulation | | | 3.6 Myr simulation | | |
|---|---|---|---|---|---|---|---|---|---|
| **Model** | **RMSE** | $R^2$ | **BIC** | **RMSE** | $R^2$ | **BIC** | **RMSE** | $R^2$ | **BIC** |
| Orbital (ORB) | 16.04 m | 0.56 | 978.5 | 17.50 m | 0.47 | 1469.8 | 15.69 m | 0.64 | 1389.8 |
| Abrupt (ABR) | 12.07 m | 0.75 | 616.0 | 12.24 m | 0.74 | 790.7 | 12.27 m | 0.78 | 913.0 |
| Gradual (GRAD) | 11.71 m | 0.76 | 579.1 | 11.86 m | 0.76 | 741.7 | 11.17 m | 0.82 | 769.4 |
| Ramp-like (RAMP) | 11.59 m | 0.77 | 584.9 | 11.38 m | 0.78 | 708.0 | 10.69 m | 0.83 | 730.7 |

The projected GICs for the future 250 kyr are visualized in the yellow-shaded area of Fig. 2a. The ORB model predicts six full glacial cycles for this period. They look similar to the preceding cycles, characterized by small frequencies and amplitudes. Although interglacial melting does not reduce the ice volume below 28 m, the glacial accumulation of the ice volume does not exceed 62 m. $\bar{v}$ for the next 250 kyr is predicted to be 45.9 m with the ORB model.

Quantitatively, the weak performance of the ORB model can be seen in Table 2. The coefficient of determination $R^2$ has the lowest value of 0.47 for the ORB model (2.6 Myr simulation), indicating the lowest correlation with the Berends sea level reconstruction. The same holds for the RMSE, which has the highest value of 17.50 m. Running the ORB model over a shorter (2 Myr) or longer (3.6 Myr) period does have a significant effect on its correlation with the Berends data, which is highest for the 3.6 Myr simulation and lowest for the 2.6 Myr simulation. Nevertheless, for all three simulation periods, the ORB model

exhibits the worst results.

Running the ORB model for a reduced duration of 2 Myr (Fig. B2a) or an extended duration of 3.6 Myr (Fig. B3a) considerably influences the outcomes. The simulated GICs for the 2.6 and 3.6 Myr simulations are characterized by similarly low frequencies and low amplitudes, accompanied by a general trend of increasing global ice volume throughout the simulation period. In contrast, the 2 Myr ORB simulation shows a different picture. The simulated GICs exhibit stronger amplitudes and

prolonged periodicities, and the late Pleistocene GICs demonstrate improved alignment with the Berends data. Additionally, a transition from lower to higher frequencies following the MPT is visible, although no detectable shift in amplitude occurs. While some glacial peaks in the early Pleistocene are too strong, certain late Pleistocene glacials are too weak.

(ii) **ABR model:** Introducing a simple abrupt change in the climatic forcing visually improves the reconstructed results of

the model quite a lot compared to the ORB model. The timing of the abrupt change in the model parameters $v_0(t)$ and $\tau_d(t)$ (Eq. 8) is another tuning parameter. It was tuned to a value of $t_{abr} = 1246$ ka, visualized as the sudden jump in the deglaciation parameter (black dotted line) in Fig. 2b. The deglaciation parameter $v_0(t)$ increases from 10.6 m before the jump to 96.8 m afterwards, while the relaxation time $\tau_d(t)$ increases from -113.3 kyr to 7.0 kyr. There is now a clear distinction between the early and late Pleistocene cycles. While the early ones are characterized by low periodicity and amplitude, they abruptly

increase in periodicity and amplitude, following the sudden increase in the deglaciation parameter.





Before the jump, the GICs are bounded by -8 m and 43 m with a $\bar{v}$ of 21.3 m. After the jump, the GICs are bounded by 1 m and 112 m with a $\bar{v}$ of 44.5 m. This corresponds to an increase of the glacial maxima of almost 70 m. Throughout the Pleistocene, $\bar{v}$ is 33.4 m, which is almost identical to the Berends data. Almost all terminations and interglacial endpoints are correctly identified by the model. An exception is Marine Isotope Stage 35 (MIS35), which is delayed in the ABR model.

While the timings of the GICs are well captured in the early Pleistocene, some of the interglacial minima and glacial maxima are underestimated, e.g. MIS 82, MIS 63, MIS 52 and MIS 47. The "double peak" at MIS 15, i.e., the two adjacent minima in global ice volume, is not correctly resolved, as only the first peak is captured. The same holds for MIS 13. While the glacial-interglacial amplitudes are much better resolved with the ABR compared to the ORB model, the general bias of underestimated glacial peaks and interglacial minima remains valid for the late Pleistocene. This can be seen for MIS 17, MIS 16, MIS 12,

MIS 11 and MIS 6.

The ABR model predicts two full glacial cycles over the next 250 kyr. While the first cycle is very similar to the preceding one, the latter one is interrupted by a very short interglacial, before the next glaciation continues. These future cycles are bounded by -2 m and 111 m in global ice volume and have a $\bar{v}$ of 49.7 m. The modelled start point of the Holocene is 17 ka and should have ended 4 ka, with the next glacial inception lasting until 105 kyr in the future.

The ABR model demonstrates a significant improvement in the statistics regarding the alignment between the simulated and the target data, as is evident from Table 2. The RMSE improves by more than 5 m to a value of 12.24 m and the correlation with the Berends data improves significantly with an $R^2$ value of 0.74. The strong improvement is also visible for a shorter (2 Myr) or a longer (3.6 Myr) simulation. The ABR model performs similarly well for all three simulation periods, with the best correlation for the longest simulation run.

Reducing the simulation period to 2 Myr yields similar results (Fig. B2b). The abrupt MPT is delayed to 1094 ka. The pre-MPT GICs continue to be dominated by low amplitude and low periodicity. The abrupt jump in the deglaciation threshold leads to a clear shift towards smaller frequencies and larger amplitudes. Instead of two full glacial cycles over the next 250 kyr, the ABR model now forecasts three full glacials of intermediate strength, in close agreement with the prediction by the RAMP model. On the other hand, a prolonged simulation period of 3.6 Myr leads to worse results (Fig. B3b). While the timing of the

MPT is consistently located at 1163 ka and the post-MPT cycles are similar, the pre-MPT cycles have changed significantly. Similarly to the GICs produced by the ORB model, their amplitude and period are too small. In contrast to the ORB model, the increasing trend of global ice volume is not linear, but shows a multi-millennial cyclic pattern. The future cycles are shorter in period and of smaller amplitude, such that there are four GICs forecasted over the next 250 kyr. Hence, future predictions of the ABR model are highly inconsistent and always depend on the simulation period.


(iii) **GRAD model:** In contrast to the abrupt change in internal forcing, implemented in the ABR model, the GRAD model incorporates a gradual transition throughout the Pleistocene. This change in internal forcing further improves the match between the simulated and the target data, as can be seen in Fig. 2c. The simulated global ice volume curve is in very good agreement with the Berends reconstruction. Over the course of the Pleistocene, the deglaciation parameter $v_0(t)$ increases from 20.1 m to

126.4 m and the relaxation time $\tau_d(t)$ decreases from 19.8 kyr to 5.2 kyr. During the first 1 Myr of the simulation, the GICs are





bounded by -2 m and 52 m with $\bar{v}$ of 19.5 m. For the last 1 Myr, this transitions to bounds of 1 m and 118 m with $\bar{v}$ of 49.2 m. During the simulation period, the mean global ice volume $\bar{v}$ is 33.5 m, which is in agreement with that of Berends.

As is the case for the ABR model, the GRAD model can correctly detect most terminations and interglacial endpoints. Even MIS 35 is captured, but an artificial intermediate glacial is added. Some of the weaker interglacials, such as MIS 23,

are missed, as is the case for the ABR model. Moreover, the interglacial double peak during MIS 15 is now captured, due to the cost of extending the subsequent glacial MIS 14 too much. The amplitude of pre-MPT GICs increases, leading to better reconstructions, especially visible for GICs in the interval 1.6 - 1.2 Ma. Following the MPT, certain glacial maxima continue to be underestimated. This is particularly evident in the cases of MIS 16, MIS 12, and MIS 6. Similarly, some interglacial minima are not consistent with the Berends reconstruction. This is particularly the case for MIS 31 (demonstrating improvement relative

to the ABR model) and MIS 17, while MIS 11 now aligns well with the Berends reconstruction.

The GRAD model projects two complete glacial cycles within the forthcoming 250 kyr. Both cycles exhibit a substantial amplitude of $\sim$ 110 m and have a periodicity of 100 kyr. The initial future GIC is nearly identical between the ABR and GRAD models. However, while a minor interglacial phase disrupts the second GIC in the ABR model, the GRAD model omits this interglacial period, resulting in the formation of a second pronounced glacial phase. These future cycles are bounded by -1 m

and 113 m in global ice volume and have a $\bar{v}$ of 52.2 m. The Holocene started 16 ka and ended 1 ka. The proceeding glacial cycle would last until 108 kyr in the future.

In comparison to the ABR model, the GRAD model demonstrates better agreement between the modelled and the target data, evidenced by a reduction in its RMSE by 0.38 m, resulting in a value of 11.86 m. Furthermore, its $R^2$ metric exhibits a slight increase, reaching a value of 0.76. The same holds for longer and shorter simulation periods. While the 2 Myr and 2.6

Myr simulations yield very similar results, the prolonged 3.6 Myr run correlates the closest to the Berends data.

The GRAD model yields very consistent results if run for 2 Myr (Fig. B2c) or for 3.6 Myr (Fig. B3). The MPT can be reproduced for all three simulation periods. The GRAD model consistently forecasts 2 strong glacial cycles over the next 250 kyr when run for 2 or 2.6 Myr, whereas the ice volume for the 3.6 Myr simulation diverges after 110 kyr in the future.

**(iv) RAMP model:** In comparison to the GRAD model, the RAMP model features a gradual trend in internal forcing only over a fixed period, i.e. it features a ramp-like forcing scenario. The onset of the ramp occurs at $t_2 = 2159$ ka, concluding at $t_1 = 858$ ka. Before and after the ramp, the deglaciation parameter $v_0(t)$ and the relaxation time $\tau_d(t)$ remain constant, while during the ramp, they exhibit a linear increase and decrease, respectively. While $v_0(t)$ increases from an initial value of -12.9 m to a final one of 98.4 m, $\tau_d(t)$ decreases from 31.4 kyr to 6.3 kyr. The first 1 Myr of simulation yields GICs between -1 m

and 41 m with $\bar{v}$ of 19.4 m. During the last 1 Myr of simulation, the GICs shift towards bounds of 0 m and 114 m with $\bar{v}$ of 48.5 m. The global mean ice volume over the entire Pleistocene corresponds to a $\bar{v}$ of 33.5 m, in perfect agreement with Berends, as already the case for the GRAD model. In general, the RAMP model looks very similar to the GRAD model, as can be seen in Fig. 2d. The most remarkable difference concerns future predictions.

As for the GRAD model, the timings of terminations and glacial endpoints are captured very well for the RAMP model.

Some general biases remain, which were already present in the GRAD model. For example, MIS 14 remains too long, leading





to a delayed MIS 13, the glacial maxima of MIS 12, MIS 10 and MIS 6 are underestimated, whereas certain interglacial minima are not reached, as for MIS 31, and MIS 17.

The most obvious difference between the GRAD and the RAMP model is concerning the simulated GICs for the next 250 kyr. While the GRAD model predicts two strong glacial cycles, the RAMP model anticipates three intermediate glaciations.

They are characterized by glacial periods between 60 and 80 kyr and are delimited by 6 m and 83 m with $\bar{v}$ of 43.4 m. In the RAMP model, the Holocene lasted from 16 ka until 1 ka. The proceeding glaciation would continue till 65 kyr in the future.

Table 2 indicates that the RAMP model shows the best agreement with the Berends data out of the four different model configurations. It lowers the RMSE compared to the GRAD model by nearly half a meter, resulting in a value of 11.38 m. Its correlation with the Berends data improves to the highest value of 0.78. The situation is the same for the 2 Myr and the 3.6

Myr runs, where the RAMP model consistently yields the best results among all models.

The RAMP model produces very consistent results when run for a shorter (2 Myr, Fig. B2d) or a longer (3.6 Myr, Fig. B3d) period. While the onset of the ramp is always tuned to an early stage of the simulation ($t_2 = 1,884$ ka for 2 Myr simulation; $t_2 = 2,159$ ka for 2.6 Myr simulation; $t_2 = 3,479$ ka for 3.6 Myr simulation), the end is always located between 600 and 900 ka ($t_1 = 664$ ka for 2 Myr simulation; $t_1 = 858$ ka for 2.6 Myr simulation; $t_1 = 840$ ka for 3.6 Myr simulation). Compared to

the other models, the RAMP model uniquely provides consistent projections of future glacial cycles, predicting the occurrence of three intermediate glacial cycles over the next 250 kyr.

To investigate how a different tuning target affects the simulation outcome, the four models are tuned to the Rohling sea level curve (Rohling et al., 2022) instead of the Berends data (Fig B4, Fig B5). The different tuning target affects the outcome of the

ORB model the most. When tuned to the Rohling data, the ORB model simulates GICs with larger amplitudes and extended periods. It is in closer agreement with the results obtained for the 2 Myr run, compared to the 2.6 and 3.6 Myr runs. However, the ORB model is still incapable of reconstructing the MPT, since there is no change in the amplitude and frequency of the GICs throughout the simulation.

For the ABR model, the different tuning target affects the simulation outcome only slightly. For the Rohling-based solution,

the double peak during MIS 15 is reconstructed and the model now predicts three intermediate strong GICs for the upcoming 250 kyr, in very close agreement with the RAMP predictions. The timing of $t_{abr}$ is tuned to 1098 ka, a slightly younger age compared to the Berends-based simulation.

As visible from supplementary Fig. B5c, the simulated global ice volume over the past 2.6 Ma for the GRAD model is barely affected by the chosen tuning target. The main difference is that the post-MPT interglacial minima are consistently

lower for the Rohling-based simulation. For the next 250 kyr, the forecasted timings and amplitudes of the GICs are in very close agreement.

Similarly to the GRAD model, the RAMP model demonstrates minimal impact when the tuning target is switched from the Berends data to the Rohling data (Fig. B5d). A notable difference lies in the earlier onset of the ramp ($t_2 = 2286$ ka), while the end stays similar ($t_1 = 881$ ka). However, the different timings do not appear to influence the simulated GICs, nor the





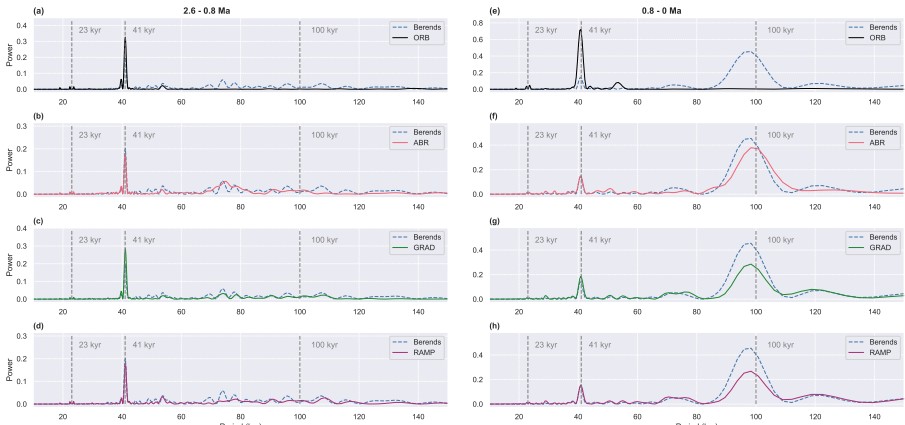

**Figure 3.** Lomb-Scargle Periodogram of the four different models (coloured solid lines) and of the Berends sea-level reconstruction (blue dashed lines). Panel **(a)** - **(d)** display the periodogram during the early- and mid-Pleistocene (2.6 - 0.8 Ma), while panel **(e)** - **(h)** show the late-Pleistocene (0.8 Ma to present). Minimum frequency: $f_{min} = 150^{-1}$ kyr$^{-1}$, maximum frequency: $f_{max} = 10^{-1}$ kyr$^{-1}$.

projected future cycles. Therefore, the RAMP model stands out as the sole model providing consistent outcomes across all three evaluated simulation periods and for a different tuning target.

### 3.3   Spectral analysis of the different models

Potential frequency shifts due to the MPT can be seen in the periodograms in Fig. 3. The Berends data exhibits a dominant 41 kyr peak when analyzed on a timeframe of 2.6 - 0.8 Ma, which transitions to a combination of a minor 41 kyr and a dominant 420   100 kyr peak for the last 800 ka. The ORB model does not feature such a transition in periodicity with a 41 kyr period being dominant in both analysis timeframes. In contrast, the other three models exhibit a distinctly different pattern. All three models demonstrate a clear change in periodicity, transitioning from a dominant 41 kyr cyclicity in the early- to mid-Pleistocene towards a combination of a minor 41 kyr and a dominant roughly 100 kyr period. This is in good agreement with the Berends reconstruction. The peak around 100 kyr for the last 800 ka in the ABR model is slightly larger compared to the GRAD and 425   RAMP models, which agrees better with the Berends reconstruction.

In summary, the ABR, GRAD and RAMP models are capable of reproducing a shift in periodicity due to the MPT in good agreement with the results obtained from the Berends data. On the other hand, the ORB model is the only model configuration that is incapable of reproducing a change in frequency.

### 3.4   Model performance

To further quantify the performance of each model and to identify the importance of adding certain parameters to the model, we focus on the $\Delta$BIC values. Table 3 provides a complete summary of all $\Delta$BIC values between the four models for the three different simulation periods. It can be seen that the ORB model performs the worst for all three simulation periods, despite



**Table 3.** ΔBIC values between all four models for the three simulation periods used.

| Model | # Parameters | 2 Myr simulation | | | 2.6 Myr simulation | | | 3.6 Myr simulation | | |
|---|---|---|---|---|---|---|---|---|---|---|
| | | ΔBIC vs ORB | ΔBIC vs ABR | ΔBIC vs GRAD | ΔBIC vs ORB | ΔBIC vs ABR | ΔBIC vs GRAD | ΔBIC vs ORB | ΔBIC vs ABR | ΔBIC vs GRAD |
| Orbital (ORB) | 12 | 0 | / | / | 0 | / | / | 0 | / | / |
| Abrupt (ABR) | 15 | 362.5 | 0 | / | 679.1 | 0 | / | 476.8 | 0 | / |
| Gradual (GRAD) | 14 | 399.4 | 36.9 | 0 | 728.1 | 49.0 | 0 | 620.4 | 143.6 | 0 |
| Ramp-like (RAMP) | 16 | 393.6 | 31.1 | -5.8 | 761.5 | 82.7 | 33.7 | 659.1 | 182.3 | 38.7 |

having the least number of parameters (12). The ABR model contains three additional parameters (15) which significantly improve the performance of the model. It exceeds a ΔBIC of more than 300 for all three simulation periods compared to the
ORB model, indicating very strong evidence in favour of the ABR model. In addition to smaller RMSEs (see Table 2), the GRAD model has one parameter less (14) than the ABR model. As a result, the GRAD model outperforms the ABR model, as indicated by ΔBIC values of 36.9, 49.0 and 143.6 for the three different simulation runs. Despite having two additional parameters compared to the GRAD model, the RAMP model (16) shows better performance over simulation periods of 2.6 Myr and 3.6 Myr. Only for a reduced simulation period of 2 Myr the ΔBIC value becomes negative, indicating positive
evidence in favour of the GRAD model. This result is reflected in the small difference between their RMSEs (see Table 2) for the 2 Myr simulation, while the difference in RMSEs becomes larger for longer simulation periods. For the 2.6 Myr and 3.6 Myr simulations, its ΔBIC calculated with respect to the GRAD model is greater than 30, indicating very strong evidence in favour of the RAMP model.

In summary, the analysis of the Bayesian information criterion underlines the finding that the ORB model performs signifi-
cantly worse than the other three models. While the GRAD model is superior to the ABR model, the RAMP model outperforms all other models, except for a reduced simulation period of 2 Myr, when the evidence for the GRAD model is positive.

### 3.5 The MPT in the RAMP model

The RAMP model produces an almost abrupt change in the amplitude and frequency of glacial cycles after around 900 ka, despite having a gradual change in internal forcing. To understand how a gradual change in forcing can produce a sudden
change in global ice volume dynamics, it is important to investigate how the mechanisms for triggering deglaciations and glaciations have changed during this period. This transition for the period of 1.2 - 0.6 Ma is shown in Fig. 4.

As outlined in the Methods section, two conditions must be fulfilled to trigger a deglaciation (Eq. 6) and two conditions for the onset of a new glaciation (Eq. 7). A termination is triggered once the ice volume exceeds a critical volume (black line) and the current insolation is sufficiently large (red-shaded areas). Once both conditions hold (purple dots), a state change from
a glaciation to a deglaciation mode is performed and the dynamics of the system follow Eq. (5). This is the case until two criteria are fulfilled: Firstly, the ice volume falls again below a critical volume (black line; note the inverted y-axis in Fig. 4), and secondly, the insolation is sufficiently small (blue-shaded area). This allows the model to enter the next glacial state (red dots), and its evolution follows Eq. (4).



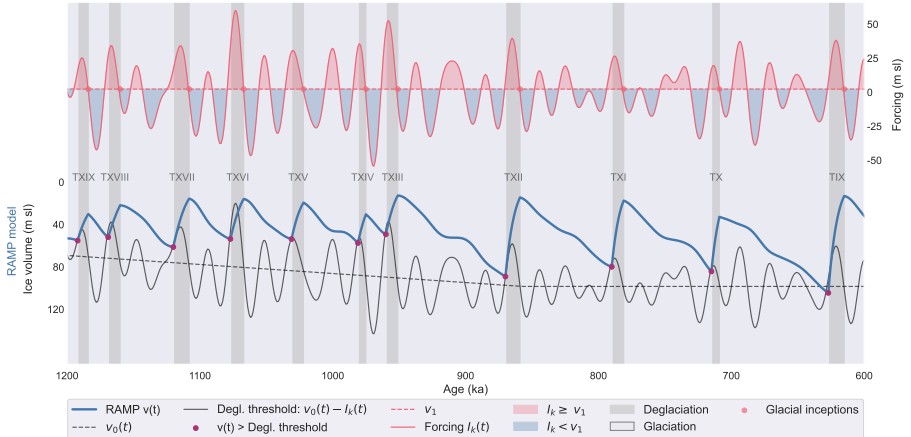

**Figure 4.** Simulated ice volume by the RAMP model (blue line) for the period 1.2 - 0.6 Ma, together with its deglaciation and glaciation thresholds. The forcing curve $I_k(t)$ is shown in red. While the red shaded area marks values above the model threshold $v_1$, the blue shaded area marks values below. The time-dependent deglaciation parameter $v_0(t)$ is represented by the black dotted line, while the solid black line indicates the deglaciation threshold. The purple dots indicate terminations, and the red dots mark glacial inceptions. Modelled terminations are labelled with Roman numerals.

As apparent in Fig. 4, there is an abrupt change in amplitude and frequency of the glacial cycles after glacial termination
TXIII. Before this termination, each or every second insolation peak resulted in a deglaciation. After TXIII, more insolation peaks were missed, leading to extended glacial periods and amplitudes. Termination TXIII was triggered by a strong insolation peak and led to a very short preceding glacial period. Due to the short glacial period, only a small global ice volume was accumulated. Subsequently, the interglacial led to the lowest global ice volume (12.6 m) in the last 400 ka. In the RAMP model, the deglaciation threshold (black line) increases over time, due to the increase of the model parameter $v_0(t)$ (black
dotted line). Therefore, the required critical ice volume to trigger the next termination increases as well. The combination of the low global ice volume at the onset of the next glacial period after TXIII and the increasing deglaciation threshold in the RAMP model enabled the model to skip three insolation peaks to yield the first glacial maximum exceeding 80 m in global ice volume and led to the longest glacial period in the Pleistocene so far. After TXIII, the extended periodicities and amplitudes prevailed.

## 3.6   Performance of the RAMP-l model

The RAMP-l model includes three additional parameters which make the forcing term $\tilde{I}_k(t)$ ice volume dependent. This significantly improves the simulation outcomes (Fig. 5a). The RMSE, compared to the standard RAMP model, is improved by 1.36 m to a new optimal value of 10.02 m. Likewise, $R^2$ increases to a new best value of 0.83. The improved performance comes with the cost of adding three new parameters to the model. Despite these extra parameters, the calculation of the $\Delta$BIC



between the RAMP-l and RAMP models demonstrates with a value of $\Delta\text{BIC}_{\text{RAMP-l,RAMP}} = 107.3$ that there is very strong evidence in favour of the RAMP-l model over the RAMP model.

The simulated global ice volume by the RAMP-l model can be seen in Fig. 5a. While the early onset of the ramp stays similar for the RAMP-l model ($t_2 = 1978$ ka), the endpoint is shifted by over 400 kyr ($t_1 = 454$ ka) compared to the RAMP model. Therefore, the total duration of the ramp is extended. The deglaciation parameter $v_0(t)$ increases from an initial value

of 20.7 m to a final value of 127.4 m. The relaxation parameter $\tau_d(t)$ decreases from 41.2 kyr to 5.3 kyr during the Pleistocene. During the first 1 Myr, the simulated global ice volume is confined to values between -4 m and 48 m with a mean ice volume of 19.8m. The last 1 Myr are characterized by an ice volume between -1 m and 111 m and a mean ice volume of 47.9 m. The global mean ice volume over the entire Pleistocene corresponds to a $\bar{v}$ of 33.3 m, in very close agreement with the Berends data.

Compared to the standard RAMP model, the RAMP-l model simulates interglacials with smaller ice volumes, and glacial with larger ones. Hence, the interglacial-glacial variability is increased. This is in better agreement with the Berends data, especially visible for the GICs in the interval 1.6 - 1.2 Ma. Another major difference is that the RAMP-l model no longer resolves the two interglacial double peaks (MIS15, MIS7), but in turn, the succeeding glacial (MIS14) is better resolved.

A further difference to the standard RAMP model concerns the simulated GICs for the next 250 kyr. While the RAMP model

projects three intermediate strong glacials, the RAMP-l model projects, similar to the GRAD model, the occurrence of two complete glacial cycles within the next 250 kyr. Both cycles exhibit major glaciations of ~110 m and have glacial periods of around 100 kyr. The Holocene started 15 ka and would continue for the next 3 kyr. The following glacial termination would be 107 kyr in the future.

Figure 5 shows that the periodograms for the simulated ice volume by the RAMP-l model and the ice volume reconstruction

by Berends et al. (2021a) are in close agreement. Although the early to mid-Pleistocene GICs (2.6-0.8 Ma) are mainly driven by a ~ 41 kyr periodicity (Fig. 5b), the late Pleistocene (last 0.8 Ma) GICs show a combination of ~ 41 kyr and 100 kyr periods with a dominant peak around ~ 100 kyr (Fig. 5c).

### 3.7 Forcing in the RAMP-l model

In the model, the external driver of the temporal evolution of the ice volume is the orbital forcing $I_\alpha$. It is composed of a

linear combination of three orbital parameters, namely: obliquity, precession, and phase-shifted precession. The three tuning parameters $\alpha_{\text{Esi}}$, $\alpha_{\text{Eco}}$ and $\alpha_{\text{O}}$ determine how this linear combination adds up and Fig. 6a shows the resulting orbital forcing in the RAMP-l model. To identify the insolation curve corresponding to the obtained $I_\alpha$ curve, we compare it to daily insolation curves at specific latitudes, monthly averaged ones, and various integrated summer insolations (ISI) reported in the literature. The closest correlation is found with the ISI above $285 \text{ W m}^{-2}$ at $67°$ N (calculations done by code provided in Leloup and

Paillard (2022)). Both curves align almost perfectly (Fig. 6a) with an RMSE of 0.24 m and an $R^2$ value of 0.94. The spectral analysis (Fig. 6b) of these insolation curves demonstrates that they consist of minor precession peaks around 19 kyr, 22.4 kyr and 23.7 kyr and a dominant obliquity peak around 41 kyr.



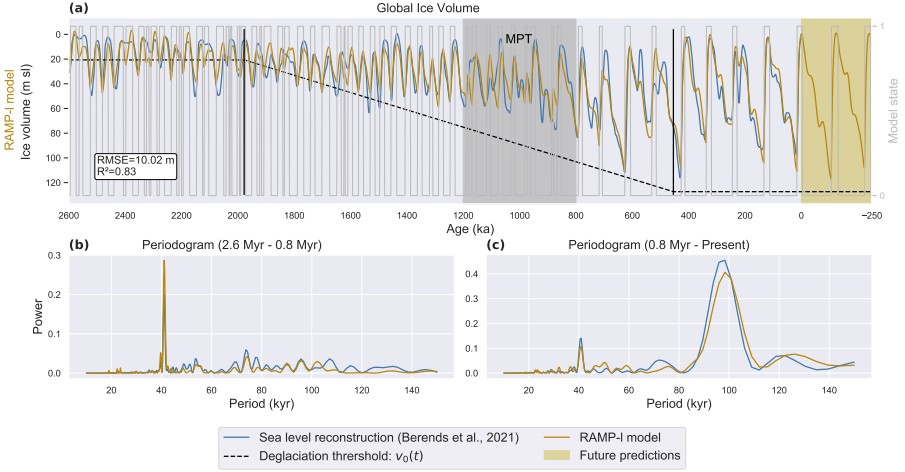

**Figure 5. (a)** Simulated global ice volume by the RAMP-l model (golden line) over the whole Pleistocene, together with the ice volume reconstruction by Berends et al. (2021a) (blue line). The black dotted line shows the deglaciation parameter $v_0(t)$. The yellow shaded area marks the future 250 kyr and the respective model predictions. The grey-shaded area highlights the classical perspective of where the MPT is located in time. The periodograms of both ice volume curves are shown for the time interval 2.6 Ma - 0.8 Ma **(b)** and for the time interval 0.8 Ma - present **(c)**. Minimum frequency: $f_{min} = 150^{-1}$ kyr$^{-1}$, maximum frequency: $f_{max} = 10^{-1}$ kyr$^{-1}$.

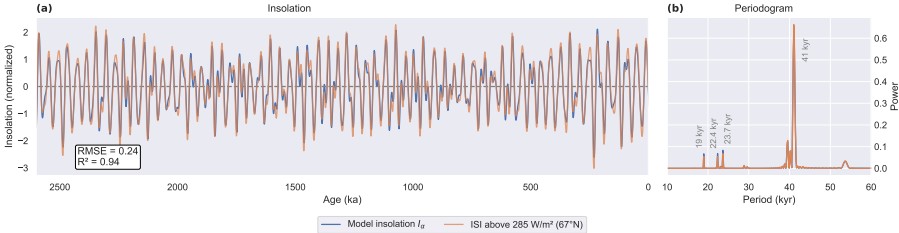

**Figure 6. (a)** Orbital forcing $I_\alpha$ (Eq. 4, 5) in the RAMP-l model (orange line) together with the best fitting insolation curve, the ISI above 285 W m$^{-2}$ at 67° N (blue line, calculated by code from Leloup and Paillard (2022)). **(b)** Periodogram of both insolation curves.

The second forcing term in the RAMP-l model does not drive the time evolution of the ice volume directly, as the $I_\alpha$ term does. Instead, it controls the associated thresholds (Eq. 6, 7) to initiate state changes in the model. This forcing is the $\tilde{I}_k(t)$ term.
Due to its ice volume dependency, its amplitude increases over time (Fig. 7a). Hence, with the progression of the Quaternary period, an increase in global ice volume corresponds with a rise in the $\tilde{I}_k(t)$ term. While in the early Quaternary, the forcing is confined to values between $\pm$ 25 m, it increases towards values exceeding $\pm$ 50 m in the late Quaternary. The first threshold condition requires $\tilde{I}_k(t)$ to exceed or fall below the model threshold $v_1$ (red-shaded or blue-shaded areas, respectively, in Fig. 7a). The forcing term also affects the deglaciation threshold $(v_0(t) - \tilde{I}_k(t))$. A spectral analysis of the forcing term shows
that the dominant frequency has shifted during the Quaternary. The $\tilde{I}_k(t)$ term is always composed of precession peaks at 19 kyr, 22.4 kyr and 23.7 kyr and an obliquity peak around 41 kyr. However, while in the early Quaternary (2.6 Ma - 1.2 Ma), the





dominant peak was the obliquity one, it shifted in the late Quaternary (0.8 Ma - present) towards dominant precession peaks at 22.4 kyr and 23.7 kyr (Fig. 7b,c).

Given that $\tilde{I}_k(t)$ undergoes temporal changes, it is interesting to investigate how this impacts the state changes in the model. For this reason, we identify each time step $t_i$ in the simulation in which a state change occurs. These time steps are visualized in Fig. 7d, colour-coded with the current model state. While an interglacial state is followed by a glacial inception in the next simulation step, a glacial state is followed by a glacial termination. At least one of the two necessary conditions (for termination: Eq. 6, for gl. inception: Eq. 7) was not fulfilled in the previous time step $t_{i-1}$. Hence, we classify the time steps $t_i$ according to the decisive condition that finalized the state change, i.e. the condition related to $v_1$ (for termination: $\tilde{I}_k(t) \geq v_1$, for gl.

inception: $\tilde{I}_k(t) < v_1$), related to $v_0(t)$ (for termination: $v(t) + \tilde{I}_k(t) > v_0(t)$, for gl. inception: $v(t) + \tilde{I}_k(t) \leq v_0(t)$), or both conditions are fulfilled simultaneously. The pattern in Fig. 7d shows a regime change around 1.7 Ma in the model regarding the decisive condition for triggering a state change. Before 1.7 Ma, glacial terminations and inceptions are mainly timed by the $v_1$ condition, i.e. when the $\tilde{I}_k$ term falls below or exceeds the $v_1$ threshold. In contrast, after 1.7 Ma, each interglacial state is terminated by the next glacial inception due to the $v_1$ condition and each glacial state ends in a termination due to the $v_0$

condition (three exceptions where $v_0$ and $v_1$ conditions are simultaneously fulfilled). Hence, the glacial terminations are timed by crossing some critical ice volume.

### 3.8 Sensitivity of large ice sheets to precession in the RAMP-l model

During the tuning process, the following values were found for the l-parameters:

$$l_{\mathrm{Esi}} = 0.68, \qquad l_{\mathrm{Eco}} = 0.14, \qquad l_{\mathrm{O}} = 0.14. \tag{15}$$

Hence, for large ice volumes, the precession term $l_{\mathrm{Esi}} v(t) \mathrm{Esi}(t)$ is dominant. This can also explain the apparent shift towards dominant precession peaks (Fig. 7b,c) in the late Quaternary when the amplitude of glacials increased.

To investigate the sensitivity of large ice sheets to precession and obliquity, it is convenient to separate the forcing term $\tilde{I}_k(t)$ (Eq. 11) in two terms, one including all precession and co-precession terms and the other containing all obliquity terms:

$$\tilde{I}_{k,Pr}(t) = [k_{\mathrm{Esi}} + l_{\mathrm{Esi}} v(t)] \, \mathrm{Esi}(t) + [k_{\mathrm{Eco}} + l_{\mathrm{Eco}} v(t)] \, \mathrm{Eco}(t) \tag{16}$$

$$\tilde{I}_{k,Ob}(t) = [k_{\mathrm{O}} + l_{\mathrm{O}} v(t)] \, \mathrm{Ob}(t) \tag{17}$$

$$\tilde{I}_k(t) = \tilde{I}_{k,Pr}(t) + \tilde{I}_{k,Ob}(t) \tag{18}$$

For the last 900 ka, the larger l parameter for precession leads to larger amplitudes in the precession term $\tilde{I}_{k,Pr}(t)$ compared to the obliquity term $\tilde{I}_{k,Ob}(t)$ (Fig. 8a).

To investigate the sensitivity of large ice sheets to orbital forcing, we focus on the triggering of glacial terminations during

the past 900 ka in the RAMP-l model (Fig. 8b). Glacial terminations are triggered once $\tilde{I}_k(t)$ exceeds $v_1$ and the sum of the current forcing and ice volume $v(t) + \tilde{I}_k(t)$ exceeds $v_0(t)$. As previously shown in Fig. 7d, the decisive condition for glacial terminations for the last 1.7 Ma is the $v_1$ condition. This means that the timing of these glacial terminations is determined by the timing when the critical ice volume threshold is crossed. For the last 900 ka, almost all terminations are associated around



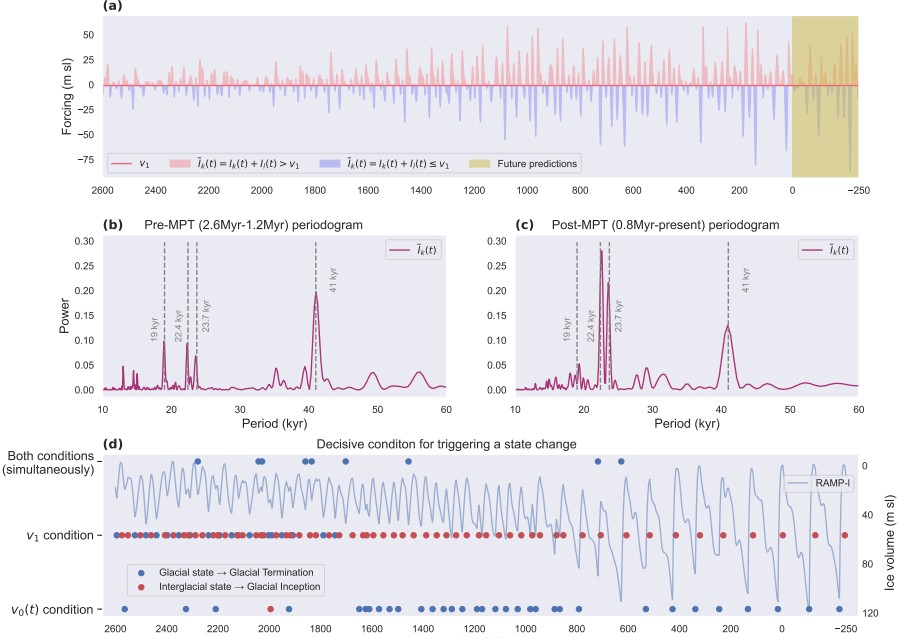

**Figure 7. (a)** Forcing $\tilde{I}_k(t)$ in the RAMP-l model (Eq. 11) and model threshold $v_1$ (red line). One condition to trigger a termination is that $\tilde{I}_k(t)$ needs to exceed the threshold $v_1$ (red-shaded area, Eq. 6). Conversely, a necessary condition for glacial inception is that $\tilde{I}_k(t)$ is below $v_1$ (blue-shaded area, Eq. 7). The yellow shaded area marks the future 250 kyr. **(b)** The periodogram of the forcing term $\tilde{I}_k(t)$ for the interval 2.6 Ma - 1.2 Ma. **(c)** The periodogram of the forcing term $\tilde{I}_k(t)$ for the interval 0.8 Ma - present. **(d)** Decisive condition for triggering a state change. The coloured dots mark time steps when a state change occurs in the model. Blue dots indicate that the model is currently in a glacial state and will experience a glacial termination in the next time step. Conversely, red dots indicate that the model is currently in an interglacial state and will experience a glacial inception in the next time step. To change from a glacial to an interglacial state, or vice versa, two conditions, including the model parameters $v_0(t)$ and $v_1$, must be fulfilled (Eq. 6,7). The y-axis marks which of these two conditions was the decisive one (last one that was fulfilled) to trigger a state change, or in rare occasions when both conditions were fulfilled at the same time step.

peak $\tilde{I}_k$ forcing conditions (purple dots in Fig 8b). An exception is MIS15, when a strong minimum in forcing allowed to build
up such a large ice volume that the next moderately positive forcing values were sufficient to trigger a termination. During the last 900 ka, 11 terminations occurred. Figure 8a demonstrates that for 10 out of those 11 terminations, the precession term of the forcing was larger than the obliquity one. Hence, the precession term $\tilde{I}_{k,Pr}$ played the key role in achieving the required forcing to end glacial periods during the past 900 ka.





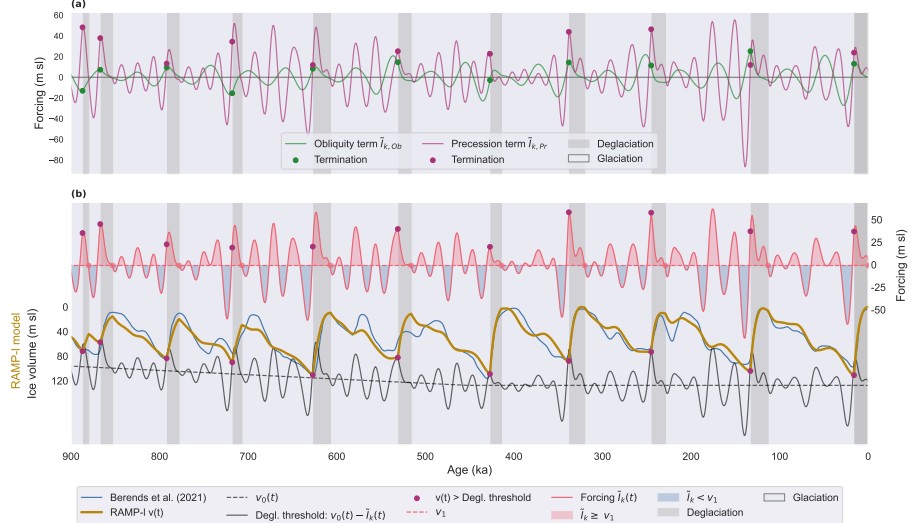

**Figure 8. (a)** The two components of the forcing term $\tilde{I}_k(t)$: precession and co-precession components $\tilde{I}_{k,Pr}(t)$ (purple line, Eq. 16) and the obliquity components $\tilde{I}_{k,Ob}(t)$ (green line, Eq. 17). Their respective values at glacial terminations are marked with coloured dots. **(b)** Simulated ice volume by the RAMP-l model (golden line) for the past 900 ka, together with its deglaciation and glaciation thresholds. The forcing $\tilde{I}_k(t)$ is shown in red. While the red shaded area marks forcing values above the model threshold $v_1$, the blue shaded area marks values below. The time-dependent deglaciation parameter $v_0(t)$ is represented by the black dotted line, while the solid black line indicates the deglaciation threshold. The purple dots indicate terminations, and the red dots mark glacial inceptions.

## 4  Discussion

### 4.1  Using only orbital forcing

The four standard models (ORB, ABR, GRAD, RAMP) are conceptualisations of four distinct forcing scenarios. The ORB simulation represents a climate system influenced solely by orbital forcing, excluding any internal feedback mechanisms. Our results showed that this approach fails in reconstructing the MPT along with its associated change in amplitude and frequency. This could already be observed in the ORB model of Legrain et al. (2023), which simulates glacial amplitudes before the MPT that are consistently too high, while those after the MPT are too low. Moreover, a clear transition from 41 kyr to 100 kyr cycles is not observed. In contrast, a pronounced 200 kyr signal during the period of 1-2 Ma is visible, and the 41 kyr signal intensifies in the late Pleistocene, almost matching the power of the 100 kyr signal.

We cannot exclude the possibility that different orbital background conditions or a different parameterization could recreate the MPT. However, our ORB model supports the idea that orbital changes alone cannot account for this major climatic transition. This suggests the presence of non-linearities within the climate system that modify Earth's response to changes in the orbital forcing. This conclusion aligns with findings from other studies (Clark et al., 2006; Leloup and Paillard, 2022; Berends et al., 2021b).





## 4.2 Abrupt versus gradual scenario

If a purely external driver for the MPT can be ruled out, it remains open whether the MPT was triggered by an abrupt or a
gradual scenario. Although improving the model performance did not alter the general finding of Legrain et al. (2023) that
the model is in favor of a gradual transition, the ABR model does reproduce the MPT, and the timing of its modelled abrupt
change (1.25 Ma) agrees with some proxy reconstructions. While some climatic marine records indicate an abrupt transition
approximately 0.9 Ma (Elderfield et al., 2012; Yehudai et al., 2021), the modelled jump in the ABR model is more consistent
with the proxy records from Site U1537, which show an abrupt change in dust deposition around 1.25 Ma (Weber et al.,
2022). It also corresponds to the onset of gradual cooling trends evident in multiple sea-surface temperature (SST) records
(McClymont et al., 2013). Moreover, it aligns with a study by Nyman and Ditlevsen (2019) that claims that the MPT did not
lead to a stationary 100 kyr world, but rather that there was an abrupt transition from $\sim$41 kyr to $\sim$ 80 kyr cycles about 1.2 Ma,
followed by a gradual increase towards 120 kyr thereafter.

Although the ABR model can reproduce the MPT, the obtained results with the GRAD model are better. The GRAD model
shows an increasing secular trend in the deglaciation parameter over the entire Quaternary. While the superior performance
of the GRAD model is in favour of a long-lasting gradual trend in the climate system, its underlying physical mechanism
cannot be directly inferred but just hypothesized since the conceptual modelling approach lacks the physical representation
of these mechanisms. Whether a gradual erosion of regolith (Clark and Pollard, 1998; Clark et al., 2006), a gradual decrease
in atmospheric $CO_2$ (Berends et al., 2021b; Scherrenberg et al., 2024), some sea-ice feedback mechanism, a mixture of these
(Willeit et al., 2019) or something else, is the underlying mechanism behind this trend, requires a more physics-based model
which can explicitly resolve these mechanisms.

The hypothesis of a long-term decrease of atmospheric $CO_2$ concentrations as the cause of the MPT remains difficult to
verify since the continuous direct $CO_2$ records from Antarctic ice cores are currently limited to the past $\sim$ 800 ka (Willeit et al.,
2019; Bereiter et al., 2015) and beyond that, the discontinuous ice core samples from the Allan Hills Blue Ice area provide
only three snapshots of direct $CO_2$ estimates attached to a large uncertainty (Yan et al., 2019). $CO_2$ estimates over the pre-MPT
and the MPT worlds can be obtained using indirect proxies inferred from terrestrial and marine archives. For instance, $CO_2$
reconstructions are based on $\delta_{13}C$ carbonates from leaf-wax (Yamamoto et al., 2022), from paleosols (Da et al., 2019), or $\delta^{11}B$
boron based reconstructions (Chalk et al., 2017; Hönisch et al., 2009). Based on the latter, there seems to be a decrease in
minimal $CO_2$ concentrations during glacial maxima across the MPT. However, looking into a recent synthesis compiling all
proxy-based paleo $CO_2$ data (CenCO$_2$PIP Consortium, 2023), alternative $CO_2$ reconstructions suggest different trends. Hence,
there is no clear answer yet regarding the long-term evolution of the atmospheric $CO_2$ concentration across the Pleistocene.

## 4.3 Ramp-like internal forcing

On top of the ABR and GRAD models, we implemented another forcing scenario, the RAMP model, characterized by a time-
limited linear trend between two constant states. Unlike the ABR and GRAD models, it is more flexible since it does not
determine the internal forcing a priori. The tuning process aims to find the most likely parameters to obtain the best agreement





with the target data. If the start and endpoints of the ramp are the same, or very close, the RAMP model simulates an abrupt change in the internal system, similar to the ABR model. Conversely, if these points are positioned at the beginning and end of the simulation period, the RAMP model simulates a linear trend in the climate system across the entire Quaternary, similar to the GRAD model. Thus, tuning the RAMP model facilitates the evaluation of the relative significance of these two scenarios

without prior assumptions. Moreover, it allows identifying a start point, an endpoint and the duration of a possible linear trend in the climate system. By tuning the RAMP model, we found that a long-lasting linear trend of increasing deglaciation parameter $v_0(t)$ and decreasing relaxation time $\tau_d(t)$ yields the best results. For the two simulations run for Quaternary climate (2 Myr and 2.6 Myr), we found that the onset of the ramp occurs early, around 2 Ma and the end around 600 - 900 ka. The tuned RAMP model demonstrated greater relevance than the GRAD and ABR models (for 2.6 Myr and 3.6 Myr simulations),

suggesting that a gradual trend within the Quaternary, that lasted over a limited period, is key to reproducing the MPT.

As Verbitsky and Crucifix (2023) point out, while some phenomenological models can very accurately recreate observational time series (i.e. paleo records), it does not necessarily reflect their physical similarity to Nature. Hence, it is crucial to investigate whether such a ramp-like forcing in our RAMP model can be justified by paleo records or modelling studies. A review study by McClymont et al. (2013) investigated the imprint of the MPT on the surface ocean through the analysis of various SST records.

They found that most sites do exhibit a pronounced cooling trend over the period 1.2 - 0.8 Ma, but regional differences exist, especially regarding the magnitude of the cooling and the onset of the cooling trend, starting as early as 1.8 Ma. An analysis of Atlantic and Pacific benthic $\delta_{13}C$ as proxies for obliquity and precession responses in Atlantic overturning revealed a major transition which occurred 1.6 - 1.5 Ma (Lisiecki, 2014).

A statistical model by Tzedakis et al. (2017) relates glacial terminations with a required energy threshold and shows that

this deglaciation threshold had to rise over the Pleistocene. They found that the increase is ramp-like with a linear trend lasting from 1.55 Ma until 0.61 Ma. The conceptual model by Ganopolski (2024) can also recreate the MPT by changing its critical ice volume parameter ramp-like rather than linear. However, it uses a smoother transition function, namely a hyperbolic tangent centered around 1050 ka with a transition time of 250 kyr. The asymptotic behavior of the hyperbolic tangent leads to quasi-constant parameter values prior to 1.55 Ma (twice the transition time) and after 0.55 Ma. The author stresses the similarity

with the modelled regolith mask in Willeit et al. (2019), connecting the ramp-like structure with the erosion of the Northern Hemisphere regolith. The optimal volcanic $CO_2$ outgasing scenario (V2) found in Willeit et al. (2019) resembles a ramp-like structure for the past 2.6 Ma with a decreasing trend of volcanic $CO_2$ outgasing between around 2.1 Ma and 0.9 Ma.

In summary, the various proxy records differ remarkably in their temporal structure and range from abrupt climatic transitions and prolonged gradual trends to time-limited gradual transitions. Certain proxy records, when considered alongside the

modelling studies presented above, support the concept of a ramp-like internal forcing. The modelled endpoint in the RAMP model (0.6-0.9 Ma) aligns with findings from SST records (∼0.8 Ma, McClymont et al. (2013)), abrupt climatic transitions visible in certain records (∼ 0.9 Ma, Elderfield et al. (2012); Yehudai et al. (2021)), a statistical model of energy thresholds (∼0.6 Ma, Tzedakis et al. (2017)) and with a modelled volcanic $CO_2$ outgasing scenario (∼ 0.9 Ma, Willeit et al. (2019)). In contrast, the early onset of the gradual trend in the RAMP model around 2 Ma is less evident in the proxy records, as only later

starting points are visible. The earliest onset of a long-lasting cooling trend in SSTs was identified around 1.8 Ma (McClymont





et al., 2013), while a major transition in the Atlantic overturning was detected around 1.6 - 1.5 Ma (Lisiecki, 2014). Moreover, the ramp-like energy threshold in Tzedakis et al. (2017) suggests a later start around 1.55 Ma. The volcanic $CO_2$ outgasing scenario in Willeit et al. (2019) closely resembles the ramp-like structure in our RAMP model with an early onset around 2.1 Ma. However, atmospheric $CO_2$ concentrations as a potential source for our ramp-like forcing cannot be tested considering the large uncertainty of proxy-based reconstructions across the Pleistocene (as already discussed in Sect. 4.2). This knowledge gap is anticipated to be addressed in the future through ongoing efforts to retrieve a continuous ice core record extending beyond one million years, as pursued, for instance, by the European Beyond EPICA-Oldest Ice Core project and the Australian Million Year Ice Core project.

## 4.4 Sensitivity of large ice sheets

In the second part of this study, we selected the best-performing model (i.e. the RAMP model) and adjusted it to test if an ice volume dependency in the threshold's forcing term could further improve this new model (RAMP-l). This allows testing the sensitivity of large ice sheets to changes in orbital parameters. This model adjustment significantly improved the performance, compared to the four standard models, demonstrating that the thresholds (introduced in Eq. 6, 7) used in the base models might be too simplistic to model the entire Quaternary accurately and that it requires additional adjustments like an ice volume dependency to account for changes in the underlying climate system over the course of the Quaternary.

Recent studies suggest that the duration and timing of deglaciation and glaciation events over the last 900 ka were largely deterministic and driven by the relative phasing of precession, obliquity, and eccentricity (Barker et al., 2022, 2025). Barker et al. identified candidate precession peaks, which are precession peaks that begin while obliquity is increasing, as essential for glacial terminations. They found that glacial terminations during the last 900 ka correspond to the first candidate precession peak after a minimum in eccentricity. Meanwhile, obliquity seems to play a more important role in maintaining peak interglacial conditions and for glacial inceptions. While obliquity seems to play a more important role in the direct evolutionary equations in the RAMP-l model, the picture is different for triggering glacial terminations. During the last 900 ka, 11 terminations occurred in the RAMP-l model, out of which 10 were dominated by the precession term. Hence, our RAMP-l model supports the idea that precession plays a more important role for glacial terminations of the last 900 ka, as proposed by Barker et al. (2025).

Presumably, while precession seems to be more important for glacial terminations, obliquity appears to be more important for the time evolution. The evolution of the ice volume in the RAMP-l model is driven by a linear combination of three orbital parameters: precession, co-precession and obliquity. Therefore, the orbital forcing in the model depends on the tuned parameter values. In comparison to other conceptual models which rely on a specific insolation metric for the orbital forcing (e.g. Ganopolski (2024); Paillard (1998); Leloup and Paillard (2022)), this allows for more flexibility in the model and reduces subjective modelling choices. This approach has been successfully used in other models (Imbrie et al., 2011; Legrain et al., 2023). Choosing a specific insolation metric does affect the quality of the reconstruction result due to variations in their spectral profiles (Leloup and Paillard, 2022). For instance, the insolation at the summer solstice at 65° N has a stronger precession signal than the one obtained from the caloric season insolation. Hence, selecting an insolation curve to force the modelled ice volume



can pose a modelling bias. Furthermore, since using a linear combination of orbital parameters can reproduce most insolation curves at different latitudes and seasons (Imbrie et al., 2011; Loutre, 1993), the tuned parameters will construct an optimal curve for the orbital forcing. This optimal forcing term can then be compared to real insolation curves. For the RAMP-l model, we found that the tuned parameters result in a curve which is in almost perfect agreement ($R^2 = 0.94$) with the ISI above 285 W m$^{-2}$ at 67° N. This curve has a very strong obliquity signal, indicating that for the evolutionary term $\frac{\mathrm{d}v}{\mathrm{d}t}$, the obliquity signal is more important than the precession one.

### 4.5 Future predictions

Another feature of the conceptual models presented in this study is their ability to extrapolate future glacial-interglacial cycles. For the next 250 kyr, the GRAD and RAMP-l models predict two full glacial cycles with amplitudes of ∼ 110 m sl. On the other hand, the standard RAMP model predicts three intermediate strong glacials. Since these models do not account for $CO_2$, and thus do not include anthropogenic climate change, these results can only be considered as baseline experiments of how the glacial cycles would evolve without the anthropogenic effect of rising atmospheric $CO_2$. Besides the ORB model, all other models located the start of the Holocene between 15-17 ka and its end around 4 ka in the past and 3 kyr in the future. This early end of the Holocene does not align with results from other models coupled to the carbon cycle, which project, without any anthropogenic effect, that the current interglacial conditions will last for approximately another 50,000 years (Ganopolski et al., 2016; Talento and Ganopolski, 2021). However, a slightly reduced pre-industrial atmospheric $CO_2$ concentration of 240 ppm instead of 280 ppm could have led to a new glacial inception that started already a couple of thousand years ago (Ganopolski et al., 2016). Excluding anthropogenic $CO_2$ emissions and extrapolating their analysis of candidate precession peaks, Barker et al. (2025) found that the next glacial inception would reach a maximum rate during the next 11kyr and should be terminated around 66 kyr in the future. This is in close agreement with the RAMP model, which predicts the next glacial termination in 65 kyr, while the ABR, GRAD and RAMP-l model all predict the next glacial termination in 105 to 108 kyr, which is in closer agreement with the prediction of ∼ 110 kyr by Talento and Ganopolski (2021).

### 5 Conclusions and outlook

In this study, we constructed an improved conceptual model of Quaternary global ice volume, including four different internal forcing scenarios, alongside another model configuration, designed to test the sensitivity of large ice sheets in the late Quaternary. We showed that the RAMP-l model, which incorporates a ramp-like internal forcing and an ice volume dependency in its threshold forcing, is the best-performing model. It is in favour of a long-lasting linear trend in the climate system rather than an abrupt change or a purely orbitally driven model. The optimal insolation metric for the model is the ISI above 285 W m$^{-2}$ at 67° N, demonstrating the importance of the obliquity signal for the evolutionary equations. It also suggests that large ice sheets in the late Quaternary (last 900 ka) are more sensitive to precession than obliquity and, therefore, more important in triggering glacial terminations. In addition, the RAMP and RAMP-l simulations yield a very early onset for the gradual trend around 2



Ma. This finding emphasizes the importance of intensifying the work to obtain an even older continuous ice core record than the recently drilled ∼ 1.2 Myr old ice from the European Beyond EPICA-Oldest Ice Core project.

At the moment, the conceptual models presented in this work only rely on orbital parameters as an input and they output the global ice volume. Internal feedback mechanisms and forcings are conceptualized by aggregated model parameters. This limits the physical interpretation of potential mechanisms driving climate evolution. Hence, it would be beneficial to include other climatic variables in future work, e.g. $CO_2$. This would allow to model the past evolution of other climatic variables and improve the physical interpretability of the results. Furthermore, it would make the models more robust since they would no longer depend on a single tuning target, which can lead to the reproduction of biases or errors, already present in the data.

*Code and data availability.* The source code of the conceptual model and all the code and data to re-create the figures are publicly available on GitHub: https://github.com/felyx04/Conceptual-Model-Pollak-et-al-2025. The version corresponding to this submitted manuscript is available on Zenodo: https://doi.org/10.5281/zenodo.15421084.

*Author contributions.* F.Po. and F.Pa. designed the conceptual model; F.Po. run the simulations with help of F.Pa.; Analysis performed by F.Po., with inputs from F.Pa., E.C., Z.C. and L.J.; F.Po. wrote a draft of the paper, with subsequent inputs from F.Pa., E.C., Z.C., L.J. and E.L.

*Competing interests.* The authors declare that there are no competing interests.

*Acknowledgements.* This study is an outcome of the AUFRANDE project, which is co-funded by the European Union under the Marie Skłodowska-Curie Grant Agreement No 101081465 (AUFRANDE). Views and opinions expressed are however those of the authors only and do not necessarily reflect those of the European Union or the Research Executive Agency. Neither the European Union nor the Research Executive Agency can be held responsible for them. This publication was also generated in the frame of Beyond EPICA. The project has received funding from the European Union's Horizon 2020 research and innovation programme under grant agreement No. 730258 (Oldest Ice) and No. 815384 (Oldest Ice Core). It is supported by national partners and funding agencies in Belgium, Denmark, France, Germany, Italy, Norway, Sweden, Switzerland, the Netherlands and the United Kingdom. Logistic support is mainly provided by ENEA and IPEV through the Concordia Station system. The opinions expressed and arguments employed herein do not necessarily reflect the official views of the European Union funding agency or other national funding bodies. This is Beyond EPICA publication number 45. E.C. received the financial support from the French National Research Agency under the 'Programme d'Investissements d'Avenir' through the Make Our Planet Great Again HOTCLIM project (ANR-19-MPGA-0001). E.L. was supported by the Belgian Science Policy Office (BELSPO; FROID project).



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



**Table A1.** Tuned parameter values for the best 2.6 Myr simulation runs of the five different models used in this study. The values are rounded to four decimal places. state$_{initial}$ sets the initial state in which the model starts the simulation. This parameter was not used as a tunable parameter, but instead always set to glacial mode.

| Parameter | ORB | ABR | GRAD | RAMP | RAMP-l |
|---|---|---|---|---|---|
| $\alpha_{\text{Esi}}$ (m kyr$^{-1}$) | 0.6079, | 0.0545 | -0.2797 | 0.0924 | 0.3835 |
| $\alpha_{\text{Eco}}$ (m kyr$^{-1}$) | -0.5484 | -0.2323 | -0.4106 | -0.1446 | -0.2152 |
| $\alpha_{\text{O}}$ (m kyr$^{-1}$) | 1.2801 | 0.8058 | 0.7678 | 0.5871 | 0.7968 |
| $\alpha_{\text{g}}$ (m kyr$^{-1}$) | 0.3237 | 0.8641 | 0.9035 | 0.8900 | 0.9908 |
| $\alpha_{\text{d}}$ (m kyr$^{-1}$) | -0.2705 | -0.8056 | 0.1736 | 0.1647 | -0.4266 |
| $\tau_d$ (kyr) | 1954.8438 | 7.0156 | 5.2301 | 6.3376 | 5.3454 |
| $k_{\text{Esi}}$ (m) | 583.8280 | 18.8947 | 21.8939 | 17.6969 | -10.7672 |
| $k_{\text{Eco}}$ (m) | 9198.7910 | 3.7642 | 7.7095 | 1.4041 | -6.1782 |
| $k_{\text{O}}$ (m) | 567.0562 | 6.3133 | 12.4559 | 11.5707 | 0.8189 |
| $v_0$ (m) | -4598.8362 | 96.7743 | 126.3573 | 98.3877 | 127.3623 |
| $v_1$ (m) | 4919.8064 | -1.7411 | -10.4710 | 2.3544 | -0.8391 |
| $v_i$ (m) | 19.8410 | 17.9954 | 9.2756 | 9.1767 | 17.9847 |
| state$_{initial}$ | glacial | glacial | glacial | glacial | glacial |
| $t_{abr}$ (ka) | / | 1246.0466 | / | / | / |
| $v_0'$ (m) | / | 10.5984 | / | -12.9160 | 20.6520 |
| $\tau_d'$ (kyr) | / | -113.2884 | / | 31.4175 | 41.1961 |
| $C_v$ (m kyr$^{-1}$) | / | / | 0.0409 | / | / |
| $C_\tau$ (m kyr$^{-1}$) | / | / | -0.0056 | / | / |
| $t_2$ (ka) | / | / | / | 2159.3461 | 1977.8274 |
| $t_1$ (ka) | / | / | / | 857.9326 | 454.2691 |
| $l_{\text{Esi}}$ (unitless) | / | / | / | / | 0.6750 |
| $l_{\text{Eco}}$ (unitless) | / | / | / | / | 0.1366 |
| $l_{\text{O}}$ (unitless) | / | / | / | / | 0.1446 |
| # Parameters (excluding state$_{initial}$) | 12 | 15 | 14 | 16 | 19 |



**Table A2.** Root mean square errors (RMSEs) and coefficient of determination ($R^2$) for the five different model configurations for various time periods.

| Period (Ma) | 2.6 - 1.2 | | 1.2 - 0.8 | | 0.8 - 0 | | 2.6 - 0 | |
|---|---|---|---|---|---|---|---|---|
| **Model** | **RMSE** | $R^2$ | **RMSE** | $R^2$ | **RMSE** | $R^2$ | **RMSE** | $R^2$ |
| ORB | 10.58 m | 0.49 | 17.12 m | 0.20 | 25.56 m | 0.18 | 17.50 m | 0.47 |
| ABR | 10.60 m | 0.49 | 13.90 m | 0.47 | 13.92 m | 0.76 | 12.24 m | 0.74 |
| GRAD | 10.06 m | 0.54 | 13.18 m | 0.52 | 13.87 m | 0.76 | 11.86 m | 0.76 |
| RAMP | 10.01 m | 0.54 | 11.73 m | 0.62 | 13.30 m | 0.78 | 11.38 m | 0.78 |
| RAMP-l | 8.14 m | 0.70 | 13.09 m | 0.53 | 11.16 m | 0.84 | 10.02 m | 0.83 |



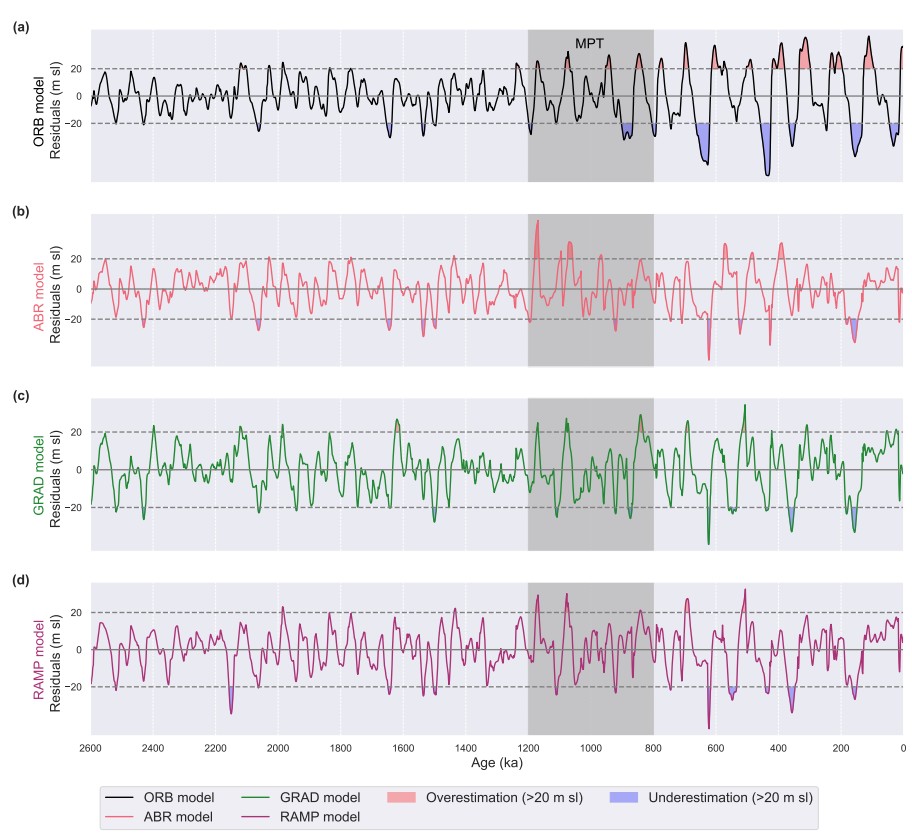

**Figure B1.** Residuals of the four different model configurations wrt the ice-volume reconstruction by Berends et al. (2021a). The residuals are calculated as *model-data*. The residuals are shown for the ORB (**a**, black curve), ABR (**b**, red curve), GRAD (**c**, green curve) and the RAMP model (**d**, purple curve) over the whole Pleistocene. The blue-shaded area marks where the simulation exceeds a residual of −20 m, while the red-shaded area marks where the simulation exceeds a residual of +20 m.



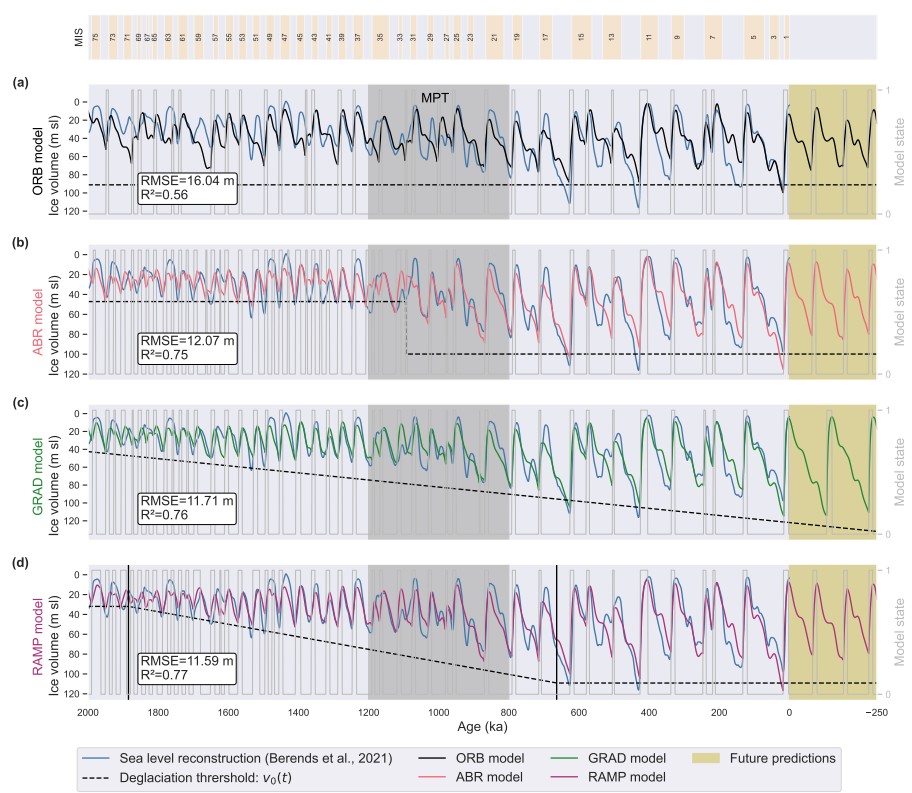

**Figure B2.** Comparison of the four different model configurations. The simulated ice volume curves are shown for the ORB (**a**, black curve), ABR (**b**, red curve), GRAD (**c**, green curve) and the RAMP model (**d**, purple curve) over the last 2 Ma. Each simulation is plotted over the ice volume reconstruction by Berends et al. (2021a) (blue curve). The deglaciation parameter $v_0(t)$ (black dotted line) is shown for each simulation, indicating the internal forcing scenario. The yellow shaded area marks the future 250 kyr and the respective predictions by each model. The grey-shaded area highlights the classical perspective of where the MPT is located in time. MIS boundaries given according to Lisiecki and Raymo (2005).



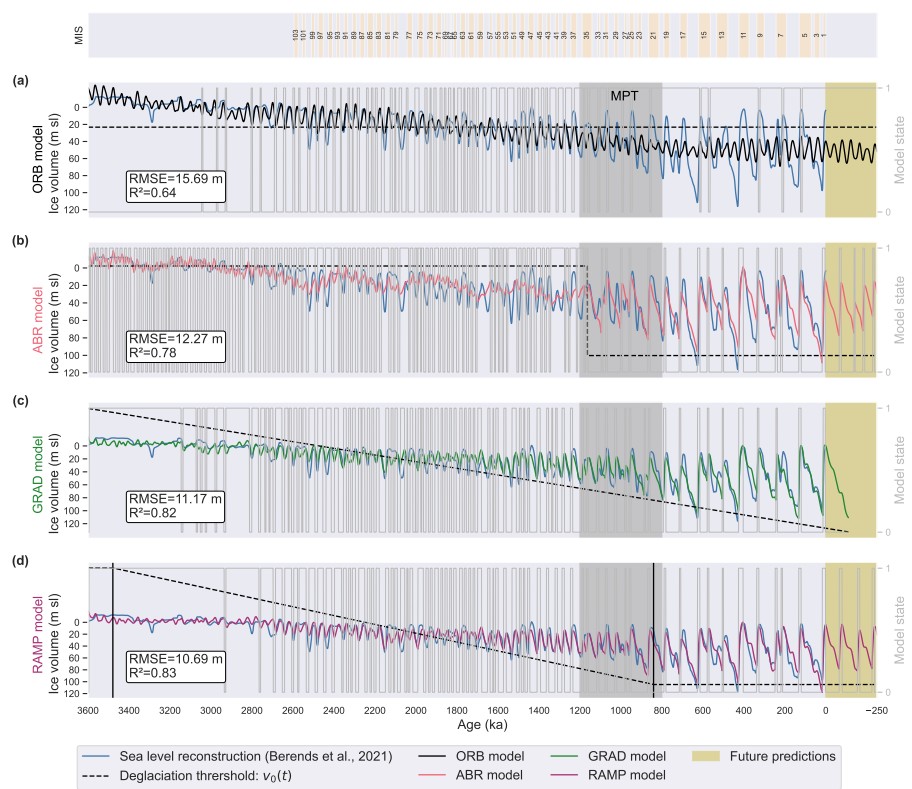

**Figure B3.** Comparison of the four different model configurations. The simulated ice volume curves are shown for the ORB (**a**, black curve), ABR (**b**, red curve), GRAD (**c**, green curve) and the RAMP model (**d**, purple curve) over the last 3.6 Ma. Each simulation is plotted over the ice volume reconstruction by Berends et al. (2021a) (blue curve). The deglaciation parameter $v_0(t)$ (black dotted line) is shown for each simulation, indicating the internal forcing scenario. The yellow shaded area marks the future 250 kyr and the respective predictions by each model. The grey-shaded area highlights the classical perspective of where the MPT is located in time. MIS boundaries given according to Lisiecki and Raymo (2005).

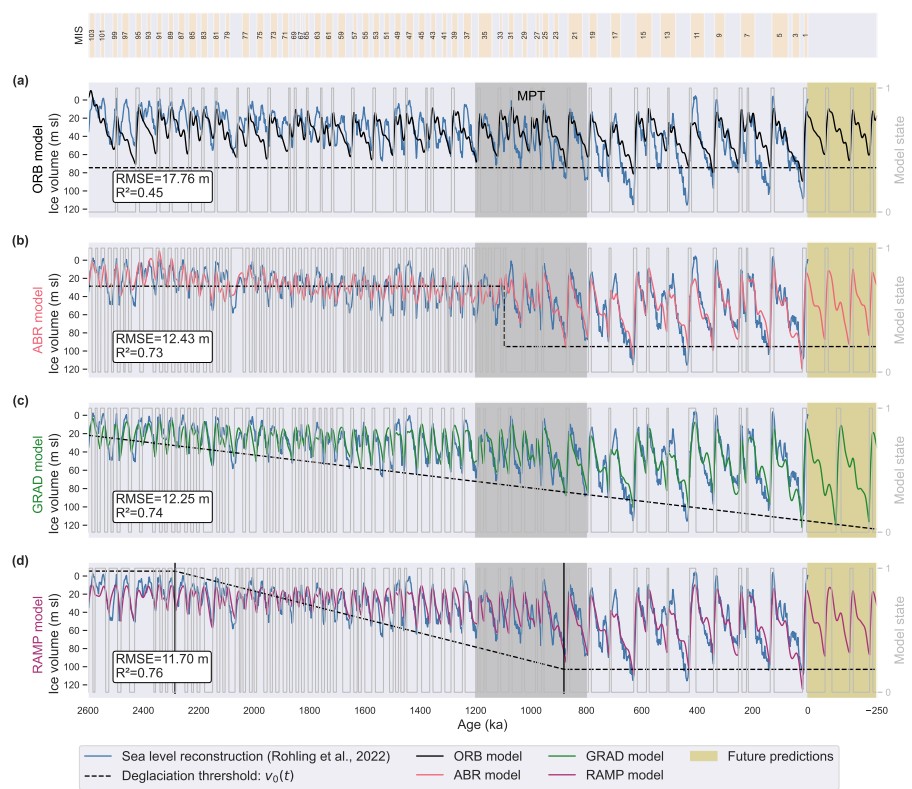

**Figure B4.** (Same Figure as Figure 2, but models are based on a different ice volume reconstruction) Comparison of the four different model configurations. The simulated ice volume curves are shown for the ORB (**a**, black curve), ABR (**b**, red curve), GRAD (**c**, green curve) and the RAMP model (**d**, purple curve) over the whole Pleistocene. Each simulation is plotted over the ice volume reconstruction by Rohling et al. (2022) (blue curve). The deglaciation parameter $v_0(t)$ (black dotted line) is shown for each simulation, indicating the internal forcing scenario. The yellow shaded area marks the future 250 kyr and the respective predictions by each model. The grey-shaded area highlights the classical perspective of where the MPT is located in time. MIS boundaries given according to Lisiecki and Raymo (2005).



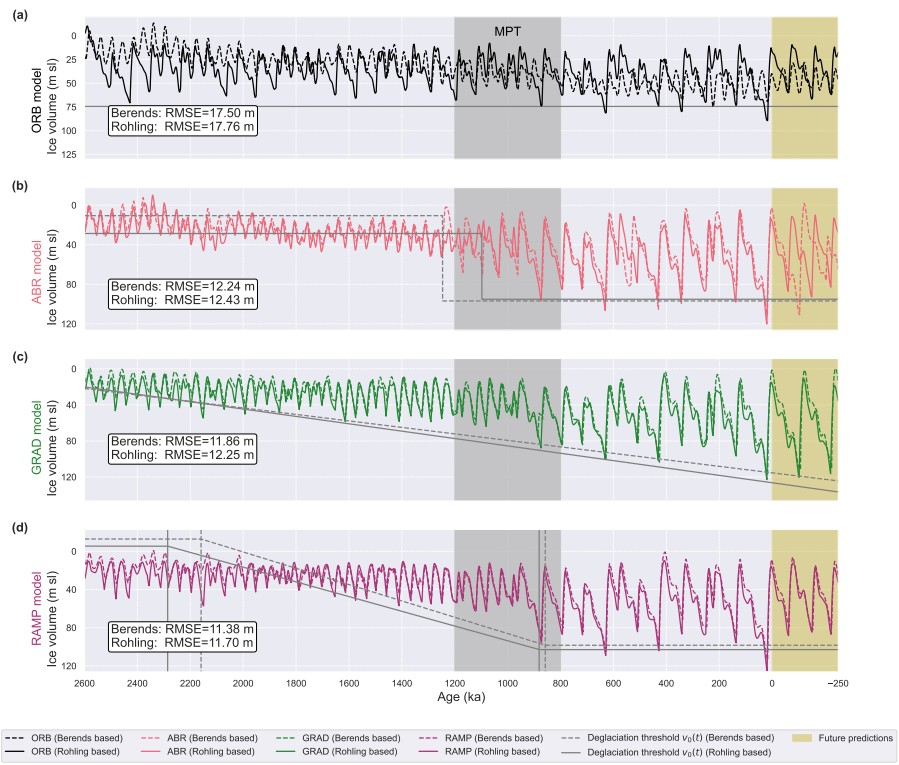

**Figure B5.** Comparison of the four model configurations, tuned for different targets, over the whole Pleistocene. Each panel shows one of the four model configurations (ORB, ABR, GRAD and RAMP model), once tuned for the Berends sea-level reconstruction (coloured dashed lines) and once tuned for the Rohling sea-level reconstruction (coloured solid lines). The corresponding deglaciation parameter $v_0(t)$ is shown for each model, indicating the internal forcing scenario (Berends-based model: grey dashed line, Rohling-based model: grey solid line). The yellow shaded area marks the future 250 kyr and the respective predictions by each model. The grey-shaded area highlights the classical perspective of where the MPT is located in time.



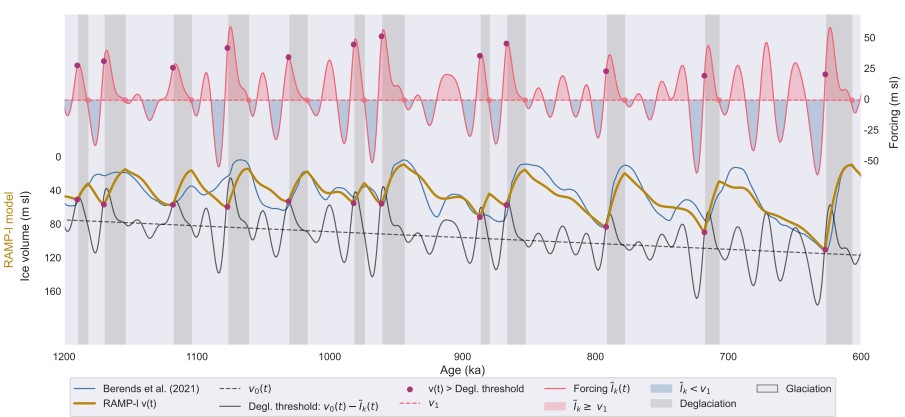

**Figure B6.** Simulated ice volume by the RAMP-l model (golden line) for period 1.2 - 0.6 Ma, together with its deglaciation and glaciation thresholds. The forcing $\tilde{I}_k(t)$ is shown in red. While the red shaded area marks forcing values above the model threshold $v_1$, the blue shaded area marks values below. The time-dependent deglaciation parameter $v_0(t)$ is represented by the black dotted line, while the solid black line indicates the deglaciation threshold. The purple dots indicate terminations, and the red dots mark glacial inceptions.