# Peer review of "An improved conceptual model of Quaternary global ice volume and the Mid-Pleistocene Transition"

_EGUsphere, 2025_

## Referee Comment (RC2)

The manuscript by Pollack et al. presents the results of global ice volume simulations performed using a model based on the Parrenin and Paillard (2012) model. The reported results represent incremental progress compared to those presented in the recent Legrain et al. (2023) paper. The main novelties are the increased number of model parameters, a more advanced procedure for model calibration, and the simulations of future glacial cycles. Below I will not only review the manuscript by Pollack, but also present my more general appraisal of their modelling approach.

**General comments**

**Conceptual model**. The model used in this study originates from the Paillard conceptual model of glacial cycles (Paillard, 1998). This model was then considerably revised by Parrenin and Paillard (2012) (hereafter referred to as PP12). Firstly, the authors of this paper abandoned Milankovitch theory of glacial cycles and delegated the determination of the 'orbital forcing' to the optimization algorithm. Secondly, they introduced two completely different 'orbital forcings': one determines the linear response of the global volume while another one determines nonlinear response, namely transitions between glacial and deglaciation regimes. However, since they tune the model only to glacial cycles of the late Quaternary, both 'orbital forcings' at least qualitatively resembles each other and boreal summer insolation. Unfortunately, the authors did not explain why they need two orbital forcings instead of one. In Legrain et al. (2023) (hereafter referred to as L23), the same model has been applied already to the entire Quaternary, and the model parameters were fitted to Berends global ice volume simulations. Since Berends ice volume contains very little precession even for the late Quaternary, the result of such tuning is easy to understand. The 'linear orbital frorcing' ($I_\alpha$) is dominated by obliquity while 'nonlinear orbital forcing' ($I_k$) by presession. (How such a situation could occur in the real world is not explained). Without strong precessional component in $I_k$, it would not be possible to simulate glacial cycles of the 100-kyr world, but too much precession disturbs the 41-kyr world. This is why, the optimization algorithm chooses model parameters for the early Quaternary in such a way that another central element of Pallard's conceptual model - the existence of two distinct glaciation and deglaciation regimes - also vanishes. As can be seen in Fig. 2, the model switches several times between glaciation and deglaciation regimes during a single glacial cycle. As the results, what is described in the paper is not a conceptual model but rather a mathematical immolator of glacial cycles, where a good agreements with 'reconstructions' is achieved by using 16, 17 or even 19 model parameters, the physical meaning of which is hard to understand.

**Model description**. Although the model used in the study has been described in several papers, some aspects require clarifications and corrections. Firstly, it is unclear whether conditions for regime changes denoted by (i) and (ii) should be met simultaneously. According to PP12 it seems that both conditions must be met :

$$\mathbf{g-to-d} : \kappa_{Esi}Esi + \kappa_{Eco}Eco + \kappa_O O + v > v_0,$$
$$(\text{and } \kappa_{Esi}Esi + \kappa_{Eco}Eco + \kappa_O O \geq v_1)$$

but it is unclear to me why the second condition is in brackets.

Secondly, the model employs two different times denoted by the same letter '$t$'! One time, used in the differential equation for ice volume has 'physical' (i.e. normal) direction, while the second time, which determines the evolution of model parameters, goes in the 'paleo' (i.e. reverse) direction. This is very confusing. Using of the same notations ($v_0$ and $\tau_d$) for two different characteristics (one constant and another time-dependent) is also not a good idea.

**Improved model**. The title of the paper begins from "*An improved conceptual model*", but, surprisingly, it is not clear from the manuscript what they meant under "improved model". The manuscript makes an impression that the 'improved model' is RAMP-l but in page 27 it is written '*we*

*constructed an improved conceptual model of Quaternary global ice volume, including four different internal forcing scenarios, alongside another model configuration... We showed that the RAMP-l model...*' I am not sure how this should be interpreted.

The improvements compared to L23 are also not clearly described. Only after the comparison of the equations in Pollak and L23, one can understand that 'deprecated truncation function' means that the authors decided not to use truncation of insolation introduced in Paillard (1998) and then uses in PP12 and L23. Why the authors consider this an improvement is not explained. At the same time, the author introduced a new time-dependent parameter td which is the relaxation time scale during deglaciation and which is in Paillard (1998) model determined the duration of glacial termination. However, what is the meaning of this parameter in Pollak et al. is unclear: during the early Quaternary its value in GRAD, RAMP and RAMP-l models is about 40 kyr, i.e. close to the duration of the entire glacial cycles. Even worse, in ABR model version, the relaxation time scale is -113 kyr, but the 'relaxation time scale' cannot be negative by definition. The meaning of negative $v_0$ in the RAMP model is equally hard to interpret.

**Insolation**. According to the Milankovitch theory, changes in boreal summer insolation is the driver of glacial cycles, but since PP12 abandoned Milankovitch theory, it is rather surprising that the term 'insolation' appears in section 3.5. The term 'insolation' has a very clear meaning: 'insolation' is the abbreviation for 'incoming solar radiation' and is measured in $W/m^2$ (or equivalent units). The 'orbital forcings' $I\alpha$ and $I_k$ used in the paper have nothing to do with the real insolation and therefore the terms 'insolation' and 'insolation maximum' in the context of the paper is misleading.

**Ice volume reconstruction**. The author used the Berends reconstruction of global ice volume. Like any other reconstruction, the Berends reconstruction contains significant uncertainties and deficiencies. For example, for the Last Glacial Maximum, for which there are numerous independent data, Berends underestimates global ice volume by more than 20% and also underestimates ice volume variability during MIS5. For MIS3 different reconstructions for global ice volume range between 30 and 90 meters (Farmer et al., 2023 PNAS), i.e. the uncertainties are about 50%. It is very likely that for the early times, the uncertainties are even larger. This makes reported improvements in RMSE order of several meters completely insignificant. Even more strange to see 6, 7 and even 8-digit numbers in Table A1. It is clear that so many digits originate from Monte Carlo, but in natural sciences, it is customary to report only significant digits. The paper also says nothing about the robustness of the modelling results with respect to the choice of model parameters.

**Simulation of MPT**. The main objective of the paper (similarly to L23) is to simulate MPT. Clearly, MPT cannot be explained by orbital forcing alone; therefore, the purpose of the ORB version is unclear. Secondly, even without modelling, it is clear from data analysis alone that the MPT (Mid-Pleistocene Transition) was a transition, not an event, and that it lasted for at least several hundred thousand years. This is why the purpose of the repeating of ABR scenarios is unclear. Regarding the mechanism(s) of the MPT (which are still debatable), it is unclear how the experiments presented in this manuscript can shed light on the cause of the MPT transition. The problem with strongly nonlinear systems, such as the Earth system, is that even gradual changes of the controlling parameters can cause a rather abrupt regime changes (e.g. Willeit et al., 2019). It is quite possible that the gradual decline in $CO_2$ and landscape evolution (including regolith removal) began not only well before the MPT but also before the Quaternary, and there is no way to derive this from the experiments described in the manuscript.

**The role of precession.** Firstly, the model used in the study of Pollak cannot be used to study the role of precession and obliquity. This is not a physically-based model and the model parameters are just chosen by the optimization algorithm in such a way that glacial termination of late Quaternary occur in the right times. And this 'right times' coincide with the periods of rising boreal summer insolation.

In turn, boreal summer insolation is dominated by precession. This is why the optimization algorithm picked up precession.

Secondly, the fact that glacial terminations of the late Quaternary are mainly determined by precession has been known well before Barker et al. (2025). Already Raymo (1997) noted that 'the length between subsequent terminations is either four or five precessional cycles long'. This idea was further developed by Ridgewell et al. (1999) who wrote that 'the spectral signature of $\delta^{18}O$ records are entirely consistent with Milankovitch mechanisms in which deglaciations are triggered every fourth or fifth precessional cycle'.

**Future glacial cycle simulations.** The authors also used their models to simulate future glacial cycles. According to these simulations, the next glacial cycle has either already begun or will begin soon in the absence of anthropogenic influence. The authors are, of course, aware that this contradicts to the results of the physically-based models and is therefore likely to be incorrect. This is why they attempted to defend their model by arguing that in Ganopolski et al. (2016) glacial inception occurs with the current orbital forcing if pre-industrial $CO_2$ were 240 ppm. This is absolutely correct, but among many uncertainties, there is one thing which we know for sure – the preindustrial concentration was 280 ppm, and this value is typical for post-MBT interglacials. Therefore, the unprecedently long Holocene is the robust and most striking feature of the next 100 kyr. Since this feature is not reproduced by Pollak et al., the value of such modelling exercises is called into serious doubt.

**Model performance**. The authors compared their version of the PP12 model with the L23 model, as well as the 19-parameters RAMP-l with the 16-parameters RAMP and using BIC, they claimed significant improvements. However, the MiM (minimal model which I reported in my 2024 paper), which is a simplification of the Leloup and Paillard (2022) model (which in turn is a simplification of original Paillard 1998 model) simulates the last 800 kyr (arguably the most interesting and difficult period of climate history) as good as RAMP (R2=0.73). But MiM has only 4 parameters (nondimensional version of MiM described in Ganopolski 2024 has three parameters). By contrast, RAMP uses 12 parameters to simulate the late Quaternary. The natural question is for what purpose we need models with so many parameters and what can we learn from them.

**Specific comments**

L10. The meaning of '*internal focing*' is unclear. The authors prescribed temporal evolution of several model parameters, not forcing.

L.12 '*support the idea of a long-term climatic shift as a cause of the MPT*'. In fact, MPT is the climate shift, whereas the cause(s) of MPT may have nothing to do with climate, like the 'regolith hypothesis'.

L.131. Unclear what the authors meant under '*versatile model*' since even a slight change in the temporal scenario (GRAD to RAMP) causes significant changes of all other model parameters.

L. 147. '*a linear combination of three orbital parameters normalized to zero mean and unit variance that can reproduce the insolation at most latitudes and seasons*'. This is incorrect. An arbitrary combination of these three parameters does not reproduce insolation at any latitude and at any time.

L. 168. '*the model can also incorporate an internal forcing mechanism to account for non-linear feedback mechanisms within the climate system*'. Which forcing and feedbacks are meant here is unclear.

Table 3. Why number of model parameters in Table 3 does not coincide with Table A1.

L. 460. '*each or every second insolation peak resulted in a deglaciation*'. What is meant under insolation peak is unclear.

L. 502 '*integrated summer insolation (ISI) reported in the literature*'. Firstly, this is not an appropriate way to refer to the published concept. ISI was introduced by Huybers in his 2006 Science paper. Secondly, the authors should explain how ISI is defined. Thirdly, ISI was introduced by Huybers to support his (former) idea that glacial cycles are paced primary by obliquity; this is why ISI is defined such a way that it is completely dominated by obliquity. But since $I_\alpha$, is also dominated by obliquity it is not surprising that two obliquity-dominated curves resemble each other. What can be learned from such a comparison?

L. 565. '*This suggests the presence of non-linearities within the climate system that modify Earth's response to changes in the orbital forcing. This conclusion aligns with findings from other studies (Clark et al., 2006; Leloup and Paillard, 2022; Berends et al., 2021b)*'. I am just curious whether the authors are aware of the works of Weertman, MacAyeal, Oerlemans and others who discussed how nonlinearities modify the Earth's response to orbital forcing already during 1970s and early 1980s? As for the Berends et al. (2021b) paper, which the authors cited ten times, it is merely a review paper. Although review papers are useful reference material, especially for early career scientists, citing them cannot substitute for reading and citing real scientific papers.

---

## Author Comment (AC1)

**Response to Anonymous Referee #1**

Dear Referee,
Thank you for reviewing our manuscript and for your effort and time spent on this. Your constructive feedback will help to improve the final revised version of this manuscript.

Based on this feedback, we will substantially revise the manuscript with the following main changes:

- we will reduce the complexity of the used model to rely on fewer parameters

- The benthic $\delta^{18}O$ curve (probstack with removed trend) from Clark et al. (2025) will be used as a new main target. Berends and Rohling SL curves will only be shown in SI

- ORB, ABR and GRAD models will be removed to focus on the novel RAMP model. This will make the manuscript more concise and highlight the novelities compared to the Legrain et al. (2023) model

In the following, we respond to your comments and propose several changes to the manuscript, motivated by your suggestions.

Author comments
Referee comments

This paper improves an exisiting conceptual model on global ice volume changes as function of orbital forcing and some internal feedbacks which is then applied to the last 2.6 Ma in order to understand the Mid-Pleistocene Transition (MPT).
In my view this is a solid piece of work, but misses one important piece of discussion. All sea level (=global ice volume) reconstructions used here as target are some sort of deconvolution of the LR04 benthic d18O stack into sea level and temperature. However, a most recent approach by Clark et al., published some weeks ago in Climate of the Past (doi: 10.5194/cp-21-973-2025) doing the same thing supported by SST and deep ocean data comes to a significantly different deconvolution with larger glacial/interglacial amplitudes in the sea level component throughout the Quaternary than the sea level records used here.
In an ideal world, this new record (data to figure 10b in Clark et al., 2025 available in the SI there) would as an alternative be used here as tuning target, but I understand that this would be a major effort and leave it up to the authors if they want to jump on this challenge. However, I noted that the authors referred to another discussion paper (Scherrenberg et al., 2024), so it is not clear to me why they not also included in their discussion of the MPT the Clark et al paper, whose discussion version was also available since September 2024. What should be and needs to be done is that this difference between the sea level component in the new Clark et al 2025 paper and those used here needs to be discussed. This is in my view important for two reasons.

The other referees agree with the point of not using the Berends sea level reconstruction as the main target for the model. Therefore, we will use the benthic $\delta^{18}O$ (Probstack with

removed trend) from Clark et al. (2025) as our new main tuning target and for model evaluation in the revised version of this manuscript.

1. The Clark paper is simply one of the newest papers published on MPT sea level changes (although only indirectly contained in the contribution to the benthic d18O stack) and should be considered/discussed in any upcoming paper on the same topic.

We agree. Other conceptual models also used benthic $\delta^{18}O$ as model targets, e.g. a normalised or scaled LR04 stack is used in the Leloup and Paillard (2022) and the Ganopolski (2024) MiM models.

2. The regolith hypothesis, formulated some decades ago, also by Clark and others, was trying to explain that there were ice sheets in North America, that were reaching as far south as 39-40°N around 2.5 Ma ago (e.g. Balco & Royery, 2010, doi: 10.1130/G30946.1). However, more complex models are not yet able to simulate these ice sheet extends. The Utrecht model (in the version of de Boer et al 2014, doi:10.1038/ncomms3999) was failing to do so, as has been shown in Köhler & van de Wal 2020 (doi: 10.1038/s41467-020-18897-5), which plotted in their Fig. 2b the latitudinal ice sheet extend of the Utrecht model as function of time.

(Side note: Köhler & van de Wal 2020 is to some extend a reinterpretation of Tzedakis et al 2017 on the appearance of interglacials during the Quaternary and might for that content also be discussed here as one recent study on the understanding of the MPT.)

We agree. Since our model relies on a specific switch to transition between a deglaciation and glaciation state, we propose to add the following paragraph to our discussion:

"The model presented above depends on a specific threshold choice to switch between a glaciation and a deglaciation state. Köhler and van de Wal (2020) challenge this binary view of interglacials and glacials. By classifying interglacials based on the absence of substantial NH land ice outside of Greenland, they found that the classification of interglacials, especially in the early Quaternary, is ambiguous. This classification depends on the defined threshold and the choice of the underlying record. This perspective questions the ability of any threshold-based, two-state model to unambiguously classify interglacial states. Therefore, the identified glacial and deglacial states in our model should only be carefully considered in combination with the applied thresholds, defined in Eq. xy, and for the used target record (benthic $\delta^{18}O$, sea-water $\delta^{18}O$ or global ice volume)."

The sea level of Berends et al (2021) used here is a follow-up study of the de Boer et al 2014, both using a model to deconvolve benthic d18O into sea level and temperature, thus both results differ in detail, but rely on the same approach and give in principle comparable results. Also the 3 Ma long simulations of the CLIMBER model (Willeit et al., 2019, doi: 10.1126/sciadv.aav7337), which is a more complex climate model than in the Utrecht approach, and which does not use benthic d18O as input, do not get large ice sheets down to 40°N at 2.5 Ma BP.

That said, these interpretations of the benthic d18O stack are still failing to explain the terrestrial evidence of Balco and others. These shortcomings are also worth mentioning in the discussion.

We agree. The Berends and Rohling sea-level curves will no longer be the main target of our model, but instead be shown in the supplementary material. To discuss the limitations of these model-based sea level reconstructions, we propose to add the following paragraph to our discussion:

"Available global mean sea level data exhibit large uncertainties. Between 50 and 30 ka, geological and geochemical reconstructions of GMSL vary up to 60 m and above (Farmer et al., 2023). Model-based deconvolutions of global $\delta^{18}$O into GMSL, like for the Berends et al. (2021) and Rohling et al. (2022) sea level reconstructions, exhibit similar large uncertainties. While the Berends curve shows an LGM lowstand of around 100 m, the Rohling curve gives around 108 m. Both values are well below observational-based reconstructions of around 130 - 135 m for the LGM (Austermann et al., 2013; Lambeck et al., 2014; Yokoyama et al., 2000). These large uncertainties limit the usability of sea level reconstructions as targets for conceptual models, as presented in this study."

With this two contrasting deconvolutions of the benthic d18O stack - one based on ice sheet model, one based on an ocean temperature data compilation - such a simple model as used here might be able to give ideas how to understand them. Eg if completely different conceptual models are necessary to satisfy both data sets, but again, maybe this is a task for a future study, especially since a final interpretation of the sea level related changes in benthic d18O from Clark et al is still not published. Nevertheless, I would say it is a missed opportunity if the authors decide not jump on this issue here and now.

We agree. In the revised version of this manuscript, we will use the benthic $\delta^{18}$O record from Clark et al. (2025) as our new main tuning target (Fig. 1b). Our reduced RAMP model yields similar results for this target, i.e. a long-lasting trend for the ramp, which started over 2 Ma. While the model performs well on this target, it has more difficulties with the seawater $\delta^{18}$O record from Clark et al. (2025) (Fig. 1c), as can be seen by comparing the R values. In general, the model cannot accurately reproduce the larger pre-MPT amplitudes of the glacial-interglacial cycles and the decreasing trend of $\delta^{18}O_{sw}$ during the MPT. The RAMP formulation seems to be less suited for this target, as can be seen by the identified ramp period, which lies now in the interval $\sim$ 400 - 200 ka. For the $\delta^{18}O_{sw}$ target, the model also cannot reproduce the shift towards the 100-kyr periodicity correctly, since it only shows a 100-kyr signal for the last 200 ka.

Minor issues:

- Barker et al. 2025, explaining the role of orbital parameters for the 100k-world after the MPT also gives a glimpse on their roles in the 41k-world (their SI Figure 8). Thus, I believe this paper is one of the most recent studies discussing underlying processes of the MPT and should already be discussed widely in the introduction.

[Figure]

Figure 1: Reduced RAMP model optimised for three different tuning targets: Berends SL curve (a), Clark et al. (2025) benthic $\delta^{18}O$ (b) and Clark et al. (2025) seawater $\delta^{18}O$ (c).

We agree. L. 651 f. will be shifted to the introduction and extended in the following way to also mention the 41-kyr world:

Recent studies suggest that the duration and timing of deglaciation and glaciation events over the last 900 ka were largely deterministic and driven by the relative phasing of precession, obliquity, and eccentricity (Barker et al., 2022, 2025). Barker et al. identified candidate precession peaks, which are precession peaks that begin while obliquity is increasing, as essential for glacial terminations. They found that all glacial terminations during the last 900 ka correspond to the first candidate precession peak after a minimum in eccentricity. This is how the 100 kyr periodicity comes into play for the post-MPT world. On the other hand, it seems like obliquity alone controls the following glacial inception, which is triggered by the start of its next decreasing phase. During and prior to the MPT, the authors found that almost all candidate precession peaks are linked to glacial terminations. Since they depend on rising obliquity values, they are mainly paced by obliquity, resulting in the observed 41-kyr world with no more influence of eccentricity on these cycles.

- Please update the reference to Scherrenberg et al., 2024 to the now available final version: https://doi.org/10.5194/cp-21-1061-2025).

Corrected.

- Figure 4: purple and red dots look the same. Make one symbol differently, eg squares.

Corrected.

- lines 520ff: Somehow now „glacial inception" is shortened to „gl. inception". Please use full words.

Corrected.

- line 614: The most recent review on MPT SST changes is in Clark et al. 2024 (doi:10.1126/science.adi1908). please revise your discussion here based on that paper, instead of using the older study by McCymont et al 2013.

We agree. We propose to replace the sentence starting in L. 613 with the following:

"A synthesis of globally distributed SST records by Clark et al. (2024) found that two long-term cooling stages occurred during the last 4.5 Ma. The first started around 4 Ma, which was followed by a second period of intensified cooling between 1.5 Ma and around 0.8 Ma. Thereafter, temperatures stabilised for the late Pleistocene."

- line 618: $\delta^{13}$C instead of $\delta_{13}$C

Corrected.

- Section 4.5. The starting sentence should already say that such a future exercise would be

one which would ignore anthropogenic impacts.

We agree. We propose to shift the sentence in L. 680 to become the second sentence in this paragraph. The new paragraph would start as following:

"Another feature of the conceptual models presented in this study is their ability to extrapolate future glacial-interglacial cycles, in the absence of any anthropogenic impacts. Since these models do not account for $CO_2$, and thus do not include anthropogenic climate change, these results can only be considered as baseline experiments of how the glacial cycles would evolve without the anthropogenic effect of rising atmospheric $CO_2$."

- line 688:"... would reach a maximum rate". A rate of what? Glacial inception can hardly be a rate.

Instead of defining sharp transitions between interglacials and glacials based on thresholds in sea level or $\delta^{18}O$, Barker et al. (2025) refer in their work to the maximum rate of $\delta^{18}O$ increase, denoted as the maximum in glacial inception, which returns the Climate system to the next glacial cycle. To avoid a misleading interpretation of this sentence, we propose to replace it with the following sentence:

"Excluding anthropogenic $CO_2$ emissions and extrapolating their analysis of candidate precession peaks, Barker et al. (2025) found that the next glaciation would reach a maximum rate, i.e. the change in $\delta^{18}O$, during the next 11kyr and should be terminated around 66 kyr in the future."

- The content of the Appendices (Tables A1-A2, Figures B1-B6) are in my view actually Supplementary Figures which should appear in an SI (in the final format an extra PDF) and not as Appendices (added extra figures in the main PDF).

We agree. This will help to make the manuscript more concise.

**References**

J. Austermann, J. X. Mitrovica, K. Latychev, and G. A. Milne. Barbados-based estimate of ice volume at Last Glacial Maximum affected by subducted plate. *Nature Geoscience*, 6(7):553–557, July 2013. ISSN 1752-0908. doi: 10.1038/ngeo1859. URL https://www.nature.com/articles/ngeo1859. Publisher: Nature Publishing Group.

S. Barker, L. E. Lisiecki, G. Knorr, S. Nuber, and P. C. Tzedakis. Distinct roles for precession, obliquity, and eccentricity in Pleistocene 100-kyr glacial cycles. *Science*, 387(6737):eadp3491, Feb. 2025. doi: 10.1126/science.adp3491. URL https://www.science.org/doi/10.1126/science.adp3491. Publisher: American Association for the Advancement of Science.

P. U. Clark, J. D. Shakun, Y. Rosenthal, P. Köhler, and P. J. Bartlein. Global and regional temperature change over the past 4.5 million years. *Science*, 383(6685):884–890, Feb. 2024. doi: 10.1126/science.adi1908. URL https://www.science.org/doi/abs/10.1126/science.adi1908. Publisher: American Association for the Advancement of Science.

P. U. Clark, J. D. Shakun, Y. Rosenthal, C. Zhu, P. J. Bartlein, J. M. Gregory, P. Köhler, Z. Liu, and D. P. Schrag. Mean ocean temperature change and decomposition of the benthic $^{18}$O record over the past 4.5 million years. *Climate of the Past*, 21(6):973–1000, June 2025. ISSN 1814-9324. doi: 10.5194/cp-21-973-2025. URL `https://cp.copernicus.org/articles/21/973/2025/`. Publisher: Copernicus GmbH.

J. R. Farmer, T. Pico, O. M. Underwood, R. Cleveland Stout, J. Granger, T. M. Cronin, F. Fripiat, A. Martínez-García, G. H. Haug, and D. M. Sigman. The Bering Strait was flooded 10,000 years before the Last Glacial Maximum. *Proceedings of the National Academy of Sciences*, 120(1): e2206742119, Jan. 2023. doi: 10.1073/pnas.2206742119. URL `https://www.pnas.org/doi/abs/10.1073/pnas.2206742119`. Publisher: Proceedings of the National Academy of Sciences.

A. Ganopolski. Toward generalized Milankovitch theory (GMT). *Climate of the Past*, 20(1):151–185, Jan. 2024. ISSN 1814-9324. doi: 10.5194/cp-20-151-2024. URL `https://cp.copernicus.org/articles/20/151/2024/`. Publisher: Copernicus GmbH.

P. Köhler and R. S. W. van de Wal. Interglacials of the Quaternary defined by northern hemispheric land ice distribution outside of Greenland. *Nature Communications*, 11(1):5124, Oct. 2020. ISSN 2041-1723. doi: 10.1038/s41467-020-18897-5. URL `https://www.nature.com/articles/s41467-020-18897-5`. Publisher: Nature Publishing Group.

K. Lambeck, H. Rouby, A. Purcell, Y. Sun, and M. Sambridge. Sea level and global ice volumes from the Last Glacial Maximum to the Holocene. *Proceedings of the National Academy of Sciences*, 111(43):15296–15303, Oct. 2014. doi: 10.1073/pnas.1411762111. URL `https://www.pnas.org/doi/10.1073/pnas.1411762111`. Publisher: Proceedings of the National Academy of Sciences.

E. Legrain, F. Parrenin, and E. Capron. A gradual change is more likely to have caused the Mid-Pleistocene Transition than an abrupt event. *Communications Earth & Environment*, 4(1):1–10, Mar. 2023. ISSN 2662-4435. doi: 10.1038/s43247-023-00754-0. URL `https://www.nature.com/articles/s43247-023-00754-0`. Number: 1 Publisher: Nature Publishing Group.

G. Leloup and D. Paillard. Influence of the choice of insolation forcing on the results of a conceptual glacial cycle model. *Climate of the Past*, 18(3):547–558, Mar. 2022. ISSN 1814-9324. doi: 10.5194/cp-18-547-2022. URL `https://cp.copernicus.org/articles/18/547/2022/`. Publisher: Copernicus GmbH.

Y. Yokoyama, K. Lambeck, P. De Deckker, P. Johnston, and L. K. Fifield. Timing of the Last Glacial Maximum from observed sea-level minima. *Nature*, 406(6797):713–716, Aug. 2000. ISSN 1476-4687. doi: 10.1038/35021035. URL `https://www.nature.com/articles/35021035`. Publisher: Nature Publishing Group.

---

## Author Comment (AC2)

**Response to the review of Tijn Berends (Referee #3)**

Dear Referee,

Thank you for reviewing our manuscript and for your effort and time spent on this. Your constructive feedback will help to improve the final revised version of this manuscript.

Based on this feedback, we will substantially revise the manuscript with the following main changes:

- we will reduce the complexity of the used model to rely on fewer parameters

- The benthic $\delta^{18}O$ curve (probstack with removed trend) from Clark et al. (2025) will be used as a new main target. Berends and Rohling SL curves will only be shown in SI

- ORB, ABR and GRAD models will be removed to focus on the novel RAMP model. This will make the manuscript more concise and highlight the novelities compared to the Legrain et al. (2023) model

In the following, we respond to your comments and propose several changes to the manuscript, motivated by your suggestions.

Author comments
Referee comments

**Major comments**

My main issue with this manuscript is that it is a continuation of the work by Legrain et al. (2023), but it is difficult to understand from the manuscript how the new model and its results differ from those by Legrain et al. Right now, a reader who is not familiar with Legrain et al. could be given the false impression that the model and the experiments are entirely novel. (In fact, even for me it is difficult to spot the differences, and I actually reviewed the Legrain et al. paper!). To prevent this, when presenting the model equations, it should be made clear how and why they are different from those already published. Similarly, since three of the four experiments (ORB, ABR, and GRAD) were also performed by Legrain et al., the results of those experiments with the earlier model version should be shown for comparison, to illustrate the effect of the changes to the model.

We agree. One major improvement compared to the Legrain et al. model is of technical nature regarding the improved tuning strategy and the improved speed of the code. However, this was barely mentioned in the manuscript, since we wanted to focus on the model output rather than on these technical details. The model speed was significantly increased by translating the code into JAX (a python library which allows for accelerated computation and just-in-time compilation). Furthermore, we added three additional solver, namely ptemcee (parallel-tempered MCMC), dynesty (dynamic nested sampling) and PyMC. With the improved tuning procedure and with the increase in speed, which allows a more rigorous

tuning, we could already reduce the model data mismatch compared to the original Legrain et al. model, without any changes in the parameterisations. We propose to add a paragraph in the model section of the revised version of the manuscript, where we highlight the increase in speed and the newly added solver to the model.

A reason for repeating the ORB, ABR and GRAD models was to show that despite changes in the model, the results obtained in the Legrain paper remained the same, i.e. the ORB model failed in reconstructing the MPT, while the GRAD model yielded better results than the ABR model. However, since the new RAMP model can converge to a constant $v_0$ parameter, an abrupt-like change, a gradual trend over the entire Quaternary, or something in between, it already tests the ORB, ABR and GRAD scenarios and yields the most likely forcing scenario. Therefore, we agree that there is no longer a need to include the ORB, ABR and GRAD models. They will be removed in the revised manuscript and the focus will be solely on the RAMP model. This will make the revised manuscript more concise and better highlight the novelities compared to the Legrain et al. (2023) paper.

My second issue regards the way the performance of the different model versions is assessed, namely by comparing to the sea-level curve from my 2021 paper (Berends et al., 2021a). I hesitate to call it a reconstruction (despite the rather optimistic title of that publication!), because, as I cautioned readers in that paper, that curve is based on a model-based deconvolution of the LR04 benthic d18O stack, and includes major uncertainties. For example, at the last glacial maximum, I simulated a sea-level drop of only 100 m, well below the accepted value of 130 m. As I stated in the discussion section of that paper, the results presented there "...should not be interpreted as a realistic reconstruction of what the world looked like in terms of global climate, ice-sheet geometry, sea level, and CO2 during these periods of geological history. Rather, ... they should be viewed as scenarios which can help in interpreting an expected new ice-core record." Considering the magnitude of the errors and uncertainties in this curve (i.e. up to 30 m), I do not think it can be used to (in)validate the results of the conceptual models presented here, which all lie well within this uncertainty (i.e. RMSE's of less than 20 m). Given that other sea-level curves (like the different ones from Rohling et al., which are also cited in the manuscript) suffer from similar uncertainties, in my view this means that the currently available sea-level data is insufficient to choose one of the author's models over another. This implies that the conclusion of the manuscript should be that, "until more accurate observations become available, both abrupt and gradual changes in the Earth system can explain the available (inaccurate) observations equally well". Of course, that does not mean it is not worthwhile to study the implications of abrupt vs. gradual changes, or of introducing non-linearities such as the ice-volume feedback in the RAMP-l model. But at present, such studies are, by necessity, exploratory in nature.

We agree. We will no longer use the Berends or Rohling sea level curves as our main tuning targets in the revised manuscript, but only show them in the SI. To discuss the limitations of these model-based sea level reconstructions, we propose to add the following paragraph to our discussion:

"Available global mean sea level data exhibit large uncertainties. Between 50 and 30 ka, geological and geochemical reconstructions of the GMSL vary up to 60 m and above (Farmer et al., 2023). Model-based deconvolutions of global $\delta^{18}O$ into GMSL, like for the Berends

et al. (2021) and Rohling et al. (2022) sea level reconstructions, exhibit similar large uncertainties. While the Berends curve shows an LGM lowstand of around 100 m, the Rohling curve gives around 108 m. Both values are well below observational-based reconstructions of around 130 - 135 m for the LGM (Austermann et al., 2013; Lambeck et al., 2014; Yokoyama et al., 2000). These large uncertainties limit the usability of sea level reconstructions as targets for conceptual models, as presented in this study."

Instead of using the Berends sea level curve, we will follow the advice from Referee 1 and use the benthic $\delta^{18}O$ record (probstack with trend removed) from Clark et al. (2025) as our main target. Furthermore, we will discuss the model performance on the newly available seawater $\delta^{18}O_{sw}$ record from Clark et al. (2025), based on a benthic $\delta^{18}O$ deconvolution motivated by an ocean temperature data compilation. It gives a contrasting view compared to the Berends and Rohling deconvolutions by exhibiting a rise in $\delta^{18}O_{sw}$ during the MPT, interrupting a previous period of increasing values and showing larger glacial interglacial amplitudes in the pre-MPT cycles. However, a final reconstruction of this $\delta^{18}O_{sw}$ record into sea level is not yet available. That's why we will mainly focus on the benthic record. Following the advice of Referee 2, we reduce the complexity of the RAMP model. The results of this reduced RAMP model for the three different tuning targets (Berends SL, Clark $\delta^{18}O_b$, Clark $\delta^{18}O_{sw}$) can be seen in Fig. 1. Our reduced RAMP model yields similar results for this target, i.e. a long-lasting trend for the ramp, which started over 2 Ma. While the model performs well on this target, it has more difficulties with the seawater $\delta^{18}O$ record from Clark et al. (2025) (Fig. 1c), as can be seen by comparing the R values. In general, the model cannot accurately reproduce the larger pre-MPT amplitudes of the glacial-interglacial cycles and the decreasing trend of $\delta^{18}O_{sw}$ during the MPT. The RAMP formulation seems to be less suited for this target, as can bee seen by the identified ramp period, which lies now in the interval $\sim$ 400 - 200 ka. For the $\delta^{18}O_{sw}$ target, the model also cannot reproduce the shift towards the 100-kyr periodicity correctly, since it only shows a 100-kyr signal for the last 200 ka.

The authors also apply their different model versions to the future. I do not immediately see the relevance of such experiments, given that present-day CO2 concentration far exceeds the Pleistocene range. The results presented here should, as far as I understand, be interpreted as "possible future patterns of glaciation in the absence of any anthropogenic CO2 emissions" – i.e., a thought experiment. Since such a thought experiment lies quite far from the scope of the rest of the manuscript, I think it might be best to leave it out altogether.

While neglected in most other conceptual models, we believe that extrapolating at least the next glacial cycle is an important tool of model evaluation, since this represents the only time interval in which a model cannot be tuned or fitted onto some existing paleoclimatic target curve. Of course, the extrapolated curves do not resemble nature, as they lack anthropogenic $CO_2$ emissions. Therefore, as stated by the referee, they must be interpreted as purely academic thought experiments. More complex models, coupled to the carbon cycle, project, even without any anthropogenic influence, an unprecedentedly long interglacial, lasting for another 50 kyr (Ganopolski et al., 2016; Talento and Ganopolski, 2021). In contrast to this, all of our models, no matter of which simulation period, project that the next glacial cycle has just started or is about to start. In their recent study, investigating the

[Figure]

Figure 1: Reduced RAMP model tuned on three different tergets: a) Berends et al. (2021) sea level curve, b) Clark et al. (2025) benthic $\delta^{18}O_b$ (probstack with trend removed) and c) Clark et al. (2025) sea water $\delta^{18}O_{sw}$.

influence of precession and obliquity on glacial interglacial cycles, Barker et al. (2025) esti-
mate that the current interglacial conditions would last for 11 kyr, when obliquity reaches
its next minimum and the succeeding glacial would be interrupted in around 66 kyr (again
neglecting anthropogenic effects). Other conceptual models similarly predict a contrasting
view for the next glaciation: Calder (1974) projected a start 5 ka and an end in around
119 kyr (see his Fig. 2), Imbrie and Imbrie (1980) estimated a start 6 ka (see their Fig. 7)
and Fig. 5 in Paillard (2015) reveals that only a large enough glaciation threshold can lead
to a prolonged Holocene in the Paillard (1998) model, while a lower threshold would have
terminated the Holocene already a few kyr ago. Therefore, we believe that the failure in
simulating a prolonged Holocene in our model, as well as in other conceptual models or
studies, is a striking feature to mention, which might be due to the missing carbon feedback
in the model or a glaciation threshold which only depends on orbital forcing or parameters.

To account for the Referee's comment, we propose to add some parts of the above paragraph
into the discussion and put more focus on the failure in simulating a prolonged Holocene in
the model.

Lastly, I believe that, with thirty pages of body (plus ten pages of bibliography and appen-
dices) the manuscript is too long, especially considering the rather simple model and limited
number of experiments. There is a lot of repetition going on, with e.g. the nature of the
four different experiments being described in the experimental set-up, in the results, and
then again in the discussion. A lot of text could, and therefore should, be removed without
losing information or coherence.

We agree. By removing the ORB, ABR and GRAD models, the revised version will be
significantly shorter. Furthermore, we follow the advice from Referee 1 and shift the appen-
dices into the SI. This will help to make the revised manuscript more concise.

**Minor comments**

Sect. 2.1: Please add a few lines explaining why it is physically justified to assign different
weights to the different orbital signals (eccentricity, obliquity, and precession) in the differ-
ent experiments. A flake of snow lying on top of an ice sheet cannot know the eccentricity
of Earth's orbit or the tilt of its axis; it can only feel how much solar heat it is receiv-
ing at that moment (which is a simple mathematical result of those orbital parameters,
with no room for "optimization"). Of course, it has to do with the asymmetry in the sensi-
tivity of the mass balance to summer vs. winter changes; that fact deserves to be mentioned.

The main idea behind using a linear combination of these three orbital parameters instead
of relying on a single metric, as has been done in many other models (e.g. Paillard (1998);
Ganopolski (2024); Leloup and Paillard (2022)), is to reduce a potential source of model
bias. We propose to include the following lines to the manuscript:

"Leloup and Paillard (2022) showed that the model outcome depends on the chosen insola-
tion forcing (e.g. summer solstice, caloric season, ISI above some threshold at 65° N). This

poses a bias to the model by selecting a specific insolation metric. Instead, we use a linear combination of precession, co-precession and obliquity, which can represent insolation at most latitudes and seasons (Imbrie et al., 2011). This approach has been used successfully in other models before (Imbrie et al., 2011; Legrain et al., 2023)."

Fig. 2: it looks like the ORB and ABR models produce higher-than-present sea levels between 2.6 and 2.2 Myr. Should I interpret this as retreat of the Antarctic ice sheet? How does your model handle negative values for the ice volume?

Negative values are possible in the model and handled normally. For extended runs longer than 2.6 Myr, they are needed to account for higher sea levels prior to the onset of Northern Hemisphere glaciations and are also visible in reconstructions by Rohling et al. (2022) and Berends et al. (2021). Due to the poor performance of the ORB model, there is no point in interpreting these negative results, since the model yields an unrealistic output. The ABR model does show some negative values around 2.4 Ma. Since the model has no spatial resolution, it does not allow for localising the melting location. However, it is most likely due to a combined melting of ice sheets from Antarctica, Greenland and glacier retreat compared to present-day conditions.

Fig. 3 and other places: I'd like to suggest again (as I did to Legrain et al. before) to use a wavelet transform (or something similar) to show the temporal changes in frequency content. This is much more intuitive and informative than showing time-averaged power spectra for two different blocks of time.

We agree. Wavelet scalograms, as in Fig. 1, will be used in the revised manuscript.

Table 2: Please explain why running the models over different lengths of time (2.0 Myr, 2.6 Myr, or 3.6 Myr) can yield such different results. Are the models also optimised separately?

Yes, the models were always optimised for the corresponding simulation period. The initial ice volume $v_i$ is a tunable parameter and has to be adjusted for each simulation period separately. Furthermore, the deglaciation parameter $v_0$ denotes for the ABR, GRAD and RAMP models the initial value of $v_0(t)$. This parameter must be different for the considered time periods as well. If the same set of parameters is used from a shorter to a longer simulation period, the obtained ice volume reconstruction would tend to have a positive offset during the start of the simulation, since the global ice volume gradually increased over time (visible in the Berends and Rohling reconstructions). To avoid this and to test how consistent the model output is, we always retuned the models for the specific time period. However, if the model output significantly differs from one time period to another, this hints towards the existence of multiple local minima in the parameter space or model deficiencies. The ORB model differed for all three time periods, since the solver could not identify a single best set of parameters for all three time periods. In combination with the failure to reproduce the MPT in any of these simulations, this shows the model deficiencies of the ORB model. To a smaller extent, this can be seen for the ABR model as well. Since it relies on a single constant $v_0$ parameter prior to the MPT, it struggles in reproducing the shift from smaller global ice volumes (at 3.6 Ma or 2.6 Ma), compared to the larger ones right before the MPT.

Only the GRAD and RAMP modes yielded similar results for all three simulation periods, showing their consistency, hence being the more favourable models.
Additionally, the reported RMSEs in Table 2 cannot be compared for different time periods, since the magnitude of the RMSE depends on the target data. Between 3.6 Ma and 2.8 Ma, the Berends reconstruction shows very small values with low variation, leading to lower RMSEs compared to the 2 Myr simulations.

**References**

J. Austermann, J. X. Mitrovica, K. Latychev, and G. A. Milne. Barbados-based estimate of ice volume at Last Glacial Maximum affected by subducted plate. *Nature Geoscience*, 6(7):553–557, July 2013. ISSN 1752-0908. doi: 10.1038/ngeo1859. URL `https://www.nature.com/articles/ngeo1859`. Publisher: Nature Publishing Group.

S. Barker, L. E. Lisiecki, G. Knorr, S. Nuber, and P. C. Tzedakis. Distinct roles for precession, obliquity, and eccentricity in Pleistocene 100-kyr glacial cycles. *Science*, 387(6737):eadp3491, Feb. 2025. doi: 10.1126/science.adp3491. URL `https://www.science.org/doi/10.1126/science.adp3491`. Publisher: American Association for the Advancement of Science.

C. J. Berends, B. de Boer, and R. S. W. van de Wal. Reconstructing the evolution of ice sheets, sea level, and atmospheric $CO_2$ during the past 3.6 million years. *Climate of the Past*, 17(1):361–377, Feb. 2021. ISSN 1814-9324. doi: 10.5194/cp-17-361-2021. URL `https://cp.copernicus.org/articles/17/361/2021/`. Publisher: Copernicus GmbH.

N. Calder. Arithmetic of ice ages. *Nature*, 252(5480):216–218, Nov. 1974. ISSN 1476-4687. doi: 10.1038/252216a0. URL `https://www.nature.com/articles/252216a0`. Publisher: Nature Publishing Group.

P. U. Clark, J. D. Shakun, Y. Rosenthal, C. Zhu, P. J. Bartlein, J. M. Gregory, P. Köhler, Z. Liu, and D. P. Schrag. Mean ocean temperature change and decomposition of the benthic $^{18}O$ record over the past 4.5 million years. *Climate of the Past*, 21(6):973–1000, June 2025. ISSN 1814-9324. doi: 10.5194/cp-21-973-2025. URL `https://cp.copernicus.org/articles/21/973/2025/`. Publisher: Copernicus GmbH.

J. R. Farmer, T. Pico, O. M. Underwood, R. Cleveland Stout, J. Granger, T. M. Cronin, F. Fripiat, A. Martínez-García, G. H. Haug, and D. M. Sigman. The Bering Strait was flooded 10,000 years before the Last Glacial Maximum. *Proceedings of the National Academy of Sciences*, 120(1): e2206742119, Jan. 2023. doi: 10.1073/pnas.2206742119. URL `https://www.pnas.org/doi/abs/10.1073/pnas.2206742119`. Publisher: Proceedings of the National Academy of Sciences.

A. Ganopolski. Toward generalized Milankovitch theory (GMT). *Climate of the Past*, 20(1):151–185, Jan. 2024. ISSN 1814-9324. doi: 10.5194/cp-20-151-2024. URL `https://cp.copernicus.org/articles/20/151/2024/`. Publisher: Copernicus GmbH.

A. Ganopolski, R. Winkelmann, and H. J. Schellnhuber. Critical insolation–CO2 relation for diagnosing past and future glacial inception. *Nature*, 529(7585):200–203, Jan. 2016. ISSN 1476-4687. doi: 10.1038/nature16494. URL `https://www.nature.com/articles/nature16494`. Publisher: Nature Publishing Group.

J. Imbrie and J. Z. Imbrie. Modeling the Climatic Response to Orbital Variations. *Science*, 207 (4434):943–953, Feb. 1980. doi: 10.1126/science.207.4434.943. URL `https://www.science.org/doi/10.1126/science.207.4434.943`. Publisher: American Association for the Advancement of Science.

J. Z. Imbrie, A. Imbrie-Moore, and L. E. Lisiecki. A phase-space model for Pleistocene ice volume. *Earth and Planetary Science Letters*, 307(1-2):94–102, July 2011. ISSN 0012821X. doi: 10.1016/j.epsl.2011.04.018. URL `http://arxiv.org/abs/1104.3610`. arXiv:1104.3610 [astro-ph, physics:physics].

K. Lambeck, H. Rouby, A. Purcell, Y. Sun, and M. Sambridge. Sea level and global ice volumes from the Last Glacial Maximum to the Holocene. *Proceedings of the National Academy of Sciences*, 111(43):15296–15303, Oct. 2014. doi: 10.1073/pnas.1411762111. URL `https://www.pnas.org/doi/10.1073/pnas.1411762111`. Publisher: Proceedings of the National Academy of Sciences.

E. Legrain, F. Parrenin, and E. Capron. A gradual change is more likely to have caused the Mid-Pleistocene Transition than an abrupt event. *Communications Earth & Environment*, 4(1):1–10, Mar. 2023. ISSN 2662-4435. doi: 10.1038/s43247-023-00754-0. URL `https://www.nature.com/articles/s43247-023-00754-0`. Number: 1 Publisher: Nature Publishing Group.

G. Leloup and D. Paillard. Influence of the choice of insolation forcing on the results of a conceptual glacial cycle model. *Climate of the Past*, 18(3):547–558, Mar. 2022. ISSN 1814-9324. doi: 10.5194/cp-18-547-2022. URL `https://cp.copernicus.org/articles/18/547/2022/`. Publisher: Copernicus GmbH.

D. Paillard. The timing of Pleistocene glaciations from a simple multiple-state climate model. *Nature*, 391(6665):378–381, Jan. 1998. ISSN 1476-4687. doi: 10.1038/34891. URL `https://www.nature.com/articles/34891`. Publisher: Nature Publishing Group.

D. Paillard. Quaternary glaciations: from observations to theories. *Quaternary Science Reviews*, 107:11–24, Jan. 2015. ISSN 0277-3791. doi: 10.1016/j.quascirev.2014.10.002. URL `https://www.sciencedirect.com/science/article/pii/S0277379114003898`.

E. J. Rohling, G. L. Foster, T. M. Gernon, K. M. Grant, D. Heslop, F. D. Hibbert, A. P. Roberts, and J. Yu. Comparison and Synthesis of Sea-Level and Deep-Sea Temperature Variations Over the Past 40 Million Years. *Reviews of Geophysics*, 60(4):e2022RG000775, 2022. ISSN 1944-9208. doi: 10.1029/2022RG000775. URL `https://onlinelibrary.wiley.com/doi/abs/10.1029/2022RG000775`. _eprint: https://onlinelibrary.wiley.com/doi/pdf/10.1029/2022RG000775.

S. Talento and A. Ganopolski. Reduced-complexity model for the impact of anthropogenic $CO_2$ emissions on future glacial cycles. *Earth System Dynamics*, 12(4):1275–1293, Nov. 2021. ISSN 2190-4979. doi: 10.5194/esd-12-1275-2021. URL `https://esd.copernicus.org/articles/12/1275/2021/`. Publisher: Copernicus GmbH.

Y. Yokoyama, K. Lambeck, P. De Deckker, P. Johnston, and L. K. Fifield. Timing of the Last Glacial Maximum from observed sea-level minima. *Nature*, 406(6797):713–716, Aug. 2000. ISSN 1476-4687. doi: 10.1038/35021035. URL `https://www.nature.com/articles/35021035`. Publisher: Nature Publishing Group.

---

## Author Comment (AC3)

**Response to the review of Andrey Ganopolski (Referee #2)**

Dear Referee,
Thank you for reviewing our manuscript and for your effort and time spent on this. Your constructive feedback will help to improve the final revised version of this manuscript.

Based on this feedback, we will substantially revise the manuscript with the following main changes:

- we will reduce the complexity of the used model to rely on fewer parameters

- The benthic $\delta^{18}$O curve (probstack with removed trend) from Clark et al. (2025) will be used as a new main target. Berends and Rohling SL curves will only be shown in SI

- ORB, ABR and GRAD models will be removed to focus on the novel RAMP model. This will make the manuscript more concise and highlight the novelities compared to the Legrain et al. (2023) model

In the following, we respond to your comments and propose several changes to the manuscript, motivated by your suggestions.

Author comments
Referee comments

**General comments**

**Conceptual model**. The model used in this study originates from the Paillard conceptual model of glacial cycles (Paillard, 1998). This model was then considerably revised by Parrenin and Paillard (2012) (hereafter referred to as PP12). Firstly, the authors of this paper abandoned Milankovitch theory of glacial cycles and delegated the determination of the 'orbital forcing' to the optimization algorithm. Secondly, they introduced two completely different 'orbital forcings': one determines the linear response of the global volume while another one determines nonlinear response, namely transitions between glacial and deglaciation regimes. However, since they tune the model only to glacial cycles of the late Quaternary, both 'orbital forcings' at least qualitatively resembles each other and boreal summer insolation. Unfortunately, the authors did not explain why they need two orbital forcings instead of one. In Legrain et al. (2023) (hereafter referred to as L23), the same model has been applied already to the entire Quaternary, and the model parameters were fitted to Berends global ice volume simulations. Since Berends ice volume contains very little precession even for the late Quaternary, the result of such tuning is easy to understand. The 'linear orbital frorcing' ($I_\alpha$) is dominated by obliquity while 'nonlinear orbital forcing' ($I_k$) by presession. (How such a situation could occur in the real world is not explained). Without strong precessional component in Ik, it would not be possible to simulate glacial cycles of the 100-kyr world, but too much precession disturbs the 41-kyr world. This is why, the optimization algorithm chooses model parameters for the early Quaternary in such a way

that another central element of Pallard's conceptual model - the existence of two distinct glaciation and deglaciation regimes - also vanishes. As can be seen in Fig. 2, the model switches several times between glaciation and deglaciation regimes during a single glacial cycle. As the results, what is described in the paper is not a conceptual model but rather a mathematical immolator of glacial cycles, where a good agreements with 'reconstructions' is achieved by using 16, 17 or even 19 model parameters, the physical meaning of which is hard to understand.

We accept the critique of relying on too many model parameters and on two different orbital forcings. Hence, we set up a reduced version of the RAMP model (described below), which allows us to show the effects of a ramp-like change in the deglaciation parameter.
The necessity of having $I_\alpha$ and $I_k$ arises from using non-normalised units and dimensions for the target. This is discussed in more detail below (see comments on Model performance). However, for the reduced model, we circumvent this issue by scaling the orbital forcing to the ice volume, which allows us to rely on a single insolation metric for the model.

**Model description**. Although the model used in the study has been described in several papers, some aspects require clarifications and corrections. Firstly, it is unclear whether conditions for regime changes denoted by (i) and (ii) should be met simultaneously. According to PP12 it seems that both conditions must be met :

$$\text{g to d: } k_{\text{Esi}}\text{Esi}(t) + k_{\text{Eco}}\text{Eco}(t) + k_{\text{Ob}}\text{Ob}(t) + v \geq v_0$$
$$(\text{and } k_{\text{Esi}}\text{Esi}(t) + k_{\text{Eco}}\text{Eco}(t) + k_{\text{Ob}}\text{Ob}(t) \geq v_1)$$

but it is unclear to me why the second condition is in brackets.

In this model, as well as for the PP12 and L23 models, both conditions must be met simultaneously. We agree that writing the second condition in brackets is misleading. However, in this manuscript, it is clearly stated that both conditions must be met simultaneously (see Eq. 6 and 7). Having a look into the source code of the Leloup and Paillard (2022) model reveals that both conditions must be fulfilled there as well, although not being mentioned in their model description (their Eq. 2).

Secondly, the model employs two different times denoted by the same letter 't'! One time, used in the differential equation for ice volume has 'physical' (i.e. normal) direction, while the second time, which determines the evolution of model parameters, goes in the 'paleo' (i.e. reverse) direction. This is very confusing. Using of the same notations ($v_0$ and $\tau_d$) for two different characteristics (one constant and another time-dependent) is also not a good idea.

We agree. We followed the convention used in Legrain et al. (2023), but we agree that this can lead to misunderstandings. Therefore, we propose to use the following formulation for the RAMP model:

$$v_0(t) = \begin{cases} v_{0,1}, & \text{if } t > t_1 \\ v_{0,1} - \frac{v_{0,2} - v_{0,1}}{t_1 - t_2}\left(t - t_1\right), & \text{if } t_1 \geq t \geq t_2 \\ v_{0,2}, & \text{if } t_2 > t \end{cases} \tag{1}$$

where $v_{0,1}$ denotes the constant value prior to the onset of the ramp at time $t_1$ and $v_{0,2}$ the constant value after the ramp at time $t_2$. The time $t$ is given in paleo units (kyr BP).

**Improved model**. The title of the paper begins from "*An improved conceptual model*", but, surprisingly, it is not clear from the manuscript what they meant under "improved model". The manuscript makes an impression that the 'improved model' is RAMP-l but in page 27 it is written *'we constructed an improved conceptual model of Quaternary global ice volume, including four different internal forcing scenarios, alongside another model configuration... We showed that the RAMP-l model...'* I am not sure how this should be interpreted. The improvements compared to L23 are also not clearly described. Only after the comparison of the equations in Pollak and L23, one can understand that 'deprecated truncation function' means that the authors decided not to use truncation of insolation introduced in Paillard (1998) and then uses in PP12 and L23. Why the authors consider this an improvement is not explained.

In the manuscript, we sometimes refer to the ORB, ABR, GRAD and RAMP models as different scenarios of one model, and sometimes we refer to them as individual models. We agree that this can lead to misunderstandings. In the revised manuscript, we will only present one model, which will eliminate this misunderstanding. By improved model, we refer to improvements in each of the individual models that were already performed in Legrain et al. (2023) (ORB, ABR and GRAD), i.e. the ORB/ABR/GRAD model in this study outperforms the ORB/ABR/GRAD model in Legrain et al. (2023). Furthermore, the RAMP and RAMP-l models yielded better results than the GRAD model.
One major improvement compared to the Legrain et al. model is of technical nature regarding the improved tuning strategy and the improved speed of the code. However, this was barely mentioned in the manuscript, since we wanted to focus on the model output rather than on these technical details. The model speed was significantly increased by translating the code into JAX (a python library which allows for accelerated computation and just-in-time compilation). Furthermore, we added three additional solvers, namely ptemcee (parallel-tempered MCMC), dynesty (dynamic nested sampling) and PyMC. With the improved tuning procedure and with the increase in speed, which allows a more rigorous tuning, we could already reduce the model data mismatch compared to the original Legrain et al. model, without any changes in the parameterisations. We propose to add a paragraph in the model section of the revised version of the manuscript, where we highlight the increase in speed and the newly added solver to the model.
The increase in speed and the refined tuning allowed us to test various parameterisations in the model and come up with a new version of the RAMP model with reduced complexity. For the revised version of the manuscript, we propose to only present this reduced RAMP model (described below), which relies on fewer model parameters and uses only one orbital forcing. The other models will be dropped in the revised manuscript. By doing so, we can clearly indicate the changes in parameterisation compared to the Legrain et al. (2023) model and discuss its improvements.

At the same time, the author introduced a new time-dependent parameter td which is the relaxation time scale during deglaciation and which is in Paillard (1998) model determined the duration of glacial termination. However, what is the meaning of this parameter in Pollak et al. is unclear: during the early Quaternary its value in GRAD, RAMP and RAMP-l models is about 40 kyr, i.e. close to the duration of the entire glacial cycles. Even worse, in ABR model version, the relaxation time scale is -113 kyr, but the 'relaxation time scale' cannot be negative by definition. The meaning of negative v0 in the RAMP model is equally hard to interpret.

$\tau_d$ cannot be directly compared to values obtained in the P98 or the Leloup and Paillard (2022) model, since the constant $\alpha_d$ was added in the PP12 model. Therefore, it always has to be considered with the corresponding value of $\alpha_d$. The initial idea in the P98 model was that once the model enters a deglaciation state, the term $\frac{v}{\tau_d}$ can quickly decay large ice volumes. This only works for positive values of $\tau_d$. Therefore, it is in fact questionable why a negative value of -113 kyr appears in the ABR model. The negative sign of $\tau_d$ makes the whole $-\frac{v}{\tau_d}$ term positive, which reduces the rate of change in $\left(\frac{\mathrm{d}v}{\mathrm{d}t}\right)_d \left(= -I + \alpha_d - \frac{v}{\tau_d}\right)$. However, $\alpha_d$ is highly negative for the ABR model ($\alpha_d = -0.81$), compensating the positive $-\frac{v}{\tau_d}$ term and still leading to a fast deglacial decay. To avoid this behaviour in the model and improve the physical interpretability of the $\tau_d$ parameter, we will remove the $\alpha_d$ parameter in the reduced model version. Furthermore, we will remove the time dependency of $\tau_d$ to focus on the change in the deglaciation parameter and reduce the number of parameters. Since we are using a non-normalised ice volume reconstruction, negative values for the ice volume appear during the early Quaternary and prior to ∼2.8 Ma, before the onset of NH glaciations. In the Berends SL curve, the values are reported with respect to present-day conditions. This can result in negative values for $v_0(t)$ during early time periods.

**Insolation**. According to the Milankovitch theory, changes in boreal summer insolation is the driver of glacial cycles, but since PP12 abandoned Milankovitch theory, it is rather surprising that the term 'insolation' appears in section 3.5. The term 'insolation' has a very clear meaning: 'insolation' is the abbreviation for 'incoming solar radiation' and is measured in W/m$^2$ (or equivalent units). The 'orbital forcings' $I_\alpha$ and $I_k$ used in the paper have nothing to do with the real insolation and therefore the terms 'insolation' and 'insolation maximum' in the context of the paper is misleading.

We agree. We will instead use the term *orbital forcing* in the revised manuscript

**Ice volume reconstruction**. The author used the Berends reconstruction of global ice volume. Like any other reconstruction, the Berends reconstruction contains significant uncertainties and deficiencies. For example, for the Last Glacial Maximum, for which there are numerous independent data, Berends underestimates global ice volume by more than 20% and also underestimates ice volume variability during MIS5. For MIS3 different reconstructions for global ice volume range between 30 and 90 meters (Farmer et al., 2023 PNAS), i.e. the uncertainties are about 50%. It is very likely that for the early times, the uncertainties are even larger. This makes reported improvements in RMSE order of several meters completely insignificant.

We agree. We will follow the advice from all three referees and no longer use the Berends or Rohling sea level curves as our main tuning targets in the revised manuscript, but only show them in the SI. To discuss the limitations of these model-based sea level reconstructions, we

[Figure]

Figure 1: Reduced RAMP model tuned on three different tergets: a) Berends et al. (2021) sea level curve, b) Clark et al. (2025) benthic $\delta^{18}O_b$ (probstack with trend removed) and c) Clark et al. (2025) sea water $\delta^{18}O_{sw}$.

propose to add the following paragraph to our discussion:

"Available global mean sea level data exhibit large uncertainties. Between 50 and 30 ka, geological and geochemical reconstructions of GMSL vary up to 60 m and above (Farmer et al., 2023). Model-based deconvolutions of global $\delta^{18}O$ into GMSL, like for the Berends et al. (2021) and Rohling et al. (2022) sea level reconstructions, exhibit similar large uncertainties. While the Berends curve shows an LGM lowstand of around 100 m, the Rohling curve gives around 108 m. Both values are well below observational-based reconstructions of around 130 - 135 m for the LGM (Austermann et al., 2013; Lambeck et al., 2014; Yokoyama et al., 2000). These large uncertainties limit the usability of sea level reconstructions as targets for conceptual models, as presented in this study."

Instead of using the Berends sea level curve, we will follow the advice from Referee 1 and use the benthic $\delta^{18}O$ record (probstack with trend removed) from Clark et al. (2025) as our main target. Furthermore, we will discuss the model performance on the newly available seawater $\delta^{18}O_{sw}$ record from Clark et al. (2025), based on a benthic $\delta^{18}O$ deconvolution motivated by an ocean temperature data compilation. It gives a contrasting view compared to the Berends and Rohling deconvolutions by exhibiting a rise in $\delta^{18}O_{sw}$ during the MPT, interrupting a previous period of increasing values and showing larger glacial-interglacial amplitudes in the pre-MPT cycles. However, a final reconstruction of this $\delta^{18}O_{sw}$ record

into sea level is not yet available. That's why we will mainly focus on the benthic record. Following your advice, we reduce the complexity of the RAMP model (discussed later). The results of this reduced RAMP model for the three different tuning targets (Berends SL, Clark $\delta^{18}O_b$, Clark $\delta^{18}O_{sw}$) can be seen in Fig. 1. Our reduced RAMP model yields similar results for this target, i.e. a long-lasting trend for the ramp, which started over 2 Ma. While the model performs well on this target, it has more difficulties with the seawater $\delta^{18}O$ record from Clark et al. (2025) (Fig. 1c), as can be seen by comparing the R values. In general, the model cannot accurately reproduce the larger pre-MPT amplitudes of the glacial-interglacial cycles and the decreasing trend of $\delta^{18}O_{sw}$ during the MPT. The RAMP formulation seems to be less suited for this target, as can be seen by the identified ramp period, which lies now in the interval $\sim$ 400 - 200 ka. For the $\delta^{18}O_{sw}$ target, the model also cannot reproduce the shift towards the 100-kyr periodicity correctly, since it only shows a 100-kyr signal for the last 200 ka.

Even more strange to see 6, 7 and even 8-digit numbers in Table A1. It is clear that so many digits originate from Monte Carlo, but in natural sciences, it is customary to report only significant digits. The paper also says nothing about the robustness of the modelling results with respect to the choice of model parameters.

We are well aware that it is not conventional to use so many digits in natural science. The reason behind this was to report the exact parameter values that were used in the model, since the model output can be sensitive to specific parameters. Hence, in order to allow interested readers who want to re-run the simulations, it is important to report the exact set of parameters used. However, since they are also documented in the available GitHub repository, we accept this critique and will change the number of significant digits in this Table to enhance readability.

**Simulation of MPT**. The main objective of the paper (similarly to L23) is to simulate MPT. Clearly, MPT cannot be explained by orbital forcing alone; therefore, the purpose of the ORB version is unclear.

One reason to include the ORB model, similarly including the ABR and GRAD models, was to show the effects of the new tuning strategy in combination with changes to the initial parameterisation as described in Legrain et al. (2023). The improved tuning strategy alone could already reduce the model-data mismatch (not shown in the manuscript). The other reason for including the ORB model was to have a 'baseline' model, which can be used to compare the RAMP model with and to see the effects of adding a ramp-like change to the model.

Actually, there are recent studies linking the occurrence of the MPT to orbital forcing alone. Ma et al. (2024) introduce the *integral of annual mean insolation anomaly* (IAMIA), which quantifies successive small step-wise insolation changes over a given period of time. The authors show that IAMIA exhibits a large shift around 935 ka, which they hypothesise to have enabled the onset of the MPT. Another recent preprint by Verbitsky and Omta (2025) discusses the idea that the MPT is due to a delayed relaxation process, which can lead to an abrupt-like jump in the dominant period. This shift in periodicity can be highly sensitive to the initial conditions, and the sensitivity depends on the amplitude of the orbital forcing.

However, since our model yields the same result as in Legrain et al. (2023), namely that orbital forcing alone cannot explain the MPT, we will, to keep the manuscript concise, drop the ORB model from the revised manuscript and just refer to the Legrain et al. (2023) paper.

Secondly, even without modelling, it is clear from data analysis alone that the MPT (Mid-Pleistocene Transition) was a transition, not an event, and that it lasted for at least several hundred thousand years. This is why the purpose of the repeating of ABR scenarios is unclear. Regarding the mechanism(s) of the MPT (which are still debatable), it is unclear how the experiments presented in this manuscript can shed light on the cause of the MPT transition. The problem with strongly nonlinear systems, such as the Earth system, is that even gradual changes of the controlling parameters can cause a rather abrupt regime changes (e.g. Willeit et al., 2019).

To highlight the novel RAMP model and the differences compared to the Legrain et al. (2023) paper, we will remove the ABR and GRAD models from the revised manuscript. Indeed, these conceptual models cannot shed light on the underlying physical mechanisms directly (e.g. whether a long-term trend in $v_0$ is due to regolith removal or a gradual $CO_2$ decrease, etc.), since they do not include the involved physical or chemical processes directly. However, their strength lies in their capability to investigate the underlying temporal structure of such a change. While the ORB, ABR and GRAD models all rely on a specific scenario, the RAMP model selects the most likely scenario to reconstruct the observed paleo record, i.e. the benthic $\delta^{18}O$ record from Clark et al. (2025). Multiple scenarios are possible in the RAMP model:

- $v_{0,1} = v_{0,2}$ (or almost; Eq. 1 above): (almost) no change in the deglac. threshold $\rightarrow$ (almost) purely orbitally forced $\widehat{=}$ ORB model

- $t_1 = t_2$ (or almost): abrupt-like scenario $\widehat{=}$ ABR model

- $t_1 = 2.6$ Ma and $t_2 = 0$ Ma (or almost): gradual scenario over the entire Quaternary $\widehat{=}$ GRAD model

- a scenario in between with a gradual change limited over a specific time period

Hence, the RAMP model already incorporates the ORB, ABR and GRAD scenarios and selects the one which can best reproduce the target record (another reason why we agree to drop these models from the revised manuscript). This yields interesting information about how such a temporal change could have looked like.
So, actually, the outcome of our RAMP model is that it supports the idea of a long-lasting change that started already very early in the Quaternary, more than 2 Ma, instead of a rather short and limited change (e.g. which only lasted during the MPT).
An alternative approach is presented in Ganopolski (2024), where the author prescribes the change in the critical ice volume parameter $v_c$ (similar to our $v_0$) by relating it to the regolith-free scenario described in Willeit et al. (2019). This shows that the regolith-free scenario can reproduce the MPT, but since it relies on a single prescribed temporal structure for this change in $v_c$, it gives no information on how likely such a temporal structure is.

It is quite possible that the gradual decline in CO2 and landscape evolution (including regolith removal) began not only well before the MPT but also before the Quaternary, and

there is no way to derive this from the experiments described in the manuscript.

This is exactly the strength of the RAMP model, since it gives information about a potential start and end of such a gradual change. The L23 GRAD model was only run for the past 2 Ma, therefore, it does not allow earlier changes in $v_0$, but our RAMP model reveals that the change in $v_0$ likely started even before 2 Ma. In contrast, other models either use a gradual trend over the entire simulation period (e.g. GRAD model, P98 model, Leloup model) or use one specific temporal structure (e.g. Model 3 in Ganopolski (2024)). While these models give no information about when such a gradual trend could have started and ended, the RAMP model does.

**The role of precession**. Firstly, the model used in the study of Pollak cannot be used to study the role of precession and obliquity. This is not a physically-based model and the model parameters are just chosen by the optimization algorithm in such a way that glacial termination of late Quaternary occur in the right times. And this 'right times' coincide with the periods of rising boreal summer insolation. In turn, boreal summer insolation is dominated by precession. This is why the optimization algorithm picked up precession.

The analysis of the RAMP-l model will be removed in the revised manuscript, since we agree to reduce the model complexity and we will only focus on the main parameters to highlight the results coming from the new ramp-like formulation.
While we cannot directly investigate the physical effects of precession and obliquity in the climate system with our model, we can investigate their influence in the dynamics of our model. By using a linear combination of precession and obliquity rather than a single insolation metric, we can explicitly see which roles these two orbital quantities play in the model. In the reduced RAMP model, we will only rely on a single orbital forcing (instead of having $I_\alpha$ and $I_k$), which e.g. allows us to investigate the spectral power of the obtained orbital forcing in this model (Fig. 3a).

Secondly, the fact that glacial terminations of the late Quaternary are mainly determined by precession has been known well before Barker et al. (2025). Already Raymo (1997) noted that 'the length between subsequent terminations is either four or five precessional cycles long'. This idea was further developed by Ridgewell et al. (1999) who wrote that 'the spectral signature of $\delta^{18}$O records are entirely consistent with Milankovitch mechanisms in which deglaciations are triggered every fourth or fifth precessional cycle'

Barker et al. (2025) presents the most recent analysis on the role of precession and obliquity for glacial cycles in the 100 kyr and 41 kyr world. Therefore, as Referee 1 also points out, a special focus should be on this paper.

**Future glacial cycle simulations**. The authors also used their models to simulate future glacial cycles. According to these simulations, the next glacial cycle has either already begun or will begin soon in the absence of anthropogenic influence. The authors are, of course, aware that this contradicts to the results of the physically-based models and is therefore likely to be incorrect. This is why they attempted to defend their model by arguing that in Ganopolski et al. (2016) glacial inception occurs with the current orbital forcing

if pre-industrial $CO_2$ were 240 ppm. This is absolutely correct, but among many uncertainties, there is one thing which we know for sure – the preindustrial concentration was 280 ppm, and this value is typical for post-MBT interglacials. Therefore, the unprecedently long Holocene is the robust and most striking feature of the next 100 kyr. Since this feature is not reproduced by Pollak et al., the value of such modelling exercises is called into serious doubt.

While neglected in most other conceptual models, we believe that extrapolating at least the next glacial cycle reveals important model dynamics. The extrapolated curves do not resemble nature, as they lack anthropogenic $CO_2$ emissions. Therefore, they must be interpreted as purely academic thought experiments. More complex models, coupled to the carbon cycle, project, even without any anthropogenic influence, an unprecedentedly long interglacial, lasting for another 50 kyr (Ganopolski et al., 2016; Talento and Ganopolski, 2021). In contrast to this, all of our models, no matter of which simulation period, project that the next glacial cycle has just started or is about to start. In their recent study, investigating the influence of precession and obliquity on glacial interglacial cycles, Barker et al. (2025) estimate that the current interglacial conditions would last for 11 kyr, when obliquity reaches its next minimum and the succeeding glacial would be interrupted in around 66 kyr (again neglecting anthropogenic effects). Other conceptual models similarly predict a contrasting view for the next glaciation: Calder (1974) projected a start 5 ka and an end in around 119 kyr (see his Fig. 2), Imbrie and Imbrie (1980) estimated a start 6 ka (see their Fig. 7) and Fig. 5 in Paillard (2015) reveals that only a large enough glaciation threshold can lead to a prolonged Holocene in the Paillard (1998) model, while a lower threshold would have terminated the Holocene already a few kyr ago. While other models like the L23, Model 3 (Ganopolski, 2024) and Leloup and Paillard (2022) were not extrapolated for the next glacial cycle, it can be expected that they would yield similar results (i.e. a short Holocene) compared to our model, since they are based on similar model dynamics. Therefore, we believe that the failure in simulating a prolonged Holocene in our model, as well as in other conceptual models or studies, is a striking feature, which is worth mentioning, and might be due to the missing carbon feedback in the model or a glaciation threshold which only depends on orbital forcing or parameters.

**Model performance**. The authors compared their version of the PP12 model with the L23 model, as well as the 19-parameters RAMP-l with the 16-parameters RAMP and using BIC, they claimed significant improvements. However, the MiM (minimal model which I reported in my 2024 paper), which is a simplification of the Leloup and Paillard (2022) model (which in turn is a simplification of original Paillard 1998 model) simulates the last 800 kyr (arguably the most interesting and difficult period of climate history) as good as RAMP (R2=0.73). But MiM has only 4 parameters (nondimensional version of MiM described in Ganopolski 2024 has three parameters). By contrast, RAMP uses 12 parameters to simulate the late Quaternary. The natural question is for what purpose we need models with so many parameters and what can we learn from them.

The MiM model and Model 3 described in Ganopolski (2024) present impressively how few parameters are needed to simulate the 100-kyr world or the entire Quaternary. These models were specifically designed to show how few parameters are needed to achieve this goal. If these models indeed present the minimal models required to reconstruct the 100-kyr world or

the Quaternary, any other conceptual model which tries to investigate some specific feature during the Quaternary will need to add further complexity to the model. While not all extra parameters are needed, we highlight the main differences in the following:

- MiM and Model 3 are normalised, while the RAMP model has units of global ice volume. Due to the normalisation, the MiM model does not need a $v_0$ and $v_1$ parameter, but can simply apply state changes whenever $v = 0$ or $v = 1$. Such a threshold also leads to unrealistic 'flat' interglacials, which always result in full deglaciations (see Fig 3, 10 in Ganopolski (2024)). Such a model cannot reproduce shifts in interglacial conditions (e.g. interglacial values of $\delta^{18}O_{sw}$ in Clark et al. (2024) differ significantly).

- MiM model uses a single metric for the orbital forcing, while RAMP depends on a linear combination of precession, co-precession and obliquity. It can be expected that the RAMP model would work equally well when using normalised summer solstice insolation at 65° N (or an equivalent measure) as an input for the orbital forcing. However, as Leloup and Paillard (2022) have shown, the model-data agreement depends on the selected forcing and can pose a bias to the results, since the different orbital metrics have different weights of precession and obliquity. Hence, we explicitly do not want to use a single metric in our model to remove any source of bias to the model. Instead, we use the same approach as in Imbrie et al. (2011) and Legrain et al. (2023), where orbital forcing is obtained as a linear combination of precession, co-precession and obliquity. This adds three extra parameters to the RAMP model ($\alpha_{\mathrm{Esi}}, \alpha_{\mathrm{Eco}}, \alpha_{\mathrm{O}}$)

- In a normalised and dimensionless model, there is no need for $I_\alpha$ and $I_k$, but just one forcing would be sufficient. First off, $I_\alpha$ (m/kyr) and $v$ (m) have different units, therefore, the threshold $I_\alpha + v$ is not possible. Secondly, $I_\alpha$ has values between -2 and 2 in our RAMP model, which is much smaller compared to $v$, which has values between 0 and 120. Hence, the influence of the orbital forcing in the threshold $I_\alpha + v$ would be negligible. Such an issue does not occur in normalised units, as can be seen in the Leloup and Paillard (2022) model. Here, they use the same threshold $I + v \geq v_0$, but can rely on a single orbital forcing $I(t)$. To circumvent this issue, the k-parameters were originally introduced in the Parrenin and Paillard (2012) model. This adds three extra parameters to the RAMP model: $k_{\mathrm{Esi}}, k_{\mathrm{Eco}}, k_{\mathrm{O}}$

- The objective of the RAMP model is to investigate the temporal structure of an internal change in the system causing the MPT. Hence, we explicitly want to have freely adjustable time points $t_1$ and $t_2$ in the model, as well as $v_{0,0}$ and $v_{0,1}$. In contrast, Model 3 prescribes the temporal structure of its change in $v_c$, by using the regolith-free scenario from Willeit et al. (2019), which requires two parameters less (no $t_1$ or $t_2$), but therefore, it only tests one specific temporal structure.

- Note: the initial ice volume $v_i$ is also counted as a model parameter, since it has to be adjusted for different simulation periods and targets. If we neglect this parameter, as done in Ganopolski (2024), we always have one parameter less than reported in the manuscript

To reduce the complexity of the model (formulas shown below), we will remove the following parameters from the model, which we identified to be less important or that reduce the interpretability of the results:

- $\alpha_{\mathrm{Eco}}$: While we will still use a linear combination of precession and obliquity for our forcing, which allows the model to adjust its obliquity and precession signals on its own, we found that precession and co-precession are not needed both, but just one of them is sufficient to obtain this goal. Hence we will drop co-precession (Eco) and its related parameters $(\alpha_{\mathrm{Eco}}, k_{\mathrm{Eco}})$ from the reduced RAMP model

- $k_{\mathrm{Eco}}, k_{\mathrm{Esi}}, k_{\mathrm{O}}$: We will adjust our threshold $(I + v \geq v_0)$ in such a way that it is possible to rely on a single orbital forcing and that we no longer require these three parameters

- $\alpha_d$: free model parameter which was first introduced in the PP12 model. While it allows more flexibility in the model by adding a constant offset to $\left(\frac{\mathrm{d}v}{\mathrm{d}t}\right)_d$ it does not alter the general findings of the RAMP model. It also reduces the physical interpretability of $\tau_d$, since it always needs to be considered in combination with $\alpha_d$. Therefore, it will be removed

- $\tau_d'$: to focus on the change in the deglaciation parameter $v_0(t)$ and to show how a ramp-like change of it can affect the simulation output, we will remove the time dependency in $\tau_d$ and use it as a constant parameter again

The objective of the RAMP model was not to overfit the model to some target record by successively adding more parameters. Instead, the objective was to use a ramp-like change in $v_0$ to obtain information on how the temporal structure of a gradual change could have looked like and, doing so, to reduce the model data mismatch, compared to the L23 model. To accept the critique of relying on too many model parameters and to show that the ramp formulation yields similar results, we set up a reduced RAMP model (shown below). This reduced RAMP model uses only 10 parameters (9 if the initial ice volume is not counted). Hence, it has 4 parameters less than the GRAD model of L23, 6 less than the original RAMP model and 9 less than the RAMP-l model.

**Reduced RAMP model:**

- Orbital forcing:
$$I(t) = \alpha_{\mathrm{Esi}}\mathrm{Esi} + \alpha_{\mathrm{O}}\mathrm{Ob} \tag{2}$$

- Differential equations:

$$\mathrm{g} : \frac{\mathrm{d}v(t)}{\mathrm{d}t} = -I(t) + \alpha_g, \tag{3}$$

$$\mathrm{d} : \frac{\mathrm{d}v(t)}{\mathrm{d}t} = -I(t) - \frac{v(t)}{\tau_{\mathrm{d}}}, \tag{4}$$

- State transition **(g)** $\rightarrow$ **(d)**:

$$\text{(i)} \quad v(t) \cdot \tilde{I}(t) + v(t) > v_0(t) \qquad \& \qquad \text{(ii)} \quad v(t) \cdot \tilde{I}(t) > v_1, \tag{5}$$

- State transition **(d)** $\rightarrow$ **(g)**:

$$\text{(i)} \quad v(t) \cdot \tilde{I}(t) + v(t) < v_0(t) \qquad \& \qquad \text{(ii)} \quad v(t) \cdot \tilde{I}(t) < v_1, \tag{6}$$

- Dimensionless orbital forcing: $\tilde{I}(t) = \frac{I(t)}{[I]}$

- Ramp-like change in $v_0$:

$$v_0(t) = \begin{cases} v_{0,1}, & \text{if } t > t_1 \\ v_{0,1} + \frac{v_{0,2}-v_{0,1}}{t_2-t_1}(t-t_1), & \text{if } t_1 \geq t \geq t_2 \\ v_{0,2}, & \text{if } t_2 > t \end{cases} \tag{7}$$

- 9 model parameters: $\alpha_{\text{Esi}}, \alpha_{\text{O}}, \alpha_{\text{g}}, \tau_d, v_{0,1}, v_{0,2}, t_1, t_2, v_1$

- Note on the changed threshold: as mentioned above, $I + v > v_0$ does not work for a non-normalised model as presented here. To circumvent this, we firstly use the dimensionless orbital forcing $\tilde{I}$ for the threshold. Secondly, we have to scale the orbital forcing, s.t. it has a similar order of magnitude as the ice volume $v(t)$. Hence, we multiply $\tilde{I}(t)$ with v. Now, both terms have same units and order of magnitude. The term $v \cdot \tilde{I}$ can also be interpreted as an ice volume dependent orbital forcing, effectively making large ice sheets more sensitive to orbital forcing

- We propose including the above model description and the discussion about which model parameters to drop from the new model and why in the revised manuscript. This will highlight the changes made with respect to the L23 model and underline the novelities

**Specific comments**

L10. The meaning of '*internal focing*' is unclear. The authors prescribed temporal evolution of several model parameters, not forcing.

Will be corrected to '... a ramp-like change in the temporal evolution of the deglaciation parameter.'

L.12 '*support the idea of a long-term climatic shift as a cause of the MPT*'. In fact, MPT is the climate shift, whereas the cause(s) of MPT may have nothing to do with climate, like the 'regolith hypothesis'.
Will be corrected to: '... long-term shift in the underlying Earth-climate system as a cause of the MPT'.

L.131. Unclear what the authors meant under 'versatile model' since even a slight change in the temporal scenario (GRAD to RAMP) causes significant changes of all other model parameters.

The term 'versatile' refers to the easy-adjustability of the model. We propose to replace this sentence with the following:
"The new RAMP model is characterised by its easy adjustability. In comparison with the L23 model (Legrain et al., 2023), it can be run on different time scales, be extrapolated into the future, be run on different tuning targets (Berends sea-level, Rohling sea-level, Clark benthic $\delta^{18}O_b$, Clark seawater $\delta^{18}O_{sw}$) or climatic variables (sea level, $\delta^{18}O$), and it can be easily re-tuned for new parameterizations (e.g. allowing to test the effects of changes in the

state thresholds, model forcing, etc.)."

L. 147. '*a linear combination of three orbital parameters normalized to zero mean and unit variance that can reproduce the insolation at most latitudes and seasons*'. This is incorrect. An arbitrary combination of these three parameters does not reproduce insolation at any latitude and at any time.

This approach is not new and has been used in previous models (Imbrie et al., 2011; Parrenin and Paillard, 2012; Legrain et al., 2023). The emphasis in the sentence is on *most*, it does not state *any*.
Imbrie and Imbrie (1980) explain this approach: "Combinations of orbital functions $\epsilon$ (obliquity), $e \sin \omega$ (precession) and $e \cos \omega$ (phase-shifted precession) are used as a forcing for our model, and are taken from (Laskar et al., 2004). Insolation at most latitudes and seasons can be represented quite accurately by a combination of these three orbital functions." (Imbrie et al., 2011, p. 2)
Furthermore, they discuss: "Our model uses orbital forcing which is a linear combination of $\epsilon$ (obliquity), $e \sin \omega$ (precession) and $e \cos \omega$ (phase-shifted precession). The choice of which linear combination to use is data-driven, in that a regression analysis is used to determine which combination best explains $y'$, the rate of change in ice volume. In our view, this is preferable to working with a preconceived notion of which latitude and season is the most suitable for use in the forcing function." (Imbrie et al., 2011, p. 16).
We fully agree with Imbrie et al. (2011) that not relying on a specific orbital forcing is preferable. To demonstrate that even two orbital parameters (precession parameter and obliquity) can almost perfectly reconstruct various Insolation curves, we demonstrate this for five exemplary insolation curves (see Fig. 2). Insolation ($I$), precession (Esi) and obliquity (Ob) values are taken from Laskar et al. (2004) and were normalised. ISI computation was carried out with code from Leloup and Paillard (2022). $I_{aprox} = \alpha_{Esi}\text{Esi} + \alpha_{O}\text{Ob}$ is the linear combination. We find the following solutions for these five orbital forcings:

- $I_{SS}^{65N}$: Insolation at 65° N on 21$^{\text{st}}$ June: $\alpha_{\text{Esi}} = -0.917$, $\alpha_{\text{O}} = 0.393$

- $I_{WS}^{65N}$: Insolation at 65° N on 21$^{\text{st}}$ December: $\alpha_{\text{Esi}} = 0.093$, $\alpha_{\text{O}} = -0.991$

- $I_{SS}^{Eq}$: Insolation at the Equator on 21$^{\text{st}}$ June: $\alpha_{\text{Esi}} = -0.995$, $\alpha_{\text{O}} = -0.094$

- $I_{WS}^{90S}$: Insolation at 90° S on 21$^{\text{st}}$ December: $\alpha_{\text{Esi}} = 0.892$, $\alpha_{\text{O}} = 0.451$

- ISI($\tau = 300$)$_{65N}$: ISI above 300 W/m$^2$ at 65° N: $\alpha_{\text{Esi}} = -0.523$, $\alpha_{\text{O}} = 0.850$

L. 168. '*the model can also incorporate an internal forcing mechanism to account for non-linear feedback mechanisms within the climate system*'. Which forcing and feedbacks are meant here is unclear.

We will no longer use the misleading term '*internal forcing*' or '*internal forcing scenarios*', but rather speak of '*temporal change in the deglaciation parameter*'.

Table 3. Why number of model parameters in Table 3 does not coincide with Table A1.

[Figure]

Figure 2: Various Insolation curves from Laskar et al. (2004). ISI computation was carried out with code from Leloup and Paillard (2022). $I_{approx}$ is their optimal linear approximation, based on a linear combination of precession (Esi) and obliquity (Ob): $I_{approx} = \alpha_{\text{Esi}}\text{Esi} + \alpha_{\text{O}}\text{Ob}$.

*Perhaps I am overlooking something, but they are the same? ORB has 12, ABR has 15, GRAD has 14, RAMP has 16 and RAMP-l has 19 parameters.*

L. 460. 'each or every second insolation peak resulted in a deglaciation'. What is meant under insolation peak is unclear.

*Incorrect naming of insolation will be replaced by 'orbital forcing'.*

L. 502 'integrated summer insolation (ISI) reported in the literature'. Firstly, this is not an appropriate way to refer to the published concept. ISI was introduced by Huybers in his 2006 Science paper. Secondly, the authors should explain how ISI is defined.

*Accepted. We propose to replace the sentence by:*
*"The integrated summer insolation (ISI) was introduced by Huybers (2006) and is defined as the sum of insolations on days exceeding some threshold $\tau$:*

$$\text{ISI}(\tau) = 86,400 \cdot \sum_i \beta_i W_i, \tag{8}$$

*where $W_i$ is the mean insolation in $W/m^2$ on day i, and $\beta_i = 1$ if $W_i \geq \tau$ and 0 otherwise."*

Thirdly, ISI was introduced by Huybers to support his (former) idea that glacial cycles are paced primary by obliquity; this is why ISI is defined such a way that it is completely dominated by obliquity. But since $I_\alpha$, is also dominated by obliquity it is not surprising that two obliquity-dominated curves resemble each other. What can be learned from such a comparison?

*Since the model uses a linear combination of three orbital parameters, it is not obvious by looking at these three parameters $\alpha_{\text{Esi}}, \alpha_{\text{Eco}}, \alpha_{\text{O}}$, how the associated orbital forcing looks like. Hence, we think that it is of interest to show how this forcing, which drives the dynamics of our model, looks like. In addition, conceptual models use a variety of orbital forcings, e.g. Model 3 uses the maximum summer insolation at 65° N, P98 relies on the same metric, but with an artificial truncation function added, Tzedakis et al. (2017) uses the caloric summer insolation at 65° N (as introduced by Milankovitch (1941)). Regarding this variety of different orbital forcings in use, it is of interest to see which metric our model selects.*
*While the Berends SL curve is mainly dominated by obliquity and only shows a small precession signal, this does not imply that the orbital forcing $I(t)$ in the model shows the same pattern. Actually, it can reproduce the sea level by Berends (and thus the same spectral powers for $v(t)$), while being driven by an orbital forcing, which has a different power spectrum. For our reduced RAMP model (as described above), we see that $I(t)$ has a dominant peak in obliquity, but also a significant precessional signal (Fig. 3a). On the other hand, the Berends SL curve, which was used as the tuning target has almost no precession signal (Fig. 3). This shows that it is indeed interesting to investigate the spectral power of $I(t)$ in the model and that it cannot be directly inferred from the tuning target alone. Leloup and Paillard (2022) also showed that their model can reproduce the LR04 target based on four different orbital forcings which vary in their precession and obliquity signals.*

[Figure]

Figure 3: LombScargle periodograms of (a) the orbital forcing I(t) in the reduced RAMP model, (b) the modelled ice volume v(t) and (c) the tuning target, which was the Berends SL curve.

[Figure]

Figure 4: ISI above 300 W/m$^2$ (a) and its periodogram (c). ISI above 400 W/m$^2$ (c) and its periodogram (d). Calculations done with code from Leloup and Paillard (2022).

Furthermore, the spectral power of ISI($\tau$) depends on the chosen threshold $\tau$. While ISI above 300 W/m$^2$ is mainly dominated by Obliquity (Fig. 4c), ISI above 400 W/m$^2$ has almost equal contributions of precession and obliquity (Fig. 4d).

L. 565. '*This suggests the presence of non-linearities within the climate system that modify Earth's response to changes in the orbital forcing. This conclusion aligns with findings from other studies (Clark et al., 2006; Leloup and Paillard, 2022; Berends et al., 2021b)*'. I am just curious whether the authors are aware of the works of Weertman, MacAyeal, Oerlemans and others who discussed how nonlinearities modify the Earth's response to orbital forcing already during 1970s and early 1980s?

As for the Berends et al. (2021b) paper, which the authors cited ten times, it is merely a review paper. Although review papers are useful reference material, especially for early career scientists, citing them cannot substitute for reading and citing real scientific papers.

We are aware of earlier work on this topic, e.g. (Hays et al., 1976, p. 1131) already stated that "Unlike the correlations between climate and the higher-frequency orbital variations [...] an explanation of the correlation between climate and eccentricity probably requires an assumption of non-linearity." We agree that review papers should not serve as the sole source of information and the original work therein should be cited as well. In the manuscript, the Berends et al. (2021b) paper was only cited twice as a single source of information, otherwise always multiple citations were given with at least two or three cited papers. In most of these citations, the Berends (2021b) paper is cited as another source of information, since it simply touches a broad range of topics. With over 50 different papers cited in this manuscript, it clearly does not rely on a single source of information.

**References**

J. Austermann, J. X. Mitrovica, K. Latychev, and G. A. Milne. Barbados-based estimate of ice volume at Last Glacial Maximum affected by subducted plate. *Nature Geoscience*, 6(7):553–557, July 2013. ISSN 1752-0908. doi: 10.1038/ngeo1859. URL `https://www.nature.com/articles/ngeo1859`. Publisher: Nature Publishing Group.

S. Barker, L. E. Lisiecki, G. Knorr, S. Nuber, and P. C. Tzedakis. Distinct roles for precession, obliquity, and eccentricity in Pleistocene 100-kyr glacial cycles. *Science*, 387(6737):eadp3491, Feb. 2025. doi: 10.1126/science.adp3491. URL `https://www.science.org/doi/10.1126/science.adp3491`. Publisher: American Association for the Advancement of Science.

C. J. Berends, B. de Boer, and R. S. W. van de Wal. Reconstructing the evolution of ice sheets, sea level, and atmospheric $CO_2$ during the past 3.6 million years. *Climate of the Past*, 17(1):361–377, Feb. 2021. ISSN 1814-9324. doi: 10.5194/cp-17-361-2021. URL `https://cp.copernicus.org/articles/17/361/2021/`. Publisher: Copernicus GmbH.

N. Calder. Arithmetic of ice ages. *Nature*, 252(5480):216–218, Nov. 1974. ISSN 1476-4687. doi: 10.1038/252216a0. URL `https://www.nature.com/articles/252216a0`. Publisher: Nature Publishing Group.

P. U. Clark, J. D. Shakun, Y. Rosenthal, P. Köhler, and P. J. Bartlein. Global and regional temperature change over the past 4.5 million years. *Science*, 383(6685):884–890, Feb. 2024. doi: 10.1126/science.adi1908. URL `https://www.science.org/doi/abs/10.1126/science.adi1908`. Publisher: American Association for the Advancement of Science.

P. U. Clark, J. D. Shakun, Y. Rosenthal, C. Zhu, P. J. Bartlein, J. M. Gregory, P. Köhler, Z. Liu, and D. P. Schrag. Mean ocean temperature change and decomposition of the benthic $^{18}$O record over the past 4.5 million years. *Climate of the Past*, 21(6):973–1000, June 2025. ISSN 1814-9324. doi: 10.5194/cp-21-973-2025. URL `https://cp.copernicus.org/articles/21/973/2025/`. Publisher: Copernicus GmbH.

J. R. Farmer, T. Pico, O. M. Underwood, R. Cleveland Stout, J. Granger, T. M. Cronin, F. Fripiat, A. Martínez-García, G. H. Haug, and D. M. Sigman. The Bering Strait was flooded 10,000 years before the Last Glacial Maximum. *Proceedings of the National Academy of Sciences*, 120(1):e2206742119, Jan. 2023. doi: 10.1073/pnas.2206742119. URL `https://www.pnas.org/doi/abs/10.1073/pnas.2206742119`. Publisher: Proceedings of the National Academy of Sciences.

A. Ganopolski. Toward generalized Milankovitch theory (GMT). *Climate of the Past*, 20(1):151–185, Jan. 2024. ISSN 1814-9324. doi: 10.5194/cp-20-151-2024. URL `https://cp.copernicus.org/articles/20/151/2024/`. Publisher: Copernicus GmbH.

A. Ganopolski, R. Winkelmann, and H. J. Schellnhuber. Critical insolation–CO2 relation for diagnosing past and future glacial inception. *Nature*, 529(7585):200–203, Jan. 2016. ISSN 1476-4687. doi: 10.1038/nature16494. URL `https://www.nature.com/articles/nature16494`. Publisher: Nature Publishing Group.

J. D. Hays, J. Imbrie, and N. J. Shackleton. Variations in the Earth's Orbit: Pacemaker of the Ice Ages. *Science*, 194(4270):1121–1132, 1976. ISSN 0036-8075. URL `https://www.jstor.org/stable/1743620`. Publisher: American Association for the Advancement of Science.

P. Huybers. Early Pleistocene Glacial Cycles and the Integrated Summer Insolation Forcing. *Science*, 313(5786):508–511, July 2006. doi: 10.1126/science.1125249. URL `https://www.science.org/doi/full/10.1126/science.1125249`. Publisher: American Association for the Advancement of Science.

J. Imbrie and J. Z. Imbrie. Modeling the Climatic Response to Orbital Variations. *Science*, 207 (4434):943–953, Feb. 1980. doi: 10.1126/science.207.4434.943. URL `https://www.science.org/doi/10.1126/science.207.4434.943`. Publisher: American Association for the Advancement of Science.

J. Z. Imbrie, A. Imbrie-Moore, and L. E. Lisiecki. A phase-space model for Pleistocene ice volume. *Earth and Planetary Science Letters*, 307(1-2):94–102, July 2011. ISSN 0012821X. doi: 10.1016/j.epsl.2011.04.018. URL `http://arxiv.org/abs/1104.3610`. arXiv:1104.3610 [astro-ph, physics:physics].

K. Lambeck, H. Rouby, A. Purcell, Y. Sun, and M. Sambridge. Sea level and global ice volumes from the Last Glacial Maximum to the Holocene. *Proceedings of the National Academy of Sciences*, 111(43):15296–15303, Oct. 2014. doi: 10.1073/pnas.1411762111. URL `https://www.pnas.org/doi/10.1073/pnas.1411762111`. Publisher: Proceedings of the National Academy of Sciences.

E. Legrain, F. Parrenin, and E. Capron. A gradual change is more likely to have caused the Mid-Pleistocene Transition than an abrupt event. *Communications Earth & Environment*, 4(1):1–10, Mar. 2023. ISSN 2662-4435. doi: 10.1038/s43247-023-00754-0. URL `https://www.nature.com/articles/s43247-023-00754-0`. Number: 1 Publisher: Nature Publishing Group.

G. Leloup and D. Paillard. Influence of the choice of insolation forcing on the results of a conceptual glacial cycle model. *Climate of the Past*, 18(3):547–558, Mar. 2022. ISSN 1814-9324. doi: 10.5194/cp-18-547-2022. URL `https://cp.copernicus.org/articles/18/547/2022/`. Publisher: Copernicus GmbH.

X. Ma, M. Yang, Y. Sun, H. Dang, W. Ma, J. Tian, Q. Jiang, L. Liu, X. Jin, and Z. Jin. The potential role of insolation in the long-term climate evolution since the early Pleistocene. *Global and Planetary Change*, 240:104526, Sept. 2024. ISSN 0921-8181. doi: 10.1016/j.gloplacha.2024.104526. URL `https://www.sciencedirect.com/science/article/pii/S0921818124001735`.

M. Milankovitch. Kanon der Erdbestrahlung und seine Anwendung auf das Eiszeitenproblem. *Royal Serbian Academy Special Publication*, 133:1–633, 1941. URL `https://cir.nii.ac.jp/crid/1572824499597240704`. Publisher: Israel Program for Sceientific Translation, U. S. Department of Commerce.

D. Paillard. The timing of Pleistocene glaciations from a simple multiple-state climate model. *Nature*, 391(6665):378–381, Jan. 1998. ISSN 1476-4687. doi: 10.1038/34891. URL `https://www.nature.com/articles/34891`. Publisher: Nature Publishing Group.

D. Paillard. Quaternary glaciations: from observations to theories. *Quaternary Science Reviews*, 107:11–24, Jan. 2015. ISSN 0277-3791. doi: 10.1016/j.quascirev.2014.10.002. URL `https://www.sciencedirect.com/science/article/pii/S0277379114003898`.

F. Parrenin and D. Paillard. Terminations VI and VIII ( 530 and  720 kyr BP) tell us the importance of obliquity and precession in the triggering of deglaciations. *Climate of the Past*, 8(6):2031–2037, Dec. 2012. ISSN 1814-9332. doi: 10.5194/cp-8-2031-2012. URL `https://cp.copernicus.org/articles/8/2031/2012/`.

S. Talento and A. Ganopolski. Reduced-complexity model for the impact of anthropogenic $CO_2$ emissions on future glacial cycles. *Earth System Dynamics*, 12(4):1275–1293, Nov. 2021. ISSN 2190-4979. doi: 10.5194/esd-12-1275-2021. URL `https://esd.copernicus.org/articles/12/1275/2021/`. Publisher: Copernicus GmbH.

P. C. Tzedakis, M. Crucifix, T. Mitsui, and E. W. Wolff. A simple rule to determine which insolation cycles lead to interglacials. *Nature*, 542(7642):427–432, Feb. 2017. ISSN 1476-4687. doi: 10.1038/nature21364. URL `https://www.nature.com/articles/nature21364`. Publisher: Nature Publishing Group.

M. Verbitsky and A. W. Omta. Rapid Communication: Middle Pleistocene Transition as a Phenomenon of Orbitally Enabled Sensitivity to Initial Values. *EGUsphere*, pages 1–9, July 2025. doi: 10.5194/egusphere-2025-3334. URL `https://egusphere.copernicus.org/preprints/2025/egusphere-2025-3334/`. Publisher: Copernicus GmbH.

M. Willeit, A. Ganopolski, R. Calov, and V. Brovkin. Mid-Pleistocene transition in glacial cycles explained by declining CO2 and regolith removal. *Science Advances*, 5(4):eaav7337, Apr. 2019. doi: 10.1126/sciadv.aav7337. URL `https://www.science.org/doi/full/10.1126/sciadv.aav7337`. Publisher: American Association for the Advancement of Science.

Y. Yokoyama, K. Lambeck, P. De Deckker, P. Johnston, and L. K. Fifield. Timing of the Last Glacial Maximum from observed sea-level minima. *Nature*, 406(6797):713–716, Aug. 2000. ISSN 1476-4687. doi: 10.1038/35021035. URL `https://www.nature.com/articles/35021035`. Publisher: Nature Publishing Group.